# DP-FedSOFIM: Differentially Private Federated Stochastic Optimization using Regularized Fisher Information Matrix

**Sidhant Nair**                                    *sid.nairiitd@gmail.com*
*Department of Mechanical Engineering, Indian Institute of Technology Delhi*

**Tanmay Sen**                                    *tanmay.sen@isical.ac.in*
*SQC & OR Unit, Indian Statistical Institute Kolkata*

**Mrinmay Sen**                                    *senmrinmay@alumni.iith.ac.in*
*Department of Artificial Intelligence, Indian Institute of Technology Hyderabad*

**Sayantan Banerjee**                                    *sayantanb@iimidr.ac.in*
*Operations Management & Quantitative Techniques Area,*
*Indian Institute of Management Indore*

**Reviewed on OpenReview:** *https://openreview.net/forum?id=aDzj9DrwAR*

## Abstract

Differentially private federated learning (DP-FL) often suffers from slow convergence under tight privacy budgets because the noise required for privacy preservation degrades gradient quality. Although second-order optimization can accelerate training, existing approaches for DP-FL face significant scalability limitations: Newton-type methods require clients to compute Hessians, while feature covariance methods scale poorly with model dimension. We propose **DP-FedSOFIM**, a simple and scalable Hessian approximation based second-order optimization method for DP-FL. The method constructs a regularized proxy for the Fisher information matrix at the server using only privatized aggregated gradients, capturing useful curvature information without requiring full Hessian computations or feature covariance estimation. Efficient rank-one updates based on the Sherman-Morrison formula enable communication costs proportional to the model size and require only $O(d)$ client side memory. Because all curvature and preconditioning operations are performed at the server on already privatized gradients, **DP-FedSOFIM** introduces no additional privacy cost beyond the underlying privatized gradient release mechanism. Experiments on CIFAR-10 and PathMNIST demonstrate that DP-FedSOFIM converges faster and consistently achieves higher accuracy than several competitive differentially private federated learning baselines across a wide range of privacy budgets.

## 1 Introduction

Federated learning (FL) has emerged as a widely adopted paradigm for training machine learning models across geographically distributed data sources without centralizing raw data (McMahan et al., 2017a; Kairouz & McMahan, 2021; Li et al., 2020a). Its appeal is especially pronounced in privacy sensitive domains such as healthcare and finance, where regulatory constraints and ethical obligations make direct data sharing infeasible (Rieke et al., 2020; Sheller et al., 2020). Yet the mere absence of raw data transfer does not constitute a formal privacy guarantee. A growing body of work has demonstrated that shared model gradients can leak sensitive training information through gradient inversion attacks (Zhu et al., 2019; Geiping et al., 2020), motivating the integration of rigorous privacy mechanisms into the federated training pipeline.

Differential privacy (DP) provides the gold standard for bounding information leakage in iterative learning algorithms (Dwork et al., 2006; 2014). The seminal DP-SGD framework of Abadi et al. (2016) operationalized

DP for deep learning via per example gradient clipping and calibrated Gaussian noise addition, and introduced the moments accountant for tight privacy composition across training steps. Extending this approach to federated settings, McMahan et al. (2017b) proposed DP-FedAvg, which enforces user level differential privacy by clipping and noising client updates before aggregation. Subsequent work has sharpened the privacy utility tradeoff through more expressive accounting frameworks such as Rényi differential privacy (RDP) (Mironov, 2017), Gaussian DP (Dong et al., 2022), and privacy amplification by subsampling (Balle & Wang, 2018; Wang et al., 2019).

Despite these advances, differentially private federated learning (DP-FL) faces a persistent convergence–privacy tradeoff. To satisfy an $(\epsilon, \delta)$-DP guarantee, gradient updates must be clipped and perturbed with Gaussian noise scaled to the clipping threshold (Abadi et al., 2016). In the federated setting where communication costs tightly limit the number of rounds and full client participation amplifies per round noise this perturbation can dominate the true gradient signal entirely, causing convergence to stall or degrade, particularly under stringent privacy budgets ($\epsilon < 2$) (Andrew et al., 2021). The problem is further compounded by client drift in heterogeneous data regimes (Karimireddy et al., 2020; Li et al., 2020b), where local gradients diverge from the global objective, and by the cumulative effect of noise composition over many rounds. These challenges together create a regime tight privacy, many clients, limited rounds where even well tuned first-order methods such as DP-FedAvg and DP-FedGD suffer severe utility loss.

A natural remedy is to exploit second-order curvature information, which can accelerate convergence by rescaling gradients along directions of high curvature precisely the directions where noise corrupted first-order steps are most wasteful. Adaptive methods such as Adam (Kingma & Ba, 2014) and AdaGrad (Duchi et al., 2011) pursue this idea via diagonal curvature estimates, and server side adaptive methods such as FedAdam and FedYogi (Reddi et al., 2020) carry this to federated settings without requiring second-order computation on clients. However, these methods do not explicitly leverage geometric curvature information and provide limited benefit under the anisotropic noise introduced by DP clipping. Exact second-order methods, including Newton-type approaches such as FedNL (Safaryan et al., 2022) and GIANT (Wang et al., 2018), as well as the Kronecker-factored curvature approximation K-FAC (Martens & Grosse, 2015), require clients to compute or transmit $O(d^2)$ Hessian or feature covariance information per round an $O(d)$-fold overhead over first-order methods that is prohibitive for high dimensional models and resource constrained edge devices.

The most closely related prior work, DP-FedNew (Krouka et al., 2025), introduces a second-order preconditioner for DP-FL based on local feature covariance matrices. While DP-FedNew demonstrates significant convergence speedups over DP-FedGD, its requirement that each client maintain and communicate an $O(d^2)$ feature covariance matrix limits its applicability to low dimensional generalized linear models and makes it impractical at the parameter scales of modern transfer learning pipelines. Moreover, computing second-order information at the client level introduces additional sensitivity that must be accounted for in the privacy analysis, potentially requiring tighter clipping or larger noise multipliers.

We introduce **DP-FedSOFIM** (Differentially Private Federated Stochastic Optimization using Regularized Fisher Information Matrix), a framework that resolves the scalability bottleneck of prior second-order DP-FL methods by relocating all curvature estimation and preconditioning entirely to the server. DP-FedSOFIM builds an online, rank one approximation to the Fisher Information Matrix (FIM) using only the privatized, aggregated gradients already available at the server, maintaining the inverse efficiently via the Sherman-Morrison formula (Sherman & Morrison, 1950) at $O(d)$ cost per round. Because all curvature computation is applied exclusively to already privatized quantities, the post-processing theorem (Dwork et al., 2014) guarantees that DP-FedSOFIM preserves the same $(\epsilon, \delta)$-DP guarantee as the underlying DP-FedGD baseline at no additional privacy cost.

While second-order optimization has long been known to accelerate convergence in centralized learning, extending such ideas to differentially private federated learning introduces several technical difficulties. In federated systems, curvature information cannot be computed from centralized data and must instead be inferred from aggregated client updates that are themselves corrupted by privacy noise. This creates a fundamental tension: reliable curvature estimation typically requires rich second-order statistics, yet differential privacy and communication constraints restrict what information can be transmitted from clients. Classical second-order approaches rely on dense Hessian or Fisher matrices with $O(d^2)$ memory and computation,

rendering them impractical in large-scale federated environments. From a statistical perspective, the problem is further complicated by the fact that the curvature signal must be recovered from noisy gradient observations whose variance scales with the privacy mechanism.

The DP-FedSOFIM framework addresses this challenge by exploiting the structure of the aggregated gradient dynamics. Rather than attempting to estimate a full curvature matrix, we construct a rank-one Fisher proxy from the momentum buffer, which aggregates gradient information across rounds while attenuating differential privacy noise through exponential averaging. This structure enables an exact Sherman–Morrison inversion, yielding a curvature-adaptive preconditioner that can be applied in $O(d)$ time and memory. The resulting algorithm therefore integrates curvature-aware optimization with differential privacy in a manner that preserves the communication efficiency required in cross-device federated learning, while still capturing the dominant curvature directions that govern convergence in ill-conditioned optimization landscapes.

Our key contributions are:

- **Server-Side Second-Order Preconditioning:** We design a curvature-aware natural-gradient preconditioner constructed entirely from privatized aggregated gradients, eliminating the need for client-side second-order computation.

- **Efficient $O(d)$ Implementation:** By leveraging the Sherman-Morrison formula for low-rank matrix updates (Sherman & Morrison, 1950), we achieve $O(d)$ computational complexity per round, making our approach scalable to high-dimensional models.

- **Privacy Preservation via Post-Processing:** We prove that server-side preconditioning preserves $(\epsilon, \delta)$-DP guarantees through the post-processing theorem (Dwork et al., 2014), as the FIM is computed on already-privatized gradient aggregates.

- **Empirical Validation:** Experiments on CIFAR-10 and PathMNIST with ResNet-20 frozen features and $n = 20$ clients demonstrate that DP-FedSOFIM consistently outperforms all baselines across all four privacy budgets ($\varepsilon \in \{0.5, 1, 5, 10\}$). It attains the best round-10 accuracy in seven of eight dataset/privacy regimes, with a $\sim 5\times$ reduction in rounds needed to reach 95% of DP-FedGD's final accuracy. On CIFAR-10, final-round gains over DP-FedGD reach up to $+4.55\%$ ($\varepsilon = 10$); on PathMNIST, where curvature-aware preconditioning is most effective on the ill-conditioned medical imaging task, gains reach up to $+5.16\%$ ($\varepsilon = 10$). DP-FedSOFIM achieves the best final-round accuracy in six of eight regimes and degrades gracefully under stringent privacy, retaining its convergence advantage down to $\varepsilon = 0.5$.

The remainder of this paper is organized as follows. Section 2 reviews related work on differentially private federated learning, second-order optimization in federated settings, and efficient matrix inversion techniques. Section 3 presents the DP-FedSOFIM algorithm and its efficient implementation. Section 4 provides theoretical convergence guarantees and privacy analysis. Section 5 reports experimental results, and Section 6 concludes with discussion and future directions. We provide detailed proofs of all results in the Appendix, along with several other methodological plus theoretical details including computational complexity analysis and runtime analysis, and additional experimental results.

## 2 Related Work

### 2.1 Differentially Private Federated Learning

Differential privacy in federated learning has been extensively studied since the foundational contributions of Dwork et al. (2006) and Dwork et al. (2014), who established the formal framework and core mechanisms including the Gaussian and Laplace mechanisms. Abadi et al. (2016) operationalized DP for deep learning via per-example gradient clipping and the moments accountant, providing the first practical technique for tracking privacy loss under iterative training. Building on this, McMahan et al. (2017b) proposed DP-FedAvg, extending user-level differential privacy to the federated setting by clipping and noising client updates prior to aggregation.

Subsequent work has focused on tightening the privacy-utility tradeoff through advanced accounting techniques. Mironov (2017) introduced Rényi differential privacy (RDP), which provides tighter composition bounds than the moments accountant and has become the standard tool for multi-round privacy analysis in DP-FL. Dong et al. (2022) developed Gaussian DP (GDP) as an alternative analytic framework that yields tight guarantees for the Gaussian mechanism specifically, and is the foundation for the Hockey-Stick divergence accounting used in our experiments. Privacy amplification by subsampling (Balle & Wang, 2018; Wang et al., 2019) provides further improvement when clients or data are sampled randomly, reducing the effective privacy cost per round and allowing smaller noise multipliers for a given $(\epsilon, \delta)$ target.

On the optimization side, Andrew et al. (2021) demonstrated that adaptive clipping of client updates, where the clipping threshold is tuned to track a target gradient norm quantile, can substantially improve convergence under DP constraints without additional privacy cost. The interaction between client heterogeneity and differential privacy has also received attention: Karimireddy et al. (2020) introduced SCAFFOLD to correct for client drift via control variates, and our experiments show that this correction degrades under high privacy noise as the control variates become corrupted, an empirical finding we analyze in Section 5. Recent work has also studied personalized federated learning under differential privacy (Hu et al., 2020; Noble et al., 2022; Wei et al., 2023), heterogeneous privacy budgets across clients (Liu et al., 2022), and the interaction between local differential privacy and central DP (Naseri et al., 2022).

Second-order methods for DP-FL were first systematically explored by Krouka et al. (2025), who proposed DP-FedNew. DP-FedNew computes local feature covariance matrices at each client as an approximation to the FIM for generalized linear models, achieving substantial speedups over DP-FedGD. However, its requirement for local $O(d^2)$ memory per client limits scalability to high-dimensional models and resource-constrained devices. Furthermore, computing second-order statistics at the client introduces additional sensitivity that must be accounted for in the privacy analysis. Our work directly addresses these limitations by moving curvature estimation entirely to the server, preserving $O(d)$ client-side complexity and incurring no additional privacy cost by the post-processing theorem.

## 2.2 Adaptive and Second-Order Optimizers in Federated Learning

Adaptive optimization methods such as Adam (Kingma & Ba, 2014) and AdaGrad (Duchi et al., 2011) have become standard in centralized deep learning by maintaining per-parameter running estimates of gradient moments to adaptively rescale step sizes. Extending these methods to federated settings is non-trivial due to the need to aggregate adaptive statistics across heterogeneous clients (Reddi et al., 2020). FedAdam and FedYogi (Reddi et al., 2020) maintain server-side adaptive learning rates using pseudo-gradients formed by the aggregated client updates, but do not explicitly leverage second-order curvature information and therefore provide limited benefit in the highly noisy DP regime.

Beyond adaptive gradient methods, another line of work studies the Alternating Direction Method of Multipliers (ADMM) and related primal-dual optimization frameworks for distributed and federated learning. Several works have explored ADMM-based optimization for distributed and federated learning. FedADMM (Gong et al., 2022) introduces a primal-dual federated optimization framework that uses dual variables to address statistical and system heterogeneity while maintaining communication costs comparable to FedAvg. More recently, Möllenhoff et al. (2025) established a Bayesian interpretation of federated ADMM and derived Newton-like and Adam-like variants through variational Bayesian duality. ADMM based techniques have also been applied to communication efficient second-order federated optimization; for example, FedNPG-ADMM (Lan et al., 2023) approximates natural policy gradient directions while reducing communication complexity from $O(d^2)$ to $O(d)$. More broadly, ADMM has been studied as an alternative to gradient-based deep learning optimization and shown to possess favorable convergence properties in nonconvex neural network training (Zeng et al., 2021).

Natural gradient descent (Amari, 1998) uses the Fisher Information Matrix as a preconditioner, providing a principled, reparameterization-invariant approach to incorporating curvature. As established by Martens (2020), the empirical Fisher Information Matrix computed as the outer product of gradient vectors provides a tractable approximation that retains the key geometric properties of the true FIM and underpins many practical second-order methods. Computing and inverting the exact FIM requires $O(d^2)$ storage and $O(d^3)$

inversion, making it impractical for large-scale models. Approximation methods such as K-FAC (Martens & Grosse, 2015) and diagonal Fisher (Pascanu & Bengio, 2014) reduce these costs but introduce additional approximation errors and still require $O(d^2)$ factors per Kronecker block.

Several Newton-type methods have been proposed for distributed and federated settings. GIANT (Wang et al., 2018) communicates local Newton directions and performs a global approximate Newton step, achieving $O(d)$ communication per round but requiring $O(d^2)$ local Hessian computation. FedNL (Safaryan et al., 2022) uses compressed Hessian communication with contractive compressors, providing condition-number-independent local convergence, but requires clients to maintain and transmit Hessian information. SHED (Dal Fabbro et al., 2024) incrementally shares eigenvalue-eigenvector pairs to reconstruct local Hessians at the server, reducing communication at the cost of stale curvature estimates. None of these methods address the differentially private setting, where client-side second-order computation introduces additional sensitivity that must be carefully managed.

DP-FedNew (Krouka et al., 2025) computes local feature covariance matrices at each client, which approximates the FIM for generalized linear models. While effective, this approach requires each client to store and communicate $O(d^2)$ parameters. In contrast, DP-FedSOFIM constructs a global FIM proxy on the server using aggregated noisy gradients, achieving $O(d)$ memory complexity while preserving the benefits of second-order preconditioning. Crucially, the server-side construction means that privacy and preconditioning are cleanly separated: any future improvement in privacy accounting or amplification applies equally to DP-FedSOFIM and DP-FedGD, and the accuracy gains from preconditioning are preserved across all privacy regimes.

### 2.3 Efficient Matrix Inversion Techniques

The Sherman-Morrison formula (Sherman & Morrison, 1950) provides an efficient method for updating matrix inverses when a matrix is modified by a rank-one perturbation, reducing an $O(d^3)$ inversion to an $O(d)$ update given the previous inverse. This technique has been applied in online convex optimization (Hazan et al., 2007), where it enables logarithmic regret algorithms that maintain curvature estimates in $O(d)$ space, and in recursive least squares (Haykin, 2002) for sequential parameter estimation. The broader family of rank-$k$ updates is handled by the Woodbury matrix identity (Woodbury, 1950), of which Sherman-Morrison is the rank-one special case.

In the federated learning context, efficient low-rank matrix maintenance is essential for any server-side curvature method that must update its estimate every round. The SOFIM framework (Sen et al., 2024) demonstrated the viability of rank-one FIM approximations in the non-private federated setting, motivating our extension to the differentially private regime. We leverage the Sherman-Morrison formula to maintain the inverse of the regularized FIM in $O(d)$ per round, making DP-FedSOFIM computationally competitive with first-order baselines while delivering the convergence benefits of natural gradient descent.

## 3 Methodology

### 3.1 Federated Learning Formulation

We consider a synchronous federated learning system consisting of $n$ clients indexed by $i \in [n] := \{1, \ldots, n\}$. Client $i$ holds a private dataset $\mathcal{D}_i = \{(x_{i,j}, y_{i,j})\}_{j=1}^{|\mathcal{D}_i|}$. The global federated dataset is therefore $\mathcal{D} = \{\mathcal{D}_1, \ldots, \mathcal{D}_n\}$. The goal is to minimize the empirical risk

$$F(\theta) = \frac{1}{n} \sum_{i=1}^{n} F_i(\theta), \qquad F_i(\theta) = \frac{1}{|\mathcal{D}_i|} \sum_{(x,y) \in \mathcal{D}_i} \ell(\theta; x, y),$$

where $\theta \in \mathbb{R}^d$ denotes the model parameters and $\ell(\theta; x, y)$ is a differentiable loss function.

Optimization proceeds in communication rounds. At round $t$, clients compute local gradient information and transmit privatized updates to a central server. The server aggregates these updates and performs a global parameter update.

For clarity of exposition we assume full participation, i.e. all clients participate at every round. The mechanism itself does not rely on this assumption. Under partial participation, a subset $S_t \subset [n]$ of clients may be sampled at each round, in which case aggregation is performed over $S_t$ rather than $[n]$. The remainder of the algorithm remains unchanged.

*Remark* 3.1 (Notation). Throughout the paper we use subscript time indices for all iterates and buffers. For example $\theta_t$ denotes the global parameter vector at round $t$, $G_t \in \mathbb{R}^d$ the aggregated gradient, and $M_t \in \mathbb{R}^d$ the server-side momentum buffer. The number of clients is denoted by $n$; the symbol $K$ commonly used in federated learning literature is set equal to $n$ here under full participation.

### 3.2 Privacy Model: Record-Level Differential Privacy

We adopt the record-level differential privacy framework introduced by Abadi et al. (2016). In this model the protected unit is an individual training example.

**Definition 3.2** (Neighboring Datasets). Two client datasets $D_i$ and $D_i'$ are said to be neighboring if they differ in exactly one data record while having the same size. Formally, $D_i' = (D_i \setminus \{z\}) \cup \{z'\}$ for some records $z$ and $z'$.

This *replace-one* adjacency definition keeps the dataset size fixed, which simplifies sensitivity analysis for the normalized gradient releases used by the algorithm.

**Definition 3.3** (($\varepsilon, \delta$)-Differential Privacy). A randomized mechanism $\mathcal{M}$ satisfies ($\varepsilon, \delta$)-DP if for all neighboring datasets $\mathcal{D}, \mathcal{D}'$ and all measurable sets $S$,

$$\mathbb{P}[\mathcal{M}(\mathcal{D}) \in S] \leq e^\varepsilon \mathbb{P}[\mathcal{M}(\mathcal{D}') \in S] + \delta.$$

Record-level DP provides protection for individual training examples even within a participating client's dataset. This notion of privacy is standard in privacy-preserving deep learning and federated optimization (Abadi et al., 2016).

### 3.3 Baseline Mechanism: DP-FedGD

We first recall the differentially private federated gradient descent algorithm (DP-FedGD), which forms the foundation of our method.

At round $t$, each client computes per-example gradients $g(x, y) = \nabla \ell(\theta_t; x, y)$.

**Per-example gradient clipping.** To control sensitivity, gradients are clipped to an $\ell_2$ radius $C_g$:

$$\bar{g}(x, y) = g(x, y) \cdot \min\left(1, \frac{C_g}{\|g(x, y)\|_2}\right). \tag{1}$$

This guarantees $\|\bar{g}(x, y)\|_2 \leq C_g$, which yields a uniform $\ell_2$ sensitivity bound for the summed gradients.

**Client-side Gaussian perturbation.** Let $S_{i,t} = \sum_{(x,y) \in \mathcal{D}_i} \bar{g}(x, y)$ denote the sum of clipped gradients on client $i$. Client $i$ releases the noisy normalized update

$$g_{i,t} = \frac{1}{|\mathcal{D}_i|}\left(S_{i,t} + E_{i,t}\right), \qquad E_{i,t} \sim \mathcal{N}\left(0, \frac{(C_g \sigma_g)^2}{n} I_d\right). \tag{2}$$

The noise multiplier $\sigma_g$ controls the privacy–utility tradeoff.

**Server aggregation.** The server aggregates client updates as $G_t = n^{-1} \sum_{i=1}^{n} g_{i,t}$. The global model is updated via $\theta_{t+1} = \theta_t - \eta_t G_t$.

Gradient clipping bounds the $\ell_2$ sensitivity of the client update, while Gaussian noise is added according to the Gaussian mechanism. The resulting differential privacy guarantees depend on the clipping threshold,

noise scale, dataset normalization, and the privacy accountant used across rounds. A formal derivation of the $(\varepsilon, \delta)$-privacy guarantees for the full DP-FedSOFIM procedure is provided in Section 4.10. However, DP-FedGD is fundamentally a first-order optimization method. In ill-conditioned problems, anisotropic curvature and privacy-induced noise can significantly slow convergence. The concrete privacy parameters of the Gaussian release are derived in Appendix F, while Section 4.10 shows that the SOFIM preconditioning introduces no additional privacy loss beyond this release mechanism.

### 3.4 DP-FedSOFIM: Privacy Preserving Second Order Federated Optimization

To address this limitation we propose **DP-FedSOFIM**, which augments DP-FedGD with a curvature-aware server-side preconditioning step inspired by the SOFIM framework of Sen et al. (2024). The original SOFIM method constructs scalable structured Fisher approximations for non-private federated learning. Our contribution extends this idea to the differentially private regime while preserving record-level privacy guarantees. The design philosophy is deliberately conservative with respect to privacy. The client-side mechanism remains identical to DP-FedGD: clipping and Gaussian perturbation are unchanged. All curvature information is constructed solely from already-privatized aggregated gradients. Consequently, privacy guarantees follow directly from the post-processing property of differential privacy.

#### 3.4.1 Natural Gradient Motivation

For a parametric model $p(y|x; \theta)$, the Fisher Information Matrix (FIM) is defined as

$$\mathcal{I}(\theta) = \mathbb{E}\left[\nabla \log p(y|x; \theta) \nabla \log p(y|x; \theta)^\top\right].$$

Natural gradient descent (Amari, 1998) updates parameters according to

$$\theta_{t+1} = \theta_t - \eta_t \mathcal{I}(\theta_t)^{-1} \nabla F(\theta_t),$$

which corresponds to steepest descent under the Riemannian metric induced by the Fisher information. Exact computation of the FIM is typically infeasible in federated settings. We therefore adopt a structured rank-one approximation inspired by the SOFIM framework.

#### 3.4.2 Server-Side Curvature Proxy

The server maintains an exponential moving average of aggregated gradients

$$M_t = \beta M_{t-1} + (1 - \beta)G_t, \qquad M_{-1} = \mathbf{0}_d, \quad \beta \in [0, 1),$$

where $G_t$ denotes the aggregated client update at communication round $t$. The momentum buffer stabilizes curvature estimation by smoothing stochastic fluctuations arising from client sampling, data heterogeneity, and privacy-induced noise.

Using the momentum estimate, we construct the curvature proxy

$$\widehat{\mathcal{I}}_t = M_t M_t^\top + \rho I_d,$$

where $\rho > 0$ is a regularization parameter that ensures positive definiteness and numerical stability.

The proposed approximation is intentionally low-rank and is not intended to recover the full Fisher Information Matrix (FIM), which is generally impractical to compute and store in large scale federated learning systems. Instead, the momentum estimate $M_t$ serves as a stable and noise-reduced aggregation of historical gradient information, enabling the construction of a structured Fisher-inspired curvature surrogate. By accumulating information across communication rounds, the resulting proxy captures useful curvature characteristics while remaining computationally efficient.

A key advantage of the rank-one structure is that it enables efficient inverse preconditioning through the Sherman–Morrison formula, thereby avoiding the $O(d^2)$ memory and computational costs associated with

full-matrix Fisher approximations. Consequently, the preconditioner $\widehat{\mathcal{I}}_t^{-1}$ introduces anisotropic rescaling of gradient updates, yielding curvature aware optimization at essentially first-order computational complexity. Importantly, the momentum buffer is used solely to construct the curvature proxy and does not directly determine the model update. Instead, the model update is obtained by preconditioning the aggregated gradient with the inverse Fisher-inspired curvature proxy $\widehat{\mathcal{I}}_t^{-1}$, resulting in a curvature-aware optimization step.

### 3.4.3 Efficient Inversion via Sherman-Morrison

Since $\widehat{\mathcal{I}}_t$ is a rank-one perturbation of $\rho I_d$, its inverse admits the closed form

$$H_t = \widehat{\mathcal{I}}_t^{-1} = \frac{1}{\rho} I_d - \frac{M_t M_t^\top}{\rho^2 + \rho \|M_t\|_2^2},$$

which follows from the Sherman–Morrison identity (Sherman & Morrison, 1950).

Applying this inverse to the gradient yields

$$H_t G_t = \frac{1}{\rho} G_t - \frac{M_t (M_t^\top G_t)}{\rho^2 + \rho \|M_t\|_2^2}.$$

This expression requires only two inner products and vector operations, leading to an $O(d)$ computational cost per round. We discuss this in more detail in Appendix D.

**Preconditioned server update.** The global model is updated via

$$\theta_{t+1} = \theta_t - \eta_t H_t G_t.$$

**Privacy preservation.** The matrix $H_t$ is a deterministic function of the privatized aggregated gradient $G_t$ and the previous server state $(\theta_t, M_{t-1})$. Since differential privacy is closed under post-processing, the mapping $G_t \mapsto H_t G_t$ does not incur any additional privacy loss. Therefore the DP-FedSOFIM update inherits the same $(\varepsilon, \delta)$-DP guarantee as the underlying DP-FedGD mechanism.

**Interpretation.** DP-FedSOFIM can be viewed as a privacy-compatible, natural-gradient-motivated method in which curvature information is estimated entirely from privatized signals. The algorithm balances three competing factors: (i) clipping-induced bias, (ii) Gaussian noise required for privacy, and (iii) curvature-aware rescaling intended to mitigate ill-conditioning.

**Warm-started preconditioning.** Under very tight privacy budgets, the first few privatized aggregate gradients may be dominated by Gaussian noise, and the momentum buffer $M_t$ may therefore be unstable during the early rounds. A simple stabilization is to introduce the preconditioner gradually. Define

$$q_t = (1 - \lambda_t) \rho^{-1} G_t + \lambda_t H_t G_t, \qquad 0 \le \lambda_t \le 1, \qquad \lambda_t \uparrow 1,$$

and update

$$\theta_{t+1} = \theta_t - \eta_t q_t.$$

For $\lambda_t = 0$, the method reduces to a scaled DP-FedGD step, while for $\lambda_t = 1$, it recovers DP-FedSOFIM. This warm-started variant can reduce early-round instability under stringent privacy budgets by allowing the momentum buffer to accumulate a more stable privatized gradient history before full anisotropic preconditioning is applied. Since $q_t$ is still a deterministic function of the already privatized aggregate $G_t$ and the previous server state, the warm-started update incurs no additional privacy cost.

### 3.5 Algorithmic Summary

Algorithm 1 summarizes the complete DP-FedSOFIM procedure. Client side operations coincide exactly with DP-FedGD; the only modification occurs on the server through the curvature-aware preconditioned update.

---

**Algorithm 1** DP-FedSOFIM

---

    **Input:** $\theta_0$, learning rate $\eta$, clipping threshold $C_g$, noise multiplier $\sigma_g$, momentum $\beta$, regularization $\rho$, rounds $T$

    **Server initialization:** $M_{-1} = \mathbf{0}_d$

  1: **for** $t = 0, \ldots, T - 1$ **do**

      *// Client side (identical to DP-FedGD)*

  2:    **for** each client $i \in [n]$ **do**

  3:        $S_{i,t} = \sum\limits_{(x,y) \in \mathcal{D}_i} \dfrac{\nabla \ell(\theta_t; x, y)}{\max\big(1, \, \|\nabla \ell(\theta_t; x, y)\|_2 / C_g\big)}$

  4:        Sample $E_{i,t} \sim \mathcal{N}\big(0, \, (C_g \sigma_g)^2 I_d / n\big)$

  5:        $g_{i,t} = \big(S_{i,t} + E_{i,t}\big) / |\mathcal{D}_i|$

  6:    **end for**

      *// Server side (curvature-aware update)*

  7:    $G_t = \dfrac{1}{n} \sum\limits_{i=1}^{n} g_{i,t}$

  8:    $M_t = \beta M_{t-1} + (1 - \beta) G_t$

  9:    $H_t G_t = \frac{1}{\rho} G_t - \dfrac{M_t (M_t^\top G_t)}{\rho^2 + \rho \|M_t\|_2^2}$

10:    $\theta_{t+1} = \theta_t - \eta_t H_t G_t$

11: **end for**

---

## 4 Convergence and Privacy Analysis

This section establishes optimization and privacy guarantees for DP-FedSOFIM. All algorithmic quantities are exactly those defined in Section 3; in particular, the server receives the privatized aggregated gradient $G_t = \frac{1}{n} \sum_{i=1}^{n} g_{i,t}$, updates the momentum buffer, forms the rank-one preconditioner, and performs the parameter update $\theta_{t+1} = \theta_t - \eta H_t G_t$.

The algorithm allows a general stepsize sequence $\{\eta_t\}_{t \geq 0}$. For the convergence analysis below, we specialize to the constant stepsize regime $\eta_t \equiv \eta$. A technical feature of DP-FedSOFIM is that the preconditioner is formed from the *current* privatized aggregate rather than from a delayed or auxiliary quantity. Consequently, $H_t$ and $G_t$ are coupled within the same round. The analysis below handles this same-step dependence directly.

To analyze the stochastic behavior of the update, we decompose the aggregated gradient into a clipped population component and a privacy noise term.

We define the aggregated clipped gradient

$$g_{\mathrm{clip}}(\theta_t) := \frac{1}{n} \sum_{i=1}^{n} \frac{1}{|\mathcal{D}_i|} \sum_{(x,y) \in \mathcal{D}_i} \bar{g}(x, y),$$

where $\bar{g}(x, y)$ denotes the clipped per-example gradient defined in (1). The server aggregate can therefore be written as

$$G_t = g_{\mathrm{clip}}(\theta_t) + \xi_t = \nabla F(\theta_t) + \zeta_t + \xi_t, \tag{3}$$

where $\zeta_t := g_{\mathrm{clip}}(\theta_t) - \nabla F(\theta_t)$ is the clipping bias and $\xi_t$ is the noise introduced by the privacy mechanism.

From the client update rule (2), the noise term admits the explicit representation

$$\xi_t = \frac{1}{n} \sum_{i=1}^{n} \frac{E_{i,t}}{|\mathcal{D}_i|}, \tag{4}$$

where the injected noises are independent Gaussian vectors

$$E_{i,t} \sim \mathcal{N}\left(0, \frac{(C_g \sigma_g)^2}{n} I_d\right).$$

Let $\mathcal{F}_t$ denote the *pre-round filtration*, i.e., the sigma-field generated by all algorithmic randomness up to the start of round $t$. In particular, $\theta_t$, $M_{t-1}$, $\nabla F(\theta_t)$, $g_{\text{clip}}(\theta_t)$, and $\zeta_t$ are $\mathcal{F}_t$-measurable, whereas $\xi_t$ is the fresh noise generated during round $t$. This gives, $\mathbb{E}[\xi_t \mid \mathcal{F}_t] = 0$.

Using independence across clients, the covariance of $\xi_t$ is $\mathbb{E}[\xi_t \xi_t^\top \mid \mathcal{F}_t] = \nu_t^2 I_d$, where

$$\nu_t^2 = \frac{(C_g \sigma_g)^2}{n^3} \sum_{i=1}^n \frac{1}{|\mathcal{D}_i|^2}. \tag{5}$$

**Lemma 4.1** (Conditional mean under preconditioning). *Let $G_t = g_{\text{clip}}(\theta_t) + \xi_t$ and $H_t = (\rho I_d + M_t M_t^\top)^{-1}$. Then, conditional on the pre-round filtration $\mathcal{F}_t$,*

$$\mathbb{E}[G_t \mid \mathcal{F}_t] = g_{\text{clip}}(\theta_t).$$

*Equivalently,*

$$\mathbb{E}[G_t - g_{\text{clip}}(\theta_t) \mid \mathcal{F}_t] = 0.$$

*Proof.* By (3), $G_t = g_{\text{clip}}(\theta_t) + \xi_t$. Since $g_{\text{clip}}(\theta_t)$ is $\mathcal{F}_t$-measurable and $\mathbb{E}[\xi_t \mid \mathcal{F}_t] = 0$,

$$\mathbb{E}[G_t \mid \mathcal{F}_t] = g_{\text{clip}}(\theta_t) + \mathbb{E}[\xi_t \mid \mathcal{F}_t] = g_{\text{clip}}(\theta_t).$$

$\square$

Thus the aggregated gradient estimator $G_t$ is conditionally unbiased for the clipped population gradient and has isotropic Gaussian noise with variance parameter $\nu_t^2$.

### 4.1 Assumptions

**Assumption 4.2** ($L$-smoothness). Each local objective $F_i$ is $L$-smooth. Consequently, the global objective $F$ is $L$-smooth:

$$\|\nabla F(\theta) - \nabla F(\theta')\|_2 \le L\|\theta - \theta'\|_2, \qquad \forall\, \theta, \theta' \in \mathbb{R}^d.$$

Equivalently, for all $\theta, \Delta \in \mathbb{R}^d$,

$$F(\theta + \Delta) \le F(\theta) + \langle \nabla F(\theta), \Delta \rangle + \frac{L}{2}\|\Delta\|_2^2. \tag{6}$$

**Assumption 4.3** ($\mu$-strong convexity). The global objective $F$ is $\mu$-strongly convex for some $\mu > 0$, i.e.,

$$F(\theta') \ge F(\theta) + \langle \nabla F(\theta), \theta' - \theta \rangle + \frac{\mu}{2}\|\theta' - \theta\|_2^2, \qquad \forall\, \theta, \theta' \in \mathbb{R}^d.$$

In particular,

$$\|\nabla F(\theta)\|_2^2 \ge 2\mu\big(F(\theta) - F(\theta^\star)\big), \qquad \forall\, \theta \in \mathbb{R}^d, \tag{7}$$

where $\theta^\star$ denotes the unique minimizer of $F$.

*Remark* 4.4 (Strong Convexity in Frozen-Feature Models). When the feature extractor is frozen and only a linear prediction head is trained, strong convexity arises naturally in several common settings. For squared-loss linear regression, strong convexity holds whenever the feature covariance matrix is positive definite. For multiclass softmax models, strong convexity can be ensured by adding an explicit $\ell_2$ regularization term or by imposing identifiability constraints on the parameterization. These conditions are standard in transfer learning scenarios where only the final prediction layer is optimized.

**Assumption 4.5** (Bounded clipping bias). There exists $\zeta_{\max} \ge 0$ such that $\|\zeta_t\|_2 \le \zeta_{\max}$, $\forall\, t \ge 0$.

**Assumption 4.6** (Bounded gradient noise). Let $\xi_t$ denote the stochastic noise defined in (4). There exists a constant $\nu^2 < \infty$ such that $\mathbb{E}[\xi_t \mid \mathcal{F}_t] = 0$, $\mathbb{E}[\|\xi_t\|_2^2 \mid \mathcal{F}_t] \le d\nu^2$ for all iterations $t$.

*Remark* 4.7 (Noise magnitude in DP-FedSOFIM). For the mechanism defined in Section 3.3, the variance parameter in Assumption 4.6 is given explicitly by $\nu_t^2 = (C_g \sigma_g)^2 \sum_{i=1}^n |\mathcal{D}_i|^{-2}/n^3$. In the common case where all clients have equal dataset size $|\mathcal{D}_i| = m$, this simplifies to $\nu_t^2 = (C_g \sigma_g)^2/(n^2 m^2)$. Thus the variance of the aggregated noise decreases as $O(n^{-2})$ when client dataset sizes are equal, while the standard deviation decreases as $O(n^{-1})$.

**Assumption 4.8** (Bounded gradient norm along the iterate sequence). There exists $G_{\max} > 0$ such that $\|\nabla F(\theta_t)\|_2 \leq G_{\max}$, $\forall\, t \geq 0$.

Assumption 4.8 is standard in analyses of adaptive or state-dependent preconditioners. In the present setting it is used only to control the same-step coupling between the current preconditioner $H_t$ and the current privatized aggregate $G_t$. In the frozen-feature regime, it holds whenever features are bounded and the trainable head is confined to a bounded parameter region, or when explicit regularization prevents parameter growth.

*Remark* 4.9 (Role of bounded bias and bounded gradients). The bounded-bias and bounded-gradient assumptions play different roles in the analysis. The bounded-bias assumption controls the discrepancy between the privatized aggregate update and the target population gradient. In the one-step DP-FedGD setting this discrepancy is the clipping bias $\zeta_t = g_{\text{clip}}(\theta_t) - \nabla F(\theta_t)$, whereas in mini-batch or multi-local-step variants it may also contain stochastic approximation error and local-update drift. This assumption allows the descent bound to separate the optimization error from the perturbation introduced by clipping, privacy noise, and local computation.

The bounded-gradient assumption is used for a different reason. In DP-FedSOFIM, the preconditioner $H_t = (\rho I_d + M_t M_t^\top)^{-1}$ is formed using the same-round privatized aggregate $G_t$ through the momentum recursion $M_t = \beta M_{t-1} + (1 - \beta)G_t$. Thus $H_t$ and $G_t$ are statistically coupled within the same communication round. The bounded-gradient condition is used to control this same-step dependence, specifically the term arising from $\nabla F(\theta_t)^\top H_t \nabla F(\theta_t)$, where the lower bound contains a correction proportional to $G_{\max}^2 \|M_t\|_2^2/\rho^2$. The assumption is therefore not introduced to make the objective globally well-behaved, but to control the interaction between the current privatized gradient, the server-side momentum buffer, and the adaptive preconditioner. In the frozen-feature linear-head setting, such a condition is natural when features are bounded and the trainable parameter is regularized or kept in a bounded region.

The following lemma shows that the DP noise mechanism satisfies the bounded-noise assumption used in the optimization analysis. We provide the proof of the same in the Appendix.

**Lemma 4.10** (Uniform bound on DP noise variance). *Let $\xi_t$ denote the noise defined in* (4). *Assume that client dataset sizes satisfy $|\mathcal{D}_i| \geq m_{\min} > 0$ for all $i$. Then the variance parameter in* (5) *satisfies $\nu_t^2 \leq (C_g \sigma_g)^2/(n^2 m_{\min}^2) =: \nu^2$. Consequently, $\mathbb{E}[\|\xi_t\|_2^2 \mid \mathcal{F}_t] \leq d\nu^2$.*

## 4.2 Norm Control for the Clipped Aggregate and the Momentum Buffer

We begin with two elementary but fundamental estimates. The first is a deterministic norm bound coming directly from clipping. The second yields a uniform second-moment bound for the server momentum buffer. We provide proofs of both the lemmas in Appendix A.

**Lemma 4.11** (Norm bound for the aggregated clipped gradient). *For every $t \geq 0$,*

$$\|g_{\text{clip}}(\theta_t)\|_2 \leq C_g. \tag{8}$$

*Consequently,*

$$\mathbb{E}\big[\|G_t\|_2^2 \mid \mathcal{F}_t\big] = \|g_{\text{clip}}(\theta_t)\|_2^2 + d\nu_t^2 \leq C_g^2 + d\nu^2.$$

**Lemma 4.12** (Conditional second-moment recursion for the momentum buffer). *For every $t \geq 0$,*

$$\mathbb{E}\big[\|M_t\|_2^2 \mid \mathcal{F}_t\big] \leq \beta \|M_{t-1}\|_2^2 + (1 - \beta)C_g^2 + (1 - \beta)^2 d\nu^2. \tag{9}$$

*Define $u_t := \mathbb{E}\|M_t\|_2^2$. Then*

$$u_t \leq \beta u_{t-1} + (1 - \beta)C_g^2 + (1 - \beta)^2 d\nu^2, \qquad t \geq 0, \tag{10}$$

*and, since $M_{-1} = 0_d$,*

$$u_t \leq \bar{M}^2 := C_g^2 + (1 - \beta)d\nu^2, \qquad t \geq 0. \tag{11}$$

*Remark* 4.13 (Noise smoothing by the momentum buffer). Lemma 4.12 shows that the server-side buffer has uniformly bounded second moment despite the Gaussian privacy noise having unbounded support. This formalizes the stabilizing effect of the exponential moving average: the curvature proxy is built from a smoothed sequence of privatized aggregates rather than from a single noisy realization.

### 4.3 Properties of the Same-Step Preconditioner

We next establish deterministic structural bounds for the same-step preconditioner. These bounds hold pathwise for each realized $M_t$, and therefore remain valid even though $M_t$ depends on the current privacy noise. Proof of this lemma is in Appendix A.

**Lemma 4.14** (Sherman–Morrison representation and operator bounds). *For every $t \geq 0$,*

$$H_t = \frac{1}{\rho}I_d - \frac{M_t M_t^\top}{\rho(\rho + \|M_t\|_2^2)}. \tag{12}$$

*Consequently, for every $v \in \mathbb{R}^d$,*

$$\|H_t v\|_2 \leq \frac{1}{\rho}\|v\|_2, \tag{13}$$

$$\|H_t v\|_2^2 \leq \frac{1}{\rho^2}\|v\|_2^2, \tag{14}$$

$$\frac{1}{\rho + \|M_t\|_2^2}\|v\|_2^2 \leq v^\top H_t v \leq \frac{1}{\rho}\|v\|_2^2. \tag{15}$$

*In particular,*

$$\nabla F(\theta_t)^\top H_t \nabla F(\theta_t) \geq \frac{1}{\rho + \|M_t\|_2^2}\|\nabla F(\theta_t)\|_2^2. \tag{16}$$

**Corollary 4.15** (Non-vacuous lower bound on a bounded-momentum event). *Fix $T \geq 1$ and $\delta_M \in (0, 1)$. Let*

$$\bar{M}^2 = C_g^2 + (1 - \beta)d\nu^2$$

*be the uniform second-moment bound from Lemma 4.12, and define $M_\delta^2 := T\bar{M}^2/\delta_M$. Then*

$$\mathbb{P}\left(\max_{0 \leq t \leq T-1} \|M_t\|_2^2 \leq M_\delta^2\right) \geq 1 - \delta_M.$$

*On this event, for every $t = 0, \ldots, T - 1$,*

$$\nabla F(\theta_t)^\top H_t \nabla F(\theta_t) \geq \frac{1}{\rho + M_\delta^2}\|\nabla F(\theta_t)\|_2^2.$$

### 4.4 One-Step Descent for the Exact Same-Step Update

We now establish the central one-step descent inequality. Because the preconditioner $H_t$ depends on the current privatized aggregate $G_t$, the proof must control the coupling among $\nabla F(\theta_t)$, $H_t$, and $G_t$ within the same round. Before stating the one-step descent bound, we introduce two auxiliary Young parameters $\tau_1, \tau_2 > 0$. These constants are not algorithmic parameters. They are used only in the proof to control the cross-terms involving the clipping bias and the privacy noise. Specifically, Young's inequality is used to bound terms of the form

$$|\nabla F(\theta_t)^\top H_t \zeta_t| \quad \text{and} \quad |\nabla F(\theta_t)^\top H_t \xi_t|.$$

The parameter $\tau_1$ controls the bias cross-term, while $\tau_2$ controls the noise cross-term. Smaller choices of $\tau_1, \tau_2$ increase the descent coefficient but enlarge the additive bias and noise terms in the error floor.

**Lemma 4.16** (Conditional one-step descent). *Fix any $\tau_1, \tau_2 > 0$, and define*

$$c_\nabla := \frac{1}{\rho} - \frac{\tau_1 + \tau_2}{2}. \tag{17}$$

*Under Assumptions 4.2, 4.5, 4.6, and 4.8, for every $t \geq 0$,*

$$\mathbb{E}[F(\theta_{t+1}) \mid \mathcal{F}_t] \leq F(\theta_t) - \eta c_\nabla \|\nabla F(\theta_t)\|_2^2 + \frac{\eta G_{\max}^2}{\rho^2} \mathbb{E}\big[\|M_t\|_2^2 \mid \mathcal{F}_t\big] \tag{18}$$
$$+ \frac{\eta \zeta_{\max}^2}{2\tau_1 \rho^2} + \frac{\eta \, d \, \nu^2}{2\tau_2 \rho^2} + \frac{L\eta^2}{2\rho^2}(C_g^2 + d\nu^2).$$

The previous lemma still contains the conditional term $\mathbb{E}[\|M_t\|_2^2 \mid \mathcal{F}_t]$. The next lemma turns it into a deterministic constant.

**Lemma 4.17** (Uniform unconditional one-step descent). *Fix $\tau_1, \tau_2 > 0$ and define $c_\nabla$ as in (17). Under Assumptions 4.2, 4.5, 4.6, and 4.8, for every $t \geq 0$,*

$$\mathbb{E}[F(\theta_{t+1})] \leq \mathbb{E}[F(\theta_t)] - \eta c_\nabla \, \mathbb{E}\|\nabla F(\theta_t)\|_2^2 + \Gamma, \tag{19}$$

*where*

$$\Gamma := \frac{\eta G_{\max}^2}{\rho^2} \bar{M}^2 + \frac{\eta \zeta_{\max}^2}{2\tau_1 \rho^2} + \frac{\eta \, d \, \nu^2}{2\tau_2 \rho^2} + \frac{L\eta^2}{2\rho^2}(C_g^2 + d\nu^2), \tag{20}$$

*and $\bar{M}^2$ is given by (11).*

Proofs of both the above lemmas are in Appendix A.

*Remark* 4.18 (High-probability descent coefficient). The Young parameters $\tau_1, \tau_2$ in Lemma 4.16 control the bias and noise cross-terms. Combining the same Young bounds with Corollary 4.15 yields, on the bounded-momentum event, the pathwise coefficient

$$c_{\nabla, \delta} = \frac{1}{\rho + M_\delta^2} - \frac{\tau_1 + \tau_2}{2}.$$

This coefficient is non-vacuous whenever $\tau_1 + \tau_2 < 2/(\rho + M_\delta^2)$. We do not use $c_{\nabla, \delta}$ in the main expectation recursion; the main theorems use the conservative coefficient $c_\nabla = \rho^{-1} - (\tau_1 + \tau_2)/2$ and account for same-step coupling through the additive term in $\Gamma$.

## 4.5 Extension to General Privatized Aggregate Updates

The preceding analysis was stated for the one-step DP-FedGD-type update, where the server receives a clipped and privatized aggregate gradient. The same argument also applies to a broader class of privatized aggregate updates. This extension is useful for mini-batch implementations and multi-local-step DP-FedAvg-style variants.

Suppose the server receives a privatized aggregate update $G_t$ satisfying

$$G_t = \nabla F(\theta_t) + b_t + \xi_t,$$

where $b_t$ is an $\mathcal{F}_t$-measurable bias term and $\xi_t$ is the fresh noise generated at round $t$. Assume

$$\|b_t\|_2 \leq B, \qquad \mathbb{E}[\xi_t \mid \mathcal{F}_t] = 0, \qquad \mathbb{E}[\|\xi_t\|_2^2 \mid \mathcal{F}_t] \leq d\nu^2.$$

Here $b_t$ may contain clipping bias, stochastic mini-batch bias, and local-update drift. In the one-step DP-FedGD setting, $b_t$ reduces to the clipping bias and $B = \zeta_{\max}$. In a multi-local-step DP-FedAvg-style implementation, one may write abstractly $B = B_{\text{clip}} + B_{\text{drift}}$, where $B_{\text{drift}}$ captures the deviation between the aggregate local update and the global gradient at $\theta_t$.

**Proposition 4.19** (Descent under a biased privatized aggregate update)**.** *Suppose $F$ is $L$-smooth and $\|\nabla F(\theta_t)\|_2 \leq G_{\max}$ along the iterate sequence. Let $M_t = \beta M_{t-1} + (1-\beta)G_t$, $H_t = (\rho I_d + M_t M_t^\top)^{-1}$, and $\theta_{t+1} = \theta_t - \eta H_t G_t$. Fix $\tau_1, \tau_2 > 0$, and define $c_\nabla = \rho^{-1} - (\tau_1 + \tau_2)/2$. If $\mathbb{E}[\|G_t\|_2^2 \mid \mathcal{F}_t] \leq G_0^2 + d\nu^2$ for some finite constant $G_0$, then*

$$\mathbb{E}[F(\theta_{t+1}) \mid \mathcal{F}_t] \leq F(\theta_t) - \eta c_\nabla \|\nabla F(\theta_t)\|_2^2 + \frac{\eta G_{\max}^2}{\rho^2}\mathbb{E}[\|M_t\|_2^2 \mid \mathcal{F}_t]$$

$$+ \frac{\eta B^2}{2\tau_1 \rho^2} + \frac{\eta d\nu^2}{2\tau_2 \rho^2} + \frac{L\eta^2}{2\rho^2}(G_0^2 + d\nu^2).$$

*Consequently, if $\mathbb{E}\|M_t\|_2^2 \leq \overline{M}^2$ uniformly in $t$, then*

$$\mathbb{E}[F(\theta_{t+1})] \leq \mathbb{E}[F(\theta_t)] - \eta c_\nabla \mathbb{E}\|\nabla F(\theta_t)\|_2^2 + \Gamma_B,$$

*where*

$$\Gamma_B = \frac{\eta G_{\max}^2}{\rho^2}\overline{M}^2 + \frac{\eta B^2}{2\tau_1 \rho^2} + \frac{\eta d\nu^2}{2\tau_2 \rho^2} + \frac{L\eta^2}{2\rho^2}(G_0^2 + d\nu^2).$$

*Remark* 4.20 (Relation to mini-batch and multi-local-step methods). Proposition 4.19 shows that the SOFIM preconditioner is not tied to a full-gradient, one-local-step implementation. The proof only requires a privatized aggregate update with controlled bias and second moment. Mini-batching changes the noise and bias constants, while multi-local-step local training contributes an additional drift term to $B$. Thus the same descent mechanism remains valid, with the price of local training appearing explicitly through the bias radius $B$.

## 4.6 Convergence under Strong Convexity

We now combine the one-step descent bound with strong convexity. We provide the proof of this result in Appendix A.

**Theorem 4.21** (Linear convergence to a bias-and-noise neighborhood)**.** *Fix $\tau_1, \tau_2 > 0$ such that $c_\nabla$ as defined in (17) is positive. Suppose Assumptions 4.2, 4.3, 4.5, 4.6, and 4.8 hold. Define $r := 1 - 2\mu\eta c_\nabla$. If $r \in (0,1)$, equivalently $0 < \eta < 1/(2\mu c_\nabla)$, then for every $T \geq 0$,*

$$\mathbb{E}[F(\theta_T) - F(\theta^\star)] \leq r^T(F(\theta_0) - F(\theta^\star)) + \frac{1-r^T}{1-r}\Gamma \tag{21}$$

$$\leq r^T(F(\theta_0) - F(\theta^\star)) + \frac{\Gamma}{2\mu\eta c_\nabla}, \tag{22}$$

*where $\Gamma$ is defined in (20).*

*Remark* 4.22 (Interpretation of the error floor). The limiting neighborhood in (22) has three distinct sources. The term involving $d\nu^2$ is the differential-privacy noise floor. The term involving $\zeta_{\max}^2$ is the clipping-bias floor. The term involving $\bar{M}^2$ quantifies the additional cost induced by same-step coupling between the current preconditioner and the current privatized gradient. Since $\bar{M}^2 = C_g^2 + (1-\beta)d\nu^2$, the coupling penalty is itself controlled by the clipping threshold, the privacy noise scale, and the momentum parameter.

*Remark* 4.23 (Stepsize dependence of the error floor). Although the error floor is written as $\Gamma/(2\mu\eta c_\nabla)$, the quantity $\Gamma$ itself depends on $\eta$. In particular, the clipping-bias, privacy-noise, and same-step coupling terms in $\Gamma$ are of order $\eta$, whereas the smoothness term is of order $\eta^2$. Hence the limiting neighborhood consists of constant-order bias/noise/coupling terms plus an $O(\eta)$ smoothness contribution. Decreasing $\eta$ slows the contraction factor but does not by itself inflate the limiting neighborhood.

*Remark* 4.24 (Choice of the Young parameters $\tau_1, \tau_2$). The parameters $\tau_1$ and $\tau_2$ arise solely from Young's inequality in the control of the bias and noise cross-terms. Smaller values improve the descent coefficient $c_\nabla$, but enlarge the additive constants in $\Gamma$; larger values do the opposite. For the convergence theorem one only needs $c_\nabla > 0$, i.e., $\tau_1 + \tau_2 < 2/\rho$. In practice one may choose $\tau_1$ and $\tau_2$ as fixed small constants satisfying this condition.

## 4.7 Comparison with the DP-FedGD Error Floor

We next compare the DP-FedSOFIM floor with the corresponding first-order baseline under the same clipping-bias and DP-noise assumptions. Consider DP-FedGD,

$$\theta_{t+1}^{\mathrm{GD}} = \theta_t^{\mathrm{GD}} - \eta G_t, \qquad G_t = \nabla F(\theta_t^{\mathrm{GD}}) + \zeta_t + \xi_t,$$

where $\|\zeta_t\|_2 \le \zeta_{\max}$, $\mathbb{E}[\xi_t \mid \mathcal{F}_t] = 0$, and $\mathbb{E}[\|\xi_t\|_2^2 \mid \mathcal{F}_t] \le d\nu^2$.

**Lemma 4.25** (One-step descent for DP-FedGD). *Suppose $F$ is $L$-smooth and Assumptions 4.5 and 4.6 hold. Then, for any $\tau > 0$,*

$$\mathbb{E}[F(\theta_{t+1}^{\mathrm{GD}}) \mid \mathcal{F}_t] \le F(\theta_t^{\mathrm{GD}}) - \eta c_{\mathrm{GD}} \|\nabla F(\theta_t^{\mathrm{GD}})\|_2^2 + \Gamma_{\mathrm{GD}},$$

*where*

$$c_{\mathrm{GD}} := 1 - \frac{\tau}{2}, \qquad \Gamma_{\mathrm{GD}} := \frac{\eta \zeta_{\max}^2}{2\tau} + \frac{L\eta^2}{2}(C_g^2 + d\nu^2).$$

**Theorem 4.26** (DP-FedGD error floor under strong convexity). *Suppose $F$ is $\mu$-strongly convex and the assumptions of Lemma 4.25 hold. If $c_{\mathrm{GD}} > 0$ and $0 < \eta < 1/(2\mu c_{\mathrm{GD}})$, then*

$$\mathbb{E}[F(\theta_T^{\mathrm{GD}}) - F(\theta^\star)] \le (1 - 2\mu\eta c_{\mathrm{GD}})^T \{F(\theta_0) - F(\theta^\star)\} + \frac{\Gamma_{\mathrm{GD}}}{2\mu\eta c_{\mathrm{GD}}}.$$

The DP-FedGD floor is therefore $\Gamma_{\mathrm{GD}}/(2\mu\eta c_{\mathrm{GD}})$, whereas the DP-FedSOFIM floor in Theorem 4.21 is $\Gamma/(2\mu\eta c_\nabla)$. These bounds do not imply that DP-FedSOFIM has a uniformly smaller worst-case error floor than DP-FedGD. The SOFIM bound contains additional terms arising from the same-step coupling between $G_t$ and $H_t$, which is the price of using a data-dependent preconditioner constructed from the current privatized aggregate.

The theoretical advantage of DP-FedSOFIM appears when the preconditioner improves the effective descent geometry more than it increases the additive coupling term. The present analysis is conservative and does not fully exploit possible alignment between the momentum direction $M_t$, the gradient direction, and favorable curvature directions of the objective. Thus, our theorem proves stability and convergence of the server-side preconditioned method under DP noise, but it does not by itself prove a uniformly lower worst-case floor than DP-FedGD.

A sharper comparison is possible under an additional alignment or preconditioner-quality condition. For example, suppose that along the optimization path

$$\nabla F(\theta_t)^\top H_t \nabla F(\theta_t) \ge \alpha_H \|\nabla F(\theta_t)\|_2^2$$

for some $\alpha_H > 0$, and

$$\mathbb{E}\|H_t G_t\|_2^2 \le \sigma_H^2 (C_g^2 + d\nu^2).$$

Then the SOFIM floor has the schematic form

$$\frac{\eta B_H + L\eta^2 \sigma_H^2 (C_g^2 + d\nu^2)}{2\mu\eta\alpha_H},$$

where $B_H$ collects the bias and coupling terms. This floor can be smaller than the DP-FedGD floor when the effective-descent gain $\alpha_H$ and the variance reduction $\sigma_H^2$ dominate the additional coupling cost. This regime is plausible in ill-conditioned problems with a stable dominant direction in the privatized gradient trajectory, and is consistent with the empirical gains observed on PathMNIST and in the faster-convergence results.

## 4.8 Extension to the Polyak–Łojasiewicz Condition

The same descent argument extends verbatim from strong convexity to the Polyak–Łojasiewicz condition.

**Assumption 4.27** (Polyak–Łojasiewicz condition). *There exists $\mu_{\mathrm{PL}} > 0$ such that for all $\theta \in \mathbb{R}^d$,*

$$\frac{1}{2}\|\nabla F(\theta)\|_2^2 \ge \mu_{\mathrm{PL}}\big(F(\theta) - F(\theta^\star)\big). \tag{23}$$

*Remark* 4.28. The PL condition is strictly weaker than strong convexity. In particular, $\mu$-strong convexity implies (23) with $\mu_{\mathrm{PL}} = \mu$, but the converse need not hold.

**Theorem 4.29** (DP-FedSOFIM under the PL condition). *Fix $\tau_1, \tau_2 > 0$ such that $c_\nabla > 0$ in (17). Suppose Assumptions 4.2, 4.5, 4.6, 4.8, and 4.27 hold. Define $r_{\mathrm{PL}} := 1 - 2\mu_{\mathrm{PL}}\eta c_\nabla$. If $r_{\mathrm{PL}} \in (0,1)$, equivalently $0 < \eta < 1/(2\mu_{\mathrm{PL}}c_\nabla)$, then for every $T \geq 0$,*

$$\mathbb{E}[F(\theta_T) - F(\theta^\star)] \leq r_{\mathrm{PL}}^T \big(F(\theta_0) - F(\theta^\star)\big) + \frac{1 - r_{\mathrm{PL}}^T}{1 - r_{\mathrm{PL}}} \Gamma \tag{24}$$

$$\leq r_{\mathrm{PL}}^T \big(F(\theta_0) - F(\theta^\star)\big) + \frac{\Gamma}{2\mu_{\mathrm{PL}}\eta c_\nabla}, \tag{25}$$

*where $\Gamma$ is defined in (20).*

We provide the proof of the above result in Appendix A.

*Remark* 4.30 (Relationship between the strong-convexity and PL results). Since strong convexity implies the PL condition, Theorem 4.29 strictly generalizes Theorem 4.21. We state both versions explicitly because the strongly convex result aligns directly with the frozen linear-head regime used in the experiments, whereas the PL theorem extends the guarantee to a broader class of objectives.

## 4.9 Smooth Non-convex Stationarity

The preceding convergence result was stated under strong convexity, which is the most natural condition for the frozen-feature linear-head setting considered in our experiments. However, the one-step descent argument itself does not require convexity. It only requires smoothness, control of the bias in the privatized aggregate update, control of the DP noise, and a uniform bound on the gradient norm along the optimization path. Therefore, the same recursion also yields a standard stationarity guarantee for smooth non-convex objectives.

In the non-convex case, convergence to a global minimizer cannot be expected without additional geometric assumptions. We therefore measure optimization progress through the average squared gradient norm,

$$\frac{1}{T} \sum_{t=0}^{T-1} \mathbb{E}\|\nabla F(\theta_t)\|_2^2,$$

or equivalently through the squared gradient norm at a uniformly sampled iterate. This is the usual stationarity criterion in smooth non-convex optimization. The result below shows that DP-FedSOFIM reaches a stationary neighborhood whose size is determined by the same bias-and-noise term appearing in the one-step descent bound. Thus, the effects of clipping bias, privacy noise, and same-step preconditioner coupling remain explicit in the non-convex setting.

**Theorem 4.31** (Stationarity under smooth non-convex objectives). *Suppose Assumptions 4.2, 4.5, 4.6, and 4.8 hold. Let $c_\nabla = \rho^{-1} - (\tau_1 + \tau_2)/2 > 0$ for some $\tau_1, \tau_2 > 0$, and suppose $F_{\inf} := \inf_{\theta \in \mathbb{R}^d} F(\theta) > -\infty$. Then, for every $T \geq 1$,*

$$\frac{1}{T} \sum_{t=0}^{T-1} \mathbb{E}\|\nabla F(\theta_t)\|_2^2 \leq \frac{F(\theta_0) - F_{\inf}}{\eta c_\nabla T} + \frac{\Gamma}{\eta c_\nabla},$$

*where $\Gamma$ is the bias-and-noise term defined in Lemma 4.17. Equivalently, if $R$ is sampled uniformly from $\{0, \ldots, T-1\}$, independently of the algorithmic randomness, then*

$$\mathbb{E}\|\nabla F(\theta_R)\|_2^2 \leq \frac{F(\theta_0) - F_{\inf}}{\eta c_\nabla T} + \frac{\Gamma}{\eta c_\nabla}.$$

**Corollary 4.32** (Stationarity for biased privatized aggregate updates). *Under the assumptions of Proposition 4.19, suppose $F_{\inf} > -\infty$ and $c_\nabla > 0$. Then*

$$\frac{1}{T} \sum_{t=0}^{T-1} \mathbb{E}\|\nabla F(\theta_t)\|_2^2 \leq \frac{F(\theta_0) - F_{\inf}}{\eta c_\nabla T} + \frac{\Gamma_B}{\eta c_\nabla}.$$

*Remark* 4.33 (Interpretation). Theorem 4.31 does not require strong convexity or the PL condition. The first term on the right-hand side decays at the usual $O(1/T)$ rate for fixed stepsize, while the second term is the limiting stationarity floor induced by clipping, privacy noise, and the coupling between the current privatized aggregate $G_t$ and the same-round preconditioner $H_t$. When the clipping bias and privacy noise are small and the preconditioner is stable, this neighborhood is correspondingly small. Fully general global convergence for deep non-convex objectives is not claimed.

### 4.10 Privacy Analysis

We now analyze the differential privacy guarantees of DP-FedSOFIM. The key observation is that DP-FedSOFIM introduces no additional client-side data access beyond the privatized gradient release mechanism already used in standard differentially private federated learning. All additional computations occur at the server and depend only on already privatized quantities. Accordingly, the privacy guarantees of DP-FedSOFIM follow from two standard principles of differential privacy: post-processing invariance and sequential composition.

**Round-wise privatized release:** Fix a communication round $t$. Let $\mathcal{M}_t(\mathcal{D})$ denote the randomized mechanism that maps the federated dataset $\mathcal{D} = \mathcal{D}_1, \ldots, \mathcal{D}_n$) to the set of privatized client updates released to the server at round $t$. In the present algorithm this mechanism produces the vector $\mathcal{M}_t(\mathcal{D}) = (g_{1,t}, \ldots, g_{n,t})$, where each $g_{i,t}$ is the privatized update defined in (2). The exact differential privacy parameters of this mechanism depend on the clipping rule, noise scale, dataset normalization, participation pattern, and the privacy accountant used across rounds. Rather than fixing a specific accountant in the theoretical analysis, we assume that the round-wise release mechanism satisfies a valid differential privacy guarantee.

**Assumption 4.34** (Per-round privacy of the released gradients). For each round $t$, the release mechanism $\mathcal{M}_t$ satisfies $(\varepsilon_t, \delta_t)$-differential privacy with respect to the federated record-level adjacency defined in Definition 3.2.

**Privacy of the aggregated gradient:** The server aggregates the client updates as $G_t = n^{-1} \sum_{i=1}^n g_{i,t}$.

**Lemma 4.35** (Privacy of the aggregated gradient). *Under Assumption 4.34, the aggregated gradient $G_t$ is $(\varepsilon_t, \delta_t)$-differentially private.*

**Server-side privacy preservation:** DP-FedSOFIM differs from the underlying first-order private federated optimization method only through additional server-side computations based on the privatized aggregated gradients.

Specifically, the server performs the updates

$$M_t = \beta M_{t-1} + (1-\beta)G_t, \qquad \widehat{\mathcal{I}}_t = \rho I_d + M_t M_t^\top, \qquad H_t = \widehat{\mathcal{I}}_t^{-1}, \qquad \theta_{t+1} = \theta_t - \eta_t H_t G_t.$$

These quantities depend only on $G_t$ and the previous server state.

**Lemma 4.36** (Server-side post-processing). *Fix a communication round $t$. If the aggregated gradient $G_t$ is $(\varepsilon_t, \delta_t)$-differentially private conditional on the previous server state, then the full server output of round $t$, namely $(M_t, H_t, \theta_{t+1})$, is also $(\varepsilon_t, \delta_t)$-differentially private conditional on the same previous server state.*

**Adaptive composition across rounds:** Because the model parameters released at round $t$ influence the client computations performed at round $t+1$, the overall training procedure is adaptive across rounds. Differential privacy nevertheless remains valid under adaptive sequential composition.

**Theorem 4.37** (End-to-end privacy of DP-FedSOFIM). *Suppose Assumption 4.34 holds for each round $t = 0, \ldots, T-1$. Then the full $T$-round DP-FedSOFIM algorithm satisfies*

$$\left( \sum_{t=0}^{T-1} \varepsilon_t, \sum_{t=0}^{T-1} \delta_t \right)\text{-DP}.$$

*In particular, if the same per-round privacy parameters $(\varepsilon_0, \delta_0)$ are used at every round, then the overall procedure satisfies $(T\varepsilon_0, T\delta_0)$-DP.*

*Proof.* By Lemma 4.35, the aggregated gradient $G_t$ is $(\varepsilon_t, \delta_t)$-differentially private. Lemma 4.36 then implies that the complete server output of round $t$ inherits the same privacy guarantee.

Applying adaptive sequential composition across $t = 0, \ldots, T-1$ yields the stated bound.

$\square$

Note that the above bound is only a simple-composition baseline. The actual privacy guarantee of DP-FedSOFIM is accountant-agnostic: whatever final $\varepsilon, \delta)$ guarantee is certified for the underlying privatized gradient-release sequence is inherited unchanged by DP-FedSOFIM, because all SOFIM computations are post-processings of already privatized aggregate gradients.

**Theorem 4.38** (Privacy under an arbitrary valid accountant). *Fix the clipping rule, noise multiplier, participation mechanism, number of rounds, adjacency relation, and privacy accountant used for the underlying privatized gradient-release mechanism. Suppose this accountant certifies that the sequence of privatized releases $(G_0, \ldots, G_{T-1})$ satisfies $(\varepsilon, \delta)$-differential privacy. Then the full DP-FedSOFIM transcript $(\theta_1, M_0, H_0, \ldots, \theta_T, M_{T-1}, H_{T-1})$ also satisfies $(\varepsilon, \delta)$-differential privacy.*

*Proof.* The DP-FedSOFIM transcript is obtained by applying deterministic server-side maps to the privatized release sequence $(G_0, \ldots, G_{T-1})$. These maps include the momentum recursion, the rank-one regularized preconditioner, and the parameter update. Therefore, by post-processing invariance of differential privacy, the transcript has the same privacy guarantee certified for the underlying release sequence. $\square$

**Proposition 4.39** (Privacy preservation for general privatized local updates). *Let $\widetilde{\Delta}_t$ be any aggregate client update released to the server at round $t$. Suppose $\widetilde{\Delta}_t$ is $(\varepsilon_t, \delta_t)$-differentially private conditional on the previous server state. Define*

$$M_t = \beta M_{t-1} + (1-\beta)\widetilde{\Delta}_t, \qquad H_t = (\rho I_d + M_t M_t^\top)^{-1}, \qquad \theta_{t+1} = \theta_t - \eta_t H_t \widetilde{\Delta}_t.$$

*Then $(M_t, H_t, \theta_{t+1})$ is also $(\varepsilon_t, \delta_t)$-differentially private conditional on the previous server state.*

*Proof.* Condition on the previous server state $(\theta_t, M_{t-1})$. Then $M_t$, $H_t$, and $\theta_{t+1}$ are deterministic functions of the privatized aggregate update $\widetilde{\Delta}_t$. Therefore, by the post-processing property of differential privacy, the tuple $(M_t, H_t, \theta_{t+1})$ has the same round-wise privacy guarantee as $\widetilde{\Delta}_t$. $\square$

**Instantiating the privacy parameters:** Appendix F derives the concrete privacy parameters for the Gaussian release used in (2) under replace-one record adjacency. That derivation computes the sensitivity of the clipped client update and applies the chosen privacy accountant to obtain the corresponding $(\varepsilon_t, \delta_t)$ guarantees. Once these parameters are instantiated for the client-side release mechanism, Theorem 4.37 implies that the SOFIM momentum and preconditioning steps incur no additional privacy loss.

**Proposition 4.40** (Privacy-cost equivalence with DP-FedGD). *Fix a clipping rule, clipping threshold $C_g$, noise multiplier $\sigma_g$, client participation mechanism, number of communication rounds $T$, and privacy accountant. Suppose DP-FedGD with these choices satisfies $(\varepsilon, \delta)$ differential privacy. Then DP-FedSOFIM with the same choices also satisfies $(\varepsilon, \delta)$ differential privacy.*

*Proof.* Under the stated choices, DP-FedGD and DP-FedSOFIM use the same client-side private release mechanism. In each communication round, the data-dependent quantity released to the server is the same clipped and privatized aggregate gradient $G_t$.

DP-FedSOFIM differs from DP-FedGD only through the server-side computations

$$M_t = \beta M_{t-1} + (1-\beta)G_t, \qquad \widehat{I}_t = \rho I_d + M_t M_t^\top, \qquad H_t = \widehat{I}_t^{-1},$$

and

$$\theta_{t+1} = \theta_t - \eta_t H_t G_t.$$

Conditional on the previous server state, these quantities are deterministic functions of the already privatized aggregate $G_t$. Therefore, by the post-processing property of differential privacy, they introduce no additional privacy loss in round $t$.

Since the sequence of private releases across the $T$ rounds is the same as in DP-FedGD, the adaptive composition analysis under the chosen privacy accountant is also identical. Hence DP-FedSOFIM satisfies the same final $(\varepsilon, \delta)$ guarantee as DP-FedGD. $\qquad\square$

Table 1 compares the information released by representative DP-FL methods. Standard DP-FL algorithms privatize clipped gradients or client updates and track the cumulative privacy loss through composition/accounting methods (Dwork et al., 2014; Abadi et al., 2016; McMahan et al., 2017b; Mironov, 2017). Methods that release additional client-side quantities, such as control variates or curvature/covariance statistics, require privacy accounting appropriate to those releases (Karimireddy et al., 2020). DP-FedSOFIM differs in that all curvature operations are performed at the server after the aggregate gradient has already been privatized. Therefore, the SOFIM update is a deterministic post-processing of a DP quantity and incurs no additional privacy cost (Dwork et al., 2014).

Table 1: Privacy and computational comparison. DP-FedSOFIM has the same privacy cost as DP-FedGD because all SOFIM operations are post-processings of already privatized aggregate gradients.

| Method | Additional information beyond noisy aggregate | Privacy cost | Client memory |
|---|---|---|---|
| DP-FedGD | None | $(\varepsilon, \delta)$ | $O(d)$ |
| DP-FedAvg | None, if only privatized client updates are released | Accountant-dependent | $O(d)$ |
| DP-FedAdam | Server-side first- and second-moment states computed from privatized aggregates | Same as the underlying privatized update release | $O(d)$ |
| DP-FedYogi | Server-side adaptive moment states computed from privatized aggregates | Same as the underlying privatized update release | $O(d)$ |
| DP-FTRL | Accumulated privatized gradients or tree-aggregated noisy sums, depending on implementation | Accountant-dependent; often benefits from privacy accounting for cumulative releases | $O(d)$ |
| DP-AdaFedProx | Adaptive/proximal server-side state; may require additional state depending on implementation | Accountant-dependent | $O(d)$ |
| DP-SCAFFOLD | Control-variate state, depending on implementation | Accountant-dependent | $O(d)$ |
| DP-FedFC / DP-FedNew-type | Client-side curvature or covariance statistic | Requires accounting if privately released | $O(d^2)$ |
| DP-FedSOFIM | None; SOFIM is server-side post-processing of privatized aggregate gradients | Same as DP-FedGD under the same clipping, noise, participation, and accountant | $O(d)$ |

*Remark* 4.41 (Meaning of "no additional privacy cost"). Proposition 4.40 says that DP-FedSOFIM and DP-FedGD have identical privacy cost when the private release mechanism and privacy accountant are fixed. The advantage of DP-FedSOFIM is that curvature-aware preconditioning is added after privatization, and hence does not consume additional privacy budget.

### 4.11 Comparison with Existing DP-FL Theory

The convergence results above should be interpreted as structural guarantees for privacy-compatible server-side curvature adaptation. We do not claim that DP-FedSOFIM improves the minimax convergence rate of DP-FL in all regimes. Rather, the theoretical advantage is that curvature information is introduced without changing the private release mechanism.

Compared with DP-FedGD, DP-FedSOFIM uses the same clipped and privatized aggregate gradient, and therefore has the same privacy cost under the same accountant. The difference lies only in the server-side update: DP-FedGD applies an isotropic first-order step, while DP-FedSOFIM applies the preconditioned step

$$\theta_{t+1} = \theta_t - \eta_t H_t G_t, \qquad H_t = (\rho I_d + M_t M_t^\top)^{-1}.$$

Thus the privacy noise source is the same as in DP-FedGD, but the update is rescaled along the curvature direction estimated from the privatized aggregate trajectory.

Compared with DP-FedFC and DP-FedNew-type methods, DP-FedSOFIM avoids releasing client-side covariance, Hessian, or Fisher-type statistics. This distinction is important because such second-order statistics generally require $O(d^2)$ client-side memory and communication, and must be accounted for if they are released privately. DP-FedSOFIM instead constructs its rank-one Fisher proxy entirely at the server from already privatized aggregate gradients.

Compared with DP-SCAFFOLD, DP-FedSOFIM does not rely on maintaining client-level control variates. In high privacy-noise regimes, control-variate estimates may themselves become noisy or require additional accounting depending on the implementation. DP-FedSOFIM uses only the server-side momentum buffer $M_t = \beta M_{t-1} + (1 - \beta)G_t$, which is a deterministic function of privatized aggregates. Therefore, the main theoretical benefit of DP-FedSOFIM is the combination of same privacy cost as DP-FedGD, $O(d)$ client-side complexity, and server-side curvature-aware preconditioning. The empirical section then evaluates whether this structural advantage translates into improved optimization and predictive performance under fixed privacy budgets.

## 5 Experiments

### 5.1 Experimental Setup

**Datasets and Models:** We evaluate DP-FedSOFIM on two datasets. **CIFAR-10** consists of 50,000 training images and 10,000 test images across 10 classes. **PathMNIST** is a medical imaging dataset from the MedMNIST benchmark (Yang et al., 2023), consisting of colorectal cancer histology patches across 9 tissue classes. Following the transfer learning protocol for differentially private learning (Tramèr & Boneh, 2021; Krouka et al., 2025), we employ two frozen feature extractors pre-trained on CIFAR-100: a **ResNet-20** (He et al., 2016) backbone and a **VGG-16** (Simonyan & Zisserman, 2015) backbone. In both cases only the final linear classification head is optimized, yielding a low-dimensional parameter space well-suited to second-order optimization while preserving the essential challenges of private federated learning.

**Federated Setting:** We simulate a federated environment with $n = 20$ clients with full client participation in every round. To assess robustness to data heterogeneity, we evaluate each method under two data partitioning schemes: **IID**, where training data is partitioned uniformly at random across clients, and **non-IID**, where data is partitioned using a Dirichlet process with concentration parameter $\alpha = 0.5$, inducing moderate label heterogeneity across clients. For the VGG-16 backbone, we report results under the non-IID partition only, as the primary goal of the VGG-16 experiments is to assess cross-architecture robustness of the non-IID findings rather than to replicate the full evaluation protocol.

To ensure reproducibility, Tables 13 and 14 (Appendix I) report the complete per-client, per-class sample counts for both datasets under a representative seed. On PathMNIST (89,996 training samples, 9 classes), the Dirichlet partition yields client sizes ranging from 1,045 to 9,986 (mean = 4,500, std = 2,526). The within-client class distribution is highly concentrated: the mean KL divergence from the uniform class distribution is 0.581 (maximum 1.115, where 0 indicates perfectly IID), and two clients (clients 8 and 14) are

missing two classes entirely. Dominant-class fractions range up to 68% (client 8: class 4 accounts for 3,046 of 4,477 samples), confirming substantial but not extreme label heterogeneity. On CIFAR-10 (50,000 training samples, 10 classes), the same Dirichlet($\alpha = 0.5$) process yields client sizes ranging from 891 to 3,498 (mean $= 2,500$, std $= 779.6$), with a mean KL divergence from the uniform class distribution of 0.666 (max 1.039) and a mean of 0.3 absent classes per client (maximum 2, affecting clients 4 and 18). Per-client, per-class counts are reported in Table 14.

The same set of hyperparameters is used for both partitioning schemes. Each client computes gradients over its entire local dataset (full-batch), following the full-gradient protocol of Krouka et al. (2025). Training proceeds for $T = 70$ federated rounds with a single local iteration per round (except DP-FedAvg and DP-SCAFFOLD, where local iterations are tuned as a hyperparameter), representing a communication-constrained scenario typical of cross-device federated learning.

**Privacy Parameters:** We evaluate across four privacy regimes: $\varepsilon \in \{0.5, 1, 5, 10\}$ with $\delta = 10^{-5}$, spanning from stringent ($\varepsilon = 0.5$) to moderate ($\varepsilon = 10$) privacy guarantees. The noise multiplier $\sigma_g$ for each $\varepsilon$ is calibrated using the closed-form hockey-stick divergence formula for Gaussian mechanisms (Balle & Wang, 2018), with tight $T$-fold composition bounds following Zhu et al. (2022), to achieve the target $(\varepsilon, \delta)$-DP budget after $T = 70$ rounds. Under full client participation the per-round Gaussian noise scales as $\sigma_g = \Theta(\varepsilon^{-1}\sqrt{T})$, making tight privacy budgets ($\varepsilon \leq 1$) significantly more challenging for convergence.

**Baselines:** We compare DP-FedSOFIM against eight baselines. **DP-FedGD** (McMahan et al., 2017b; Krouka et al., 2025), the standard first-order baseline applying per-example gradient clipping followed by Gaussian noise addition with a single local step; **DP-FedAvg** (McMahan et al., 2017b; Abadi et al., 2016), which extends DP-FedGD to multiple local SGD steps before aggregation; **DP-FedFC** (Krouka et al., 2025), a second-order method that computes local feature covariance matrices as an approximation to the Fisher information matrix, incurring $O(d^2)$ client-side memory; **DP-SCAFFOLD** (Karimireddy et al., 2020), a variance-reduction method using control variates to correct for client drift; **DP-FedAdam** (Reddi et al., 2020), which applies an Adam-style adaptive server optimizer atop per-client DP gradient aggregation; **DP-FedYogi** (Reddi et al., 2020), which replaces the Adam server step with a Yogi update, improving stability under the large gradient variance induced by DP noise; **DP-FTRL** (Kairouz et al., 2021), which uses a tree-aggregation protocol to accumulate gradients across rounds, enabling tighter privacy accounting under continual observation; and **DP-AdaFedProx** (Sahoo et al., 2024), a proximal federated method that adapts per-coordinate learning rates to the heterogeneous client landscape. All baselines are calibrated to the same $(\varepsilon, \delta)$ privacy budget as DP-FedSOFIM using identical privacy accounting.

**Hyperparameters:** DP-FedSOFIM tunes regularization parameter $\rho$, EMA momentum $\beta$, and learning rate $\eta$ per privacy regime, with bias correction enabled for $\varepsilon \leq 1$ where noise-induced curvature bias is most pronounced.

Hyperparameters are determined via a two-stage grid search. A coarse search over a wide logarithmically-spaced grid identifies the promising region; a subsequent fine search with finer resolution around the best coarse result yields the final configuration. Both stages use a fixed seed, $T = 50$ rounds, and are conducted on a held-out validation split to avoid overfitting to test performance. The configuration yielding the best final validation accuracy is selected independently for each method, dataset, and privacy regime, and the same hyperparameters are used for both IID and non-IID partitions. Search grids are detailed in Table 2.

For the tightest privacy regime ($\varepsilon = 0.5$), we additionally employ a *warmup* strategy for DP-FedSOFIM: the Sherman–Morrison preconditioning step is disabled for the first 20 rounds, during which the server uses EMA-only updates. This delayed activation prevents the curvature estimate from being corrupted by the large Gaussian noise early in training, after which the signal-to-noise ratio is sufficient for reliable second-order information to accumulate. The warmup duration (20 rounds) is treated as a fixed implementation choice rather than a tuned hyperparameter, and is applied consistently across both datasets and both backbones.

## 5.2 Results

Tables 3–6 and Figures 1–4 present test accuracy trajectories across all privacy regimes for CIFAR-10 and PathMNIST under both ResNet-20 and VGG-16 backbones. Statistical significance of pairwise differences at

Table 2: Hyperparameter search grids for all methods, determined via a two-stage coarse-to-fine grid search. Each stage is run for $T = 50$ rounds with a fixed seed; the configuration achieving the best final validation accuracy is carried forward. Parameters fixed across all methods: $C_g = 10$, $T = 70$ rounds, $\delta = 10^{-5}$.

| Method | Hyperparameter | Search Grid (Coarse $\rightarrow$ Fine) |
|---|---|---|
| DP-FedGD | Learning rate $\eta$ | $\{10^{-4}, 10^{-3}, 0.01, 0.1, 1, 5, 10\} \rightarrow \{0.03, 0.05, 0.08, 0.1, 0.3\}$ |
| DP-FedAvg | Server learning rate $\eta_s$ 
 Local iterations | $\{10^{-3}, 0.01, 0.1, 0.5, 1\} \rightarrow \{0.3, 0.5, 0.6, 0.8, 1.0\}$ 
 $\{1, 3, 5, 10, 20\} \rightarrow \{1, 2, 5, 7, 10\}$ |
| DP-FedFC | Learning rate $\eta$ 
 Curvature scale $\gamma$ | $\{10^{-4}, 10^{-3}, 0.01, 0.1, 1, 5\} \rightarrow \{0.1, 0.15, 0.3, 0.4, 0.5, 2\}$ 
 $\{0.01, 0.1, 1, 10, 100\} \rightarrow \{0.5, 1, 2, 5, 10\}$ |
| DP-SCAFFOLD | Server learning rate $\eta_s$ 
 Client learning rate $\eta_c$ 
 Local steps | $\{0.01, 0.1, 0.5, 1, 5\} \rightarrow \{0.3, 0.5, 0.7, 1, 2\}$ 
 $\{0.001, 0.01, 0.1, 0.5\} \rightarrow \{0.05, 0.1, 0.2, 0.3\}$ 
 $\{1, 5, 10\} \rightarrow \{1, 3, 5, 7, 10\}$ |
| DP-FedAdam | Learning rate $\eta$ 
 $(\beta_1, \beta_2)$ 
 Server $\epsilon$ | $\{10^{-4}, 10^{-3}, 0.01, 0.1, 1\} \rightarrow \{0.01, 0.02, 0.05, 0.1\}$ 
 $(0, 0.8, 0.9) \times (0.8, 0.9, 0.999) \rightarrow (0.8, 0.9, 0.95) \times (0.8, 0.999, 0.9999)$ 
 $\{10^{-5}, 10^{-3}, 0.01, 0.1\} \rightarrow \{10^{-5}, 0.01, 0.05, 0.1\}$ |
| DP-FedYogi | Learning rate $\eta$ 
 $(\beta_1, \beta_2)$ 
 Server $\epsilon$ | $\{10^{-4}, 10^{-3}, 0.01, 0.1, 1\} \rightarrow \{0.01, 0.02, 0.05, 0.1, 0.2\}$ 
 $(0, 0.5, 0.9) \times (0.5, 0.9, 0.999) \rightarrow (0.5, 0.9, 0.95) \times (0.5, 0.9, 0.999)$ 
 $\{10^{-5}, 10^{-3}, 0.01, 0.1\} \rightarrow \{10^{-5}, 0.01, 0.05, 0.1\}$ |
| DP-FTRL | Learning rate $\eta$ | $\{0.5, 1, 3, 5, 7, 10\} \rightarrow \{3, 3.5, 5, 7, 8, 9\}$ |
| DP-AdaFedProx | Server learning rate $\eta_s$ 
 Client learning rate $\eta_c$ 
 Initial proximal term $\mu_0$ 
 Local steps | $\{0.1, 0.3, 0.5, 1, 3\} \rightarrow \{0.5, 0.6, 0.8, 1.0, 1.5\}$ 
 $\{0.001, 0.01, 0.05, 0.1, 0.5\} \rightarrow \{0.05, 0.1, 0.2, 0.3\}$ 
 $\{0.001, 0.01, 0.1, 0.5, 1\} \rightarrow \{0.01, 0.05, 0.1, 0.2\}$ 
 $\{1, 3, 5, 7\} \rightarrow \{1, 3, 5\}$ |
| DP-FedSOFIM | Learning rate $\eta$ 
 Regularization $\rho$ 
 EMA momentum $\beta$ | $\{10^{-3}, 0.01, 0.1, 1, 5\} \rightarrow \{0.1, 0.2, 0.5, 1, 3, 4\}$ 
 $\{0.01, 0.1, 1, 5, 10\} \rightarrow \{0.5, 1, 5, 10, 20\}$ 
 $\{0.8, 0.9, 0.99\} \rightarrow \{0.8, 0.85, 0.9, 0.95\}$ |

round 70 is assessed via McNemar's test (Appendix J). We organize our analysis around five key phenomena observed in the results.

Table 3: Test accuracy (%) on CIFAR-10 with ResNet-20 backbone across federated rounds for different privacy budgets. Results shown at 10-round intervals (mean ± std over 3 seeds). Best result per privacy regime is in **bold**.

| Method | Federated Round | | | | | | |
|---|---|---|---|---|---|---|---|
| | 10 | 20 | 30 | 40 | 50 | 60 | 70 |
| $\varepsilon = 0.5$ | | | | | | | |
| DP-FedGD | 33.98±2.76 | 45.55±1.21 | 50.15±1.10 | 53.88±1.13 | 55.60±1.04 | 57.01±0.23 | 58.37±0.68 |
| DP-FedAvg | 33.98±2.76 | 45.55±1.21 | 50.15±1.10 | 53.88±1.13 | 55.60±1.04 | 57.01±0.23 | 58.37±0.68 |
| DP-FedFC | 23.37±5.03 | 34.86±3.10 | 43.64±2.18 | 48.90±0.57 | 52.33±1.21 | 53.92±0.72 | 55.33±1.28 |
| DP-SCAFFOLD | 30.45±2.81 | 38.29±1.24 | 43.49±1.90 | 45.52±0.42 | 47.76±1.03 | 48.31±1.18 | 49.78±1.53 |
| DP-FedAdam | 34.22±3.59 | 44.62±1.94 | 50.75±1.34 | 54.43±0.74 | 56.76±0.24 | 57.74±0.18 | **59.23±0.53** |
| DP-FedYogi | 29.77±3.43 | 41.35±3.01 | 47.80±1.20 | 52.69±0.85 | 55.38±0.40 | 56.90±0.27 | 58.60±0.24 |
| DP-FTRL | 31.82±1.81 | **53.91±0.33** | **54.43±0.62** | **55.35±0.39** | 55.34±0.49 | 55.63±0.19 | 55.73±0.10 |
| DP-AdaFedProx | 33.98±2.76 | 45.55±1.21 | 50.15±1.10 | 53.88±1.13 | 55.60±1.04 | 57.01±0.23 | 58.37±0.68 |
| DP-FedSOFIM | **39.56±2.26** | 47.47±1.15 | 52.84±0.83 | 55.41±0.84 | **57.20±1.41** | **58.62±1.19** | 58.86±1.46 |
| $\varepsilon = 1$ | | | | | | | |
| DP-FedGD | 39.53±2.50 | 50.26±0.69 | 55.47±0.56 | 58.55±0.68 | 60.68±0.28 | 61.82±0.42 | 62.75±0.26 |
| DP-FedAvg | 39.53±2.51 | 50.25±0.69 | 55.46±0.56 | 58.54±0.68 | 60.68±0.27 | 61.84±0.41 | 62.74±0.26 |
| DP-FedFC | 51.20±1.10 | 57.04±0.33 | 59.44±0.48 | 60.47±0.34 | 61.60±0.96 | 61.88±0.74 | 61.57±0.72 |
| DP-SCAFFOLD | 41.99±1.27 | 49.58±0.92 | 53.77±0.16 | 54.20±0.97 | 54.96±1.54 | 56.03±1.00 | 56.03±0.72 |
| DP-FedAdam | 49.15±2.01 | 58.43±0.29 | 60.55±0.71 | 61.88±0.48 | 62.45±0.63 | 62.54±0.57 | 62.49±0.74 |
| DP-FedYogi | 40.18±4.62 | 53.93±0.66 | 57.00±0.81 | 58.00±0.72 | 58.15±1.06 | 58.37±1.44 | 57.45±1.38 |
| DP-FTRL | 29.11±7.18 | 53.44±0.48 | 58.23±0.27 | 58.82±0.29 | 58.93±0.16 | 59.19±0.37 | 59.23±0.24 |
| DP-AdaFedProx | 49.12±2.94 | 55.18±0.56 | 56.91±0.78 | 58.28±0.84 | 58.98±0.84 | 59.45±0.18 | 59.96±0.32 |
| DP-FedSOFIM | **51.44±0.84** | **58.59±0.23** | **60.89±0.17** | **62.37±0.74** | **62.83±0.85** | **63.09±0.73** | **63.43±0.62** |
| $\varepsilon = 5$ | | | | | | | |
| DP-FedGD | 40.74±2.17 | 51.80±0.38 | 56.85±0.09 | 59.98±0.69 | 61.85±0.45 | 63.21±0.62 | 64.10±0.58 |
| DP-FedAvg | 40.75±2.16 | 51.79±0.38 | 56.85±0.09 | 59.97±0.68 | 61.85±0.45 | 63.20±0.61 | 64.09±0.58 |
| DP-FedFC | 46.56±3.37 | 49.58±0.98 | 55.42±1.17 | 58.43±0.77 | 60.83±0.52 | 62.40±0.47 | 63.51±0.71 |
| DP-SCAFFOLD | 55.66±0.36 | 62.19±0.74 | 64.29±0.62 | 65.18±0.16 | 65.67±0.22 | 65.96±0.24 | 66.42±0.10 |
| DP-FedAdam | 43.92±1.63 | 59.76±1.34 | 63.64±1.06 | 66.09±0.52 | 67.11±0.06 | 67.15±0.20 | 67.28±0.20 |
| DP-FedYogi | 55.37±0.43 | 63.72±0.40 | 66.06±0.33 | 67.06±0.13 | **67.35±0.33** | **67.69±0.23** | **67.81±0.20** |
| DP-FTRL | 28.77±7.55 | 42.13±2.38 | 55.67±0.49 | 61.19±0.29 | 61.52±0.40 | 61.72±0.46 | 61.78±0.41 |
| DP-AdaFedProx | 57.53±0.81 | 63.22±0.49 | 65.01±0.09 | 65.89±0.17 | 66.45±0.07 | 66.85±0.19 | 67.02±0.07 |
| DP-FedSOFIM | **61.05±1.14** | **65.64±0.31** | **66.89±0.38** | **67.37±0.29** | 67.25±0.56 | 67.20±0.55 | 67.52±0.33 |
| $\varepsilon = 10$ | | | | | | | |
| DP-FedGD | 40.79±2.13 | 51.86±0.42 | 56.94±0.10 | 59.97±0.59 | 61.94±0.49 | 63.17±0.56 | 64.10±0.52 |
| DP-FedAvg | 40.79±2.14 | 51.84±0.41 | 56.94±0.10 | 59.97±0.59 | 61.92±0.49 | 63.18±0.57 | 64.10±0.52 |
| DP-FedFC | 40.74±2.17 | 51.67±0.45 | 57.00±0.22 | 60.03±0.56 | 61.88±0.56 | 63.14±0.56 | 64.10±0.51 |
| DP-SCAFFOLD | 61.36±0.29 | 65.24±0.46 | 66.29±0.33 | 67.00±0.13 | 67.22±0.16 | 67.60±0.19 | 67.85±0.17 |
| DP-FedAdam | 43.51±1.73 | 57.62±1.46 | 63.46±0.45 | 65.02±0.41 | 66.91±0.43 | 67.51±0.29 | 67.77±0.27 |
| DP-FedYogi | 44.04±1.32 | 58.01±1.48 | 62.90±0.34 | 64.78±0.45 | 66.60±0.33 | 67.34±0.15 | 67.72±0.18 |
| DP-FTRL | 27.12±6.46 | 45.99±3.70 | 53.02±2.27 | 61.17±1.11 | 63.17±0.54 | 63.21±0.44 | 63.21±0.43 |
| DP-AdaFedProx | 58.19±0.54 | 63.73±0.34 | 65.51±0.11 | 66.32±0.22 | 66.86±0.12 | 67.36±0.26 | 67.42±0.07 |
| DP-FedSOFIM | **61.52±1.07** | **66.06±0.13** | **67.52±0.27** | **68.04±0.19** | **68.30±0.24** | **68.40±0.22** | **68.56±0.05** |

Table 4: Test accuracy (%) on PathMNIST with ResNet-20 backbone across federated rounds for different privacy budgets. Results shown at 10-round intervals (mean ± std over 3 seeds). Best result per privacy regime is in **bold**.

| Method | Federated Round | | | | | | |
|---|---|---|---|---|---|---|---|
| | 10 | 20 | 30 | 40 | 50 | 60 | 70 |
| $\varepsilon = 0.5$ | | | | | | | |
| DP-FedGD | 50.79±0.50 | 57.48±1.96 | 59.94±1.93 | 62.35±0.68 | 63.70±0.36 | 63.79±0.69 | 64.50±0.73 |
| DP-FedAvg | 47.85±2.96 | 55.46±3.08 | 58.56±1.82 | 61.15±1.17 | 62.56±0.90 | 63.81±0.78 | 64.56±0.75 |
| DP-FedFC | 55.15±1.66 | 58.89±1.24 | 62.67±0.54 | 64.16±0.75 | 64.20±1.16 | 64.66±1.33 | 64.43±2.12 |
| DP-SCAFFOLD | 49.36±1.03 | 55.01±2.78 | 56.42±4.00 | 59.33±1.50 | 60.55±1.30 | 59.94±1.99 | 61.05±1.82 |
| DP-FedAdam | 53.82±1.42 | 60.48±1.21 | 62.81±1.21 | **64.46±1.29** | **65.23±0.97** | **65.27±1.17** | 65.48±1.31 |
| DP-FedYogi | 49.39±3.52 | 58.08±2.31 | 61.95±1.16 | 63.52±0.93 | 64.66±1.06 | 65.26±1.12 | 65.41±1.34 |
| DP-FTRL | 33.15±3.94 | 47.36±0.32 | 58.68±2.64 | 61.36±2.59 | 62.91±0.33 | 63.29±0.31 | 63.17±0.15 |
| DP-AdaFedProx | 47.85±2.96 | 55.46±3.08 | 58.56±1.82 | 61.15±1.17 | 62.56±0.90 | 63.81±0.78 | 64.56±0.75 |
| DP-FedSOFIM | **56.13±2.01** | **60.67±2.09** | **63.66±1.02** | 64.24±1.31 | 65.16±1.59 | 64.87±1.76 | **66.26±1.36** |
| $\varepsilon = 1$ | | | | | | | |
| DP-FedGD | 51.26±1.49 | 58.18±1.40 | 61.22±0.87 | 63.31±0.43 | 64.59±0.12 | 65.10±0.29 | 65.68±0.18 |
| DP-FedAvg | 51.00±2.58 | 58.10±1.85 | 61.23±0.95 | 63.41±0.76 | 64.60±0.25 | 65.73±0.72 | 66.19±0.78 |
| DP-FedFC | 55.10±1.69 | 63.03±1.51 | 64.73±1.66 | 66.20±1.25 | 65.26±0.83 | 65.42±0.50 | 65.65±1.42 |
| DP-SCAFFOLD | 55.32±2.35 | 58.46±1.58 | 62.48±1.19 | 63.37±2.54 | 64.92±0.98 | 64.77±1.22 | 64.97±1.12 |
| DP-FedAdam | 51.77±4.08 | 61.33±1.22 | 63.87±0.29 | 65.31±0.67 | 67.14±0.38 | 67.53±0.60 | 68.07±0.36 |
| DP-FedYogi | 49.70±3.92 | 60.56±2.89 | 62.60±2.71 | 64.61±1.10 | 65.45±0.50 | 65.09±0.81 | 65.04±1.77 |
| DP-FTRL | 34.34±6.71 | 52.16±9.81 | 59.09±2.88 | 61.81±1.86 | 63.21±0.19 | 63.42±0.22 | 63.35±0.03 |
| DP-AdaFedProx | **59.81±1.07** | 63.22±0.05 | 65.36±0.31 | 66.64±0.34 | 66.09±0.60 | 66.67±0.36 | 67.14±0.72 |
| DP-FedSOFIM | 58.49±3.13 | **64.67±0.97** | **66.98±1.29** | **67.32±1.18** | **67.47±0.94** | **67.88±0.95** | **68.88±0.62** |
| $\varepsilon = 5$ | | | | | | | |
| DP-FedGD | 51.51±2.57 | 58.53±1.17 | 61.88±0.42 | 63.75±0.06 | 65.15±0.26 | 65.96±0.18 | 66.49±0.20 |
| DP-FedAvg | 51.50±2.58 | 58.52±1.16 | 61.86±0.42 | 63.77±0.05 | 65.16±0.27 | 65.94±0.19 | 66.50±0.18 |
| DP-FedFC | 43.48±0.53 | 55.09±1.29 | 52.10±6.59 | 56.43±4.22 | 56.48±4.71 | 60.87±1.72 | 62.04±3.69 |
| DP-SCAFFOLD | 60.73±1.13 | 65.79±0.22 | 67.64±0.20 | 68.60±0.45 | 69.20±0.26 | 69.43±0.30 | 69.77±0.37 |
| DP-FedAdam | 47.75±4.17 | 58.66±1.17 | 65.41±1.52 | 68.43±0.27 | 70.03±0.52 | 70.94±0.43 | 71.20±0.40 |
| DP-FedYogi | 59.09±0.38 | 66.43±1.04 | 68.43±0.75 | 69.53±0.66 | 70.39±0.06 | 70.66±0.17 | 71.03±0.22 |
| DP-FTRL | 28.48±2.75 | 49.88±2.16 | 54.25±3.45 | 59.81±2.74 | 63.55±0.93 | 65.09±0.69 | 65.20±0.19 |
| DP-AdaFedProx | 63.21±0.03 | 66.20±0.14 | 67.80±0.30 | 68.54±0.33 | 68.66±0.16 | 69.30±0.16 | 69.64±0.20 |
| DP-FedSOFIM | **64.22±2.23** | **67.34±1.66** | **68.89±0.71** | **70.30±0.51** | **70.63±0.39** | **71.10±0.31** | **71.54±0.31** |
| $\varepsilon = 10$ | | | | | | | |
| DP-FedGD | 51.49±2.64 | 58.51±1.23 | 61.86±0.44 | 63.89±0.10 | 65.17±0.22 | 65.98±0.28 | 66.62±0.23 |
| DP-FedAvg | 51.48±2.65 | 58.51±1.22 | 61.86±0.42 | 63.89±0.08 | 65.17±0.22 | 65.98±0.27 | 66.64±0.20 |
| DP-FedFC | 51.44±2.86 | 58.30±1.28 | 61.87±0.27 | 63.90±0.07 | 65.13±0.21 | 65.99±0.29 | 66.59±0.24 |
| DP-SCAFFOLD | 62.47±0.43 | 66.44±0.40 | 68.10±0.22 | 68.91±0.42 | 69.35±0.32 | 69.84±0.31 | 70.13±0.36 |
| DP-FedAdam | 56.77±4.06 | 63.37±3.97 | 66.92±0.72 | 67.72±0.80 | 70.03±0.61 | 70.11±0.32 | 70.97±0.27 |
| DP-FedYogi | 56.56±3.99 | 63.44±3.65 | 66.20±2.22 | 68.50±0.84 | 70.85±0.22 | 69.58±0.77 | 70.45±0.77 |
| DP-FTRL | 24.22±7.56 | 45.99±4.16 | 52.70±6.29 | 61.60±1.31 | 62.63±2.38 | 63.69±1.08 | 66.08±0.31 |
| DP-AdaFedProx | 63.50±0.19 | 66.50±0.18 | 67.87±0.32 | 68.63±0.40 | 68.99±0.34 | 69.59±0.22 | 69.81±0.28 |
| DP-FedSOFIM | **64.25±2.25** | **67.32±1.91** | **68.94±0.82** | **70.45±0.32** | **70.92±0.39** | **71.50±0.31** | **71.78±0.28** |

**Test Accuracy vs Rounds — CIFAR10, 20 clients, Non-IID (Dirichlet α=0.5)**

Figure 1: Convergence trajectories on CIFAR-10, ResNet-20 backbone, 20 clients, Non-IID (Dirichlet $\alpha = 0.5$) across privacy regimes.

Table 5: Test accuracy (%) on CIFAR-10 with VGG-16 backbone across federated rounds for different privacy budgets. Results shown at 10-round intervals (mean ± std over 3 seeds). Best result per privacy regime is in **bold**.

| Method | Federated Round | | | | | | |
|---|---|---|---|---|---|---|---|
| | 10 | 20 | 30 | 40 | 50 | 60 | 70 |
| $\varepsilon = 0.5$ | | | | | | | |
| DP-FedGD | 40.39±1.32 | 46.63±1.06 | 48.44±0.89 | 49.87±0.86 | 50.87±0.74 | 51.50±0.54 | 51.75±0.33 |
| DP-FedAvg | 41.97±1.32 | 47.20±0.92 | 48.75±0.97 | 50.13±0.88 | 51.03±0.68 | 51.61±0.55 | 51.92±0.31 |
| DP-FedFC | 36.63±3.70 | 43.96±2.09 | 46.19±1.26 | 47.51±1.48 | 48.33±1.15 | 48.91±1.19 | 48.96±1.27 |
| DP-SCAFFOLD | 37.65±3.03 | 39.46±3.60 | 43.27±2.15 | 44.74±0.81 | 44.06±1.52 | 44.52±0.90 | 45.69±1.49 |
| DP-FedAdam | 36.76±2.65 | 44.95±2.04 | 48.15±0.94 | 49.25±0.91 | 50.68±0.60 | 51.30±0.59 | 51.83±0.38 |
| DP-FedYogi | 34.05±3.46 | 42.56±2.30 | 46.88±1.06 | 48.74±0.98 | 50.06±0.86 | 50.94±0.69 | 51.58±0.45 |
| DP-FTRL | **49.53±1.76** | **51.59±0.24** | **51.93±0.85** | **52.28±0.87** | **52.52±1.05** | **52.41±1.11** | **52.38±0.88** |
| DP-AdaFedProx | 41.97±1.32 | 47.20±0.92 | 48.75±0.97 | 50.13±0.88 | 51.03±0.68 | 51.61±0.55 | 51.92±0.31 |
| DP-FedSOFIM | 40.66±1.96 | 46.01±2.79 | 49.15±1.67 | 50.44±1.13 | 51.39±0.95 | 51.90±0.55 | 51.97±0.47 |
| $\varepsilon = 1$ | | | | | | | |
| DP-FedGD | 47.38±0.53 | 50.68±0.29 | 51.75±0.33 | 52.53±0.35 | 53.13±0.42 | 53.66±0.40 | 53.71±0.13 |
| DP-FedAvg | 45.12±0.99 | 49.45±0.41 | 51.17±0.39 | 51.93±0.38 | 52.70±0.41 | 53.09±0.39 | 53.44±0.16 |

*Continued on next page*

*Continued from previous page*

| Method | Federated Round | | | | | | |
|---|---|---|---|---|---|---|---|
| | 10 | 20 | 30 | 40 | 50 | 60 | 70 |
| DP-FedFC | 24.51±4.38 | 35.28±3.78 | 42.23±2.78 | 45.43±2.03 | 47.71±1.25 | 49.34±0.91 | 50.30±0.75 |
| DP-SCAFFOLD | 42.06±2.15 | 46.63±1.30 | 48.22±0.87 | 48.30±0.44 | 49.50±0.56 | 49.78±0.75 | 50.12±0.84 |
| DP-FedAdam | 44.24±2.06 | 49.83±0.90 | 51.55±0.75 | 52.25±0.43 | 53.00±0.40 | 53.50±0.32 | 53.78±0.12 |
| DP-FedYogi | **48.95±0.66** | 47.47±1.52 | 47.87±1.47 | 48.57±0.61 | 47.72±0.58 | 48.33±0.54 | 48.14±1.13 |
| DP-FTRL | 44.65±0.97 | **53.48±0.11** | **53.36±0.29** | **53.49±0.22** | **53.68±0.33** | **53.69±0.37** | 53.74±0.17 |
| DP-AdaFedProx | 47.67±1.22 | 49.43±0.89 | 50.92±0.47 | 50.87±0.17 | 51.48±0.56 | 51.47±0.57 | 51.67±0.89 |
| DP-FedSOFIM | 47.43±0.85 | 50.76±0.90 | 52.19±0.63 | 53.13±0.41 | 53.97±0.26 | 53.98±0.17 | **53.99±0.51** |
| $\varepsilon = 5$ | | | | | | | |
| DP-FedGD | 48.67±0.66 | 51.62±0.13 | 52.93±0.23 | 53.31±0.15 | 53.90±0.20 | 54.34±0.21 | 54.70±0.10 |
| DP-FedAvg | 48.67±0.66 | 51.62±0.13 | 52.93±0.23 | 53.31±0.15 | 53.90±0.20 | 54.34±0.21 | 54.70±0.10 |
| DP-FedFC | 52.42±0.09 | 54.09±0.24 | 55.33±0.21 | 55.80±0.19 | 56.24±0.19 | 56.45±0.21 | 56.86±0.21 |
| DP-SCAFFOLD | 51.94±0.36 | 53.89±0.28 | 54.87±0.22 | 55.04±0.31 | 55.48±0.28 | 55.61±0.28 | 56.11±0.10 |
| DP-FedAdam | 48.52±1.71 | 53.57±1.06 | 54.59±1.73 | 55.10±0.26 | 55.12±0.38 | 55.72±0.31 | 55.51±0.11 |
| DP-FedYogi | 52.18±0.26 | 53.83±0.29 | 54.70±0.28 | 55.38±0.09 | 56.04±0.19 | 56.28±0.17 | 56.49±0.11 |
| DP-FTRL | 39.70±4.56 | 53.99±0.53 | 54.39±0.16 | 54.44±0.26 | 54.48±0.19 | 54.49±0.20 | 54.53±0.16 |
| DP-AdaFedProx | **53.16±0.48** | 54.51±0.33 | 55.47±0.44 | 55.95±0.43 | 56.55±0.30 | 56.82±0.27 | 57.05±0.27 |
| DP-FedSOFIM | 53.15±0.17 | **55.59±0.50** | **56.81±0.35** | **57.29±0.12** | **57.32±0.15** | **57.54±0.21** | **57.79±0.28** |
| $\varepsilon = 10$ | | | | | | | |
| DP-FedGD | 48.82±0.61 | 51.71±0.07 | 52.88±0.15 | 53.47±0.21 | 53.96±0.17 | 54.35±0.24 | 54.71±0.13 |
| DP-FedAvg | 48.82±0.61 | 51.71±0.07 | 52.88±0.15 | 53.47±0.21 | 53.96±0.17 | 54.35±0.24 | 54.71±0.13 |
| DP-FedFC | 43.45±2.90 | 50.24±0.43 | 51.81±0.22 | 52.60±0.12 | 53.16±0.08 | 53.60±0.15 | 54.00±0.13 |
| DP-SCAFFOLD | 52.83±0.28 | 54.53±0.25 | 55.38±0.29 | 55.87±0.08 | 56.43±0.20 | 56.63±0.09 | 56.96±0.06 |
| DP-FedAdam | 49.96±3.92 | 54.60±0.83 | 54.69±0.44 | 55.96±0.77 | 56.21±0.99 | 57.07±0.67 | 57.63±0.24 |
| DP-FedYogi | 50.37±3.90 | 54.53±0.51 | 55.25±0.86 | 55.38±1.48 | 56.66±0.84 | 57.18±0.50 | 57.46±0.24 |
| DP-FTRL | 39.74±3.14 | 52.37±0.78 | 55.09±0.23 | 55.11±0.14 | 55.03±0.18 | 55.01±0.23 | 55.00±0.24 |
| DP-AdaFedProx | **53.51±0.37** | 54.87±0.32 | 55.65±0.39 | 56.29±0.35 | 56.76±0.30 | 57.17±0.31 | 57.42±0.24 |
| DP-FedSOFIM | 53.29±0.18 | **56.01±0.40** | **57.44±0.10** | **58.14±0.15** | **58.35±0.17** | **58.70±0.19** | **58.99±0.15** |

Table 6: Test accuracy (%) on PathMNIST with VGG-16 backbone across federated rounds for different privacy budgets. Results shown at 10-round intervals (mean ± std over 3 seeds). Best result per privacy regime is in **bold**.

| Method | Federated Round | | | | | | |
|---|---|---|---|---|---|---|---|
| | 10 | 20 | 30 | 40 | 50 | 60 | 70 |
| $\varepsilon = 0.5$ | | | | | | | |
| DP-FedGD | 49.27±2.67 | 52.40±2.75 | 56.04±2.05 | **57.11±2.62** | 56.77±1.46 | 58.04±1.81 | 57.99±1.50 |
| DP-FedAvg | 49.27±2.67 | 52.40±2.75 | 56.04±2.05 | **57.11±2.62** | 56.77±1.46 | 58.04±1.81 | 57.99±1.50 |
| DP-FedFC | 42.66±1.54 | 52.88±1.01 | 56.08±1.19 | 57.10±1.63 | 58.00±2.01 | 58.28±1.85 | 58.78±0.94 |
| DP-SCAFFOLD | 47.56±3.32 | 50.42±1.20 | 54.26±1.37 | 54.87±2.73 | 54.19±2.20 | 55.81±2.00 | 55.43±0.56 |
| DP-FedAdam | **49.73±1.67** | 52.24±2.86 | 55.80±1.68 | 56.38±1.29 | 56.89±2.04 | 58.21±1.67 | 58.35±1.50 |
| DP-FedYogi | 48.65±2.04 | 52.27±2.41 | 55.85±1.29 | 56.23±1.55 | 57.40±1.82 | 58.38±1.37 | 58.47±2.20 |
| DP-FTRL | 44.22±1.61 | **54.56±2.83** | 55.55±2.55 | 55.54±2.84 | 55.25±2.45 | 55.33±2.66 | 55.27±2.70 |
| DP-AdaFedProx | 49.27±2.67 | 52.40±2.75 | 56.04±2.05 | **57.11±2.62** | 56.77±1.46 | 58.04±1.81 | 57.99±1.50 |
| DP-FedSOFIM | 48.85±1.82 | 52.33±1.62 | **57.21±0.39** | 56.30±0.87 | **58.63±1.02** | **58.72±1.19** | **59.57±1.63** |
| $\varepsilon = 1$ | | | | | | | |
| DP-FedGD | 51.06±3.20 | 54.35±2.88 | 57.27±2.21 | 58.37±2.30 | 58.44±1.40 | 59.29±1.66 | 59.43±1.56 |
| DP-FedAvg | 51.06±3.20 | 54.35±2.88 | 57.27±2.21 | 58.37±2.30 | 58.44±1.40 | 59.29±1.66 | 59.43±1.56 |
| DP-FedFC | 21.10±0.53 | 38.10±0.87 | 46.02±3.01 | 50.36±2.49 | 52.77±1.29 | 53.91±0.94 | 54.77±0.68 |
| DP-SCAFFOLD | 51.56±2.01 | 54.70±1.30 | 56.43±1.25 | 57.76±1.38 | 56.77±2.07 | 58.55±1.50 | 59.04±1.36 |

*Continued on next page*

*Continued from previous page*

| Method | Federated Round | | | | | | |
|---|---|---|---|---|---|---|---|
| | 10 | 20 | 30 | 40 | 50 | 60 | 70 |
| DP-FedAdam | 53.91±0.51 | 56.23±1.61 | 58.02±1.39 | 58.99±1.08 | 59.56±1.78 | 60.47±1.45 | 60.80±1.44 |
| DP-FedYogi | 48.71±3.59 | 54.88±3.42 | 55.79±0.69 | 59.26±1.90 | 58.00±1.28 | 60.25±1.37 | 60.15±1.32 |
| DP-FTRL | 44.03±6.25 | 54.91±3.55 | 56.24±2.44 | 56.26±2.55 | 56.06±2.33 | 56.18±2.41 | 56.20±2.44 |
| DP-AdaFedProx | **55.26±1.99** | 56.99±1.24 | 58.62±2.39 | 59.76±0.43 | 59.19±3.09 | 60.11±2.23 | 60.87±1.80 |
| DP-FedSOFIM | 54.29±3.39 | **58.34±0.67** | **58.64±1.65** | **61.19±1.56** | **61.96±0.60** | **61.23±1.33** | **62.09±0.33** |
| $\varepsilon = 5$ | | | | | | | |
| DP-FedGD | 50.66±3.03 | 54.59±2.73 | 57.08±2.14 | 58.23±2.00 | 58.85±1.75 | 59.66±1.70 | 60.09±1.63 |
| DP-FedAvg | 50.66±3.03 | 54.59±2.73 | 57.08±2.14 | 58.23±2.00 | 58.85±1.75 | 59.66±1.70 | 60.09±1.63 |
| DP-FedFC | 56.97±1.91 | 59.66±1.60 | 61.06±1.35 | 61.98±1.30 | 62.82±1.31 | 63.11±1.30 | 63.64±0.89 |
| DP-SCAFFOLD | 55.68±1.50 | 58.85±1.22 | 60.52±1.25 | 61.29±1.55 | 61.92±1.80 | 62.26±1.38 | 62.86±1.04 |
| DP-FedAdam | 46.86±4.29 | 56.19±1.67 | 59.29±1.39 | 60.61±1.79 | 64.30±0.73 | 64.20±0.81 | 65.13±0.63 |
| DP-FedYogi | 52.18±4.91 | 59.29±3.52 | 60.92±2.43 | 62.95±2.64 | 63.94±0.45 | 65.03±0.22 | **65.38±0.31** |
| DP-FTRL | 33.74±1.85 | 46.25±7.35 | 51.44±3.42 | 58.19±3.35 | 59.32±1.87 | 59.39±1.90 | 59.38±1.88 |
| DP-AdaFedProx | 57.41±2.45 | 59.42±2.03 | 61.23±2.06 | 61.91±1.79 | 62.32±1.92 | 62.49±2.07 | 63.22±1.87 |
| DP-FedSOFIM | **58.67±0.56** | **62.47±0.13** | **63.55±0.70** | **64.00±0.90** | **64.78±0.53** | **65.06±0.65** | 65.35±0.32 |
| $\varepsilon = 10$ | | | | | | | |
| DP-FedGD | 50.48±2.95 | 54.56±2.69 | 57.02±2.12 | 58.13±1.95 | 58.86±1.77 | 59.66±1.67 | 60.09±1.64 |
| DP-FedAvg | 50.48±2.95 | 54.56±2.69 | 57.02±2.12 | 58.13±1.95 | 58.86±1.77 | 59.66±1.67 | 60.09±1.64 |
| DP-FedFC | 47.60±3.61 | 54.30±1.05 | 55.93±1.66 | 57.07±1.94 | 57.97±1.93 | 58.71±1.91 | 59.19±1.87 |
| DP-SCAFFOLD | 56.84±1.72 | 59.45±1.45 | 61.15±1.17 | 61.98±1.39 | 62.56±1.53 | 63.09±1.23 | 63.31±1.05 |
| DP-FedAdam | 50.25±7.44 | 56.02±0.38 | 59.49±2.12 | 61.79±1.83 | 59.30±2.64 | 62.47±1.80 | 63.46±1.63 |
| DP-FedYogi | 50.11±7.99 | 55.90±1.31 | 57.87±1.70 | 60.17±1.48 | 60.41±1.03 | 62.26±1.39 | 63.51±1.60 |
| DP-FTRL | 38.91±2.98 | 47.66±4.62 | 54.90±1.30 | 59.75±1.12 | 59.84±1.45 | 59.68±1.89 | 59.74±1.89 |
| DP-AdaFedProx | 57.40±2.49 | 59.72±2.06 | 61.20±1.97 | 62.00±1.90 | 62.46±1.88 | 62.82±1.96 | 63.39±1.94 |
| DP-FedSOFIM | **58.60±0.76** | **62.58±0.39** | **63.54±1.00** | **64.59±1.02** | **65.05±0.51** | **65.36±0.58** | **65.79±0.40** |

## 5.3 Analysis

We analyze round-by-round test accuracy across four dataset/backbone combinations (ResNet-20 and VGG-16 on CIFAR-10 and PathMNIST; $n = 20$ clients, non-IID Dirichlet $\alpha = 0.5$; Tables 3–6; IID results in Appendix G), organized around five phenomena.

### 5.3.1 Phenomenon 1: DP-FedSOFIM Achieves Superior Early Convergence Across Architectures and Datasets

The primary and most consistent advantage of DP-FedSOFIM is convergence speed. It attains the best or joint-best round-10 accuracy in the majority of dataset/backbone/privacy configurations examined (Tables 3–6, Figures 1–2). On ResNet-20 CIFAR-10, the round-10 lead over the first-order baseline DP-FedGD reaches +20.31% at $\varepsilon = 5$ (61.05% vs. 40.74%; Table 3) and +20.73% at $\varepsilon = 10$ (61.52% vs. 40.79%; Table 3). On ResNet-20 PathMNIST, round-10 leads over DP-FedGD range from +5.34% at $\varepsilon = 0.5$ to +12.76% at $\varepsilon = 10$ (Table 4). With the VGG-16 backbone, DP-FedSOFIM leads or ties for first at round 10 in five of eight configurations (Tables 5–6).

These early leads translate directly into communication savings. On ResNet-20 CIFAR-10 at $\varepsilon \in \{5, 10\}$ (Table 3, Figure 1), setting the 95%-of-final-DP-FedGD target at 60.89%: DP-FedGD requires 50 rounds to clear it, while DP-FedSOFIM surpasses it already at round 10 (61.05% and 61.52%, respectively) and exceeds DP-FedGD's entire-run final accuracy before round 20. On ResNet-20 PathMNIST at $\varepsilon \in \{5, 10\}$ (Table 4, Figure 2), DP-FedSOFIM clears the analogous threshold at round 10 versus round 40 for DP-FedGD, a 4× reduction in rounds-to-target. The same pattern holds on VGG-16: at $\varepsilon = 10$ on CIFAR-10 (Table 5), DP-FedSOFIM at round 20 (56.01%) already surpasses DP-FedGD's round-70 final accuracy (54.71%). In

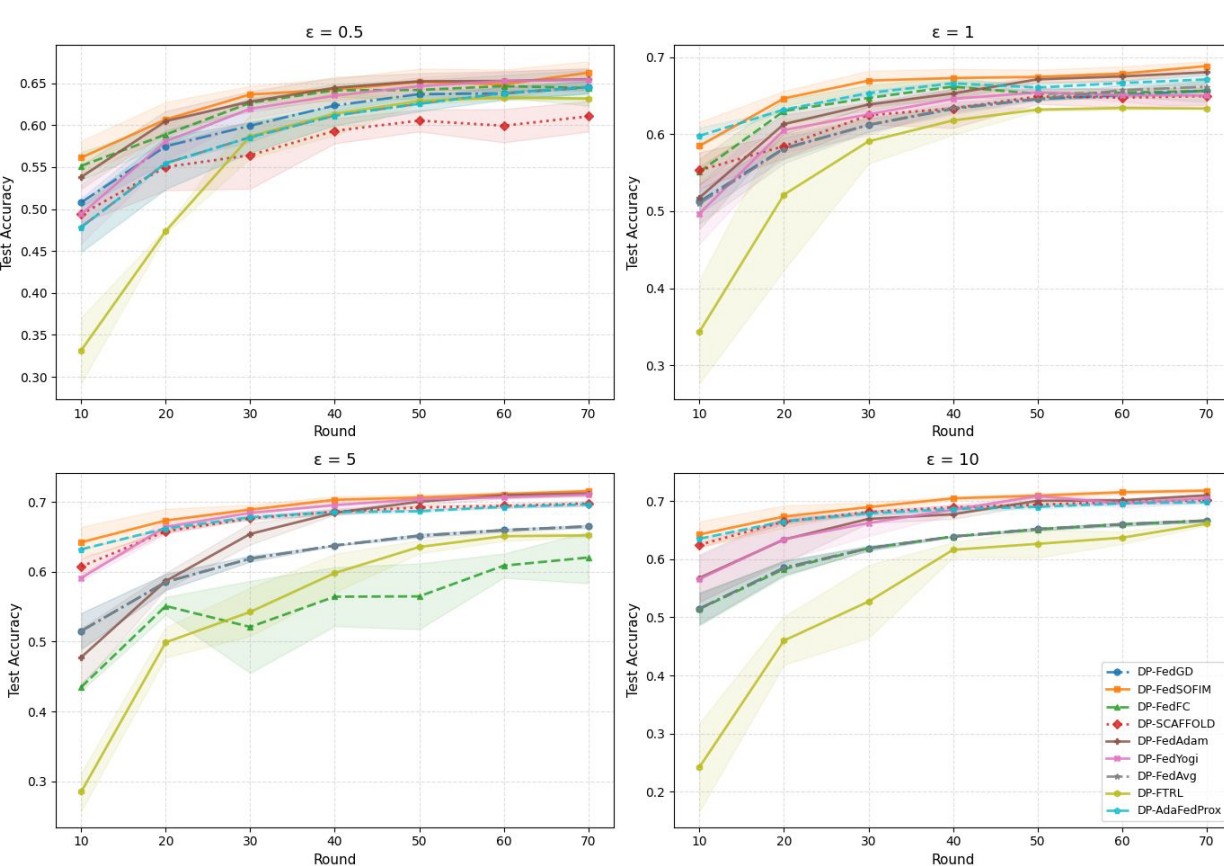

Figure 2: Convergence trajectories on PathMNIST, ResNet-20 backbone, 20 clients, Non-IID (Dirichlet $\alpha = 0.5$) across privacy regimes.

cross-device settings where each round incurs communication, latency, and on-device energy costs, this acceleration is decisive.

The mechanism is the rapid stabilization of the EMA momentum buffer. Initialized at $M_{-1} = 0_d$, the curvature proxy begins purely isotropic ($\widehat{\mathcal{I}}_{-1} = \rho I_d$, $H_{-1} = \rho^{-1} I_d$) and accumulates gradient signal as $M_t = (1 - \beta) \sum_{s \leq t} \beta^{t-s} G_s$. The EMA suppresses noise variance by a factor $\frac{1-\beta}{1+\beta}$, and because we tune $\beta$ upward as privacy tightens ($\beta = 0.9$ at $\varepsilon \geq 2$, rising to $\beta = 0.95$ at $\varepsilon = 0.5$), suppression strengthens exactly where per-round noise is largest. At $\varepsilon = 0.5$ we additionally employ a 20-round warmup in which the server applies EMA-only updates before activating the Sherman–Morrison step, ensuring the momentum buffer has accumulated sufficient signal before curvature preconditioning begins. As a result, DP-FedSOFIM exhibits no early-round instability even at the tightest budget; the preconditioner aligns with dominant gradient directions within the first 10–20 rounds.

### 5.3.2 Phenomenon 2: DP-FedSOFIM Leads Final-Round Accuracy in the Majority of Regimes and Saturates Early in the Remainder

DP-FedSOFIM attains the best round-70 accuracy in 12 of 16 dataset/backbone/privacy configurations (Tables 3–6). It sweeps all eight PathMNIST configurations across both backbones and wins on CIFAR-10 at $\varepsilon \in \{1, 10\}$ with ResNet-20 and at $\varepsilon \in \{1, 5, 10\}$ with VGG-16. In the four configurations where it does not hold the top position at round 70, three of the margins are negligible: $+0.37\%$ to DP-FedAdam on ResNet-20 CIFAR-10 $\varepsilon = 0.5$ ($59.23\%$ vs. $58.86\%$; Table 3), $+0.29\%$ to DP-FedYogi on ResNet-20 CIFAR-10 $\varepsilon = 5$

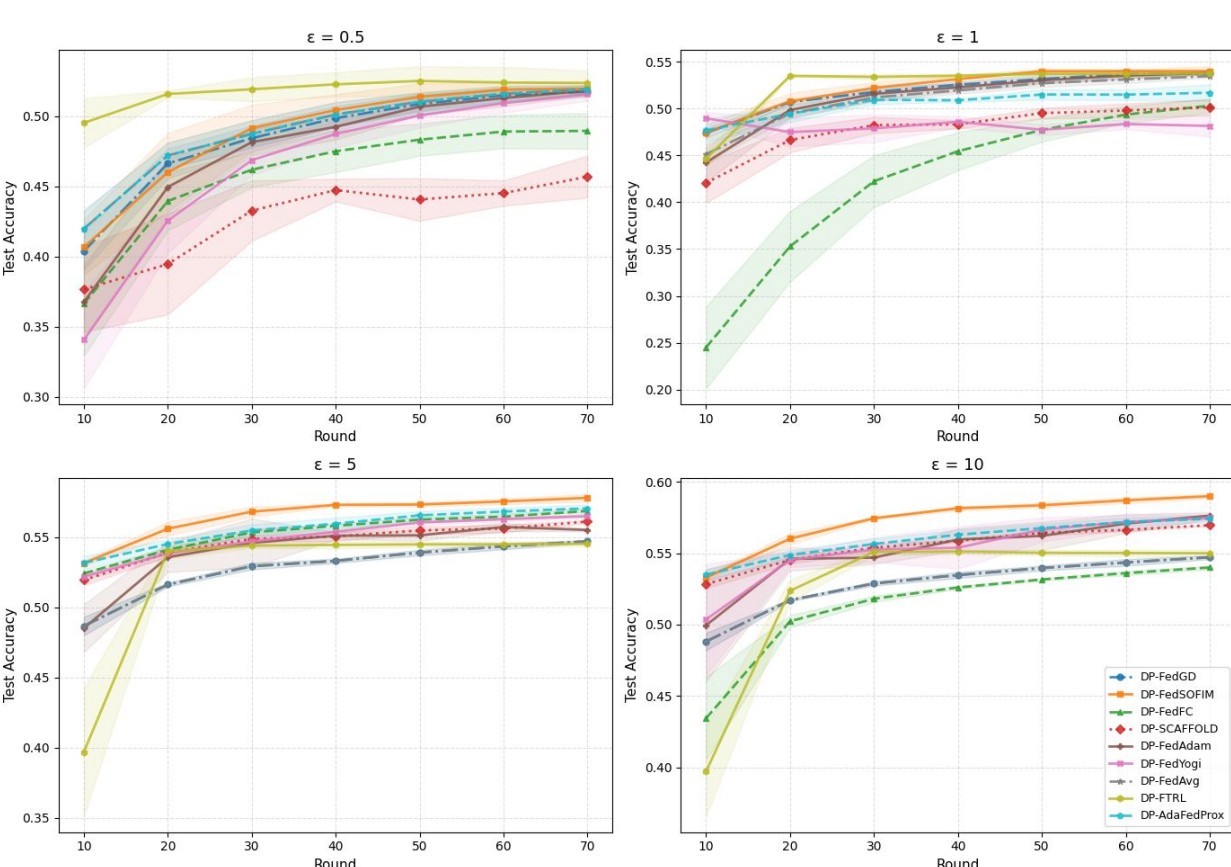

Figure 3: Convergence trajectories on CIFAR-10, VGG-16 backbone, 20 clients, Non-IID (Dirichlet $\alpha = 0.5$) across privacy regimes.

(67.81% vs. 67.52%; Table 3), and +0.03% to DP-FedYogi on VGG-16 PathMNIST $\varepsilon = 5$ (65.38% vs. 65.35%; Table 6), all within the run-to-run variability of the competing methods. The fourth, VGG-16 CIFAR-10 at $\varepsilon = 0.5$, is discussed under Phenomenon 4.

Crucially, in the regimes where adaptive baselines draw level at the final round on CIFAR-10, DP-FedSOFIM has already saturated near that accuracy long before. At $\varepsilon = 5$ on ResNet-20 CIFAR-10 (Table 3, Figure 1), DP-FedSOFIM reaches 66.89% by round 30 and settles at 67.52% by round 70; DP-FedYogi requires the full 70 rounds to arrive at 67.81%. DP-FedSOFIM has captured the bulk of its final performance before the training midpoint, while its competitors spend the remaining rounds closing a gap that DP-FedSOFIM has held throughout. In any communication-constrained setting where round budget is limited, this profile is strictly preferable.

### 5.3.3 Phenomenon 3: DP-FedAdam Is the Strongest Adaptive Baseline but Pays a Steep Early-Round Penalty

DP-FedAdam is the most consistently competitive baseline, particularly at relaxed privacy. It finishes second at round 70 on ResNet-20 PathMNIST at $\varepsilon \in \{1, 5, 10\}$ (68.07%, 71.20%, 70.97%; Table 4) and is a close contender on VGG-16 PathMNIST at the same budgets (60.80%, 65.13%, 63.46%; Table 6). Its adaptive server-side second-moment scaling thus rivals explicit Fisher preconditioning asymptotically, given sufficient rounds.

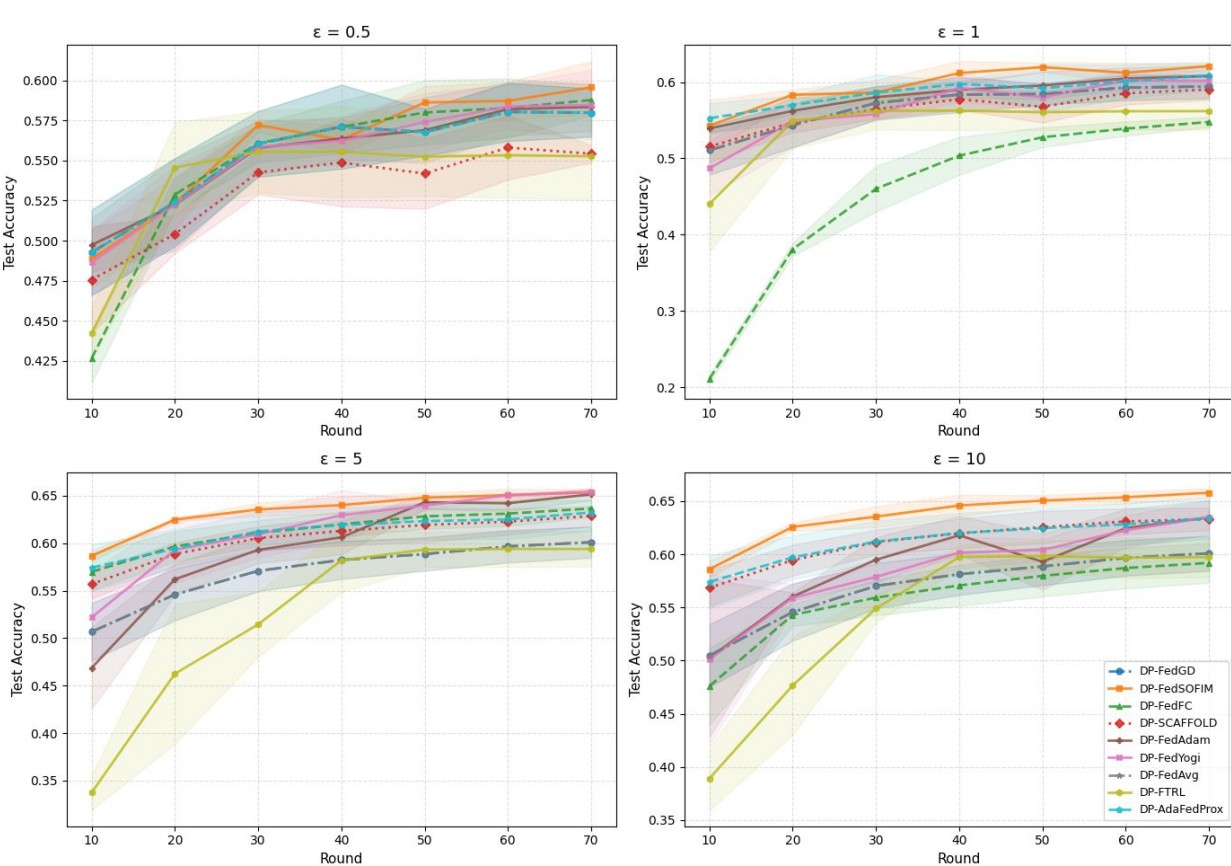

Figure 4: Convergence trajectories on PathMNIST, VGG-16 backbone, 20 clients, Non-IID (Dirichlet $\alpha = 0.5$) across privacy regimes.

However, DP-FedAdam's adaptive moments require several rounds of accumulation before the server step is well-scaled, producing slow and high-variance starts. At round 10, DP-FedSOFIM leads DP-FedAdam by +17.13% on ResNet-20 CIFAR-10 $\varepsilon = 5$ (61.05% vs. 43.92%; Table 3), +18.01% at $\varepsilon = 10$ (61.52% vs. 43.51%; Table 3), and +16.47% on ResNet-20 PathMNIST $\varepsilon = 5$ (64.22% vs. 47.75%; Table 4), with round-10 standard deviations as large as ±4.17% on PathMNIST (Table 4) and ±7.44% on VGG-16 PathMNIST $\varepsilon = 10$ (Table 6). DP-FedSOFIM's explicit Fisher proxy is informative from the very first rounds, making it the clearly superior choice whenever the round budget is limited. DP-AdaFedProx, the second-strongest adaptive baseline, leads DP-FedSOFIM narrowly at round 10 in a handful of regimes (e.g., ResNet-20 PathMNIST $\varepsilon = 1$: 59.81% vs. 58.49%; Table 4) but is overtaken by round 20 in every case and does not contend for the final-round lead in any configuration.

### 5.3.4 Phenomenon 4: DP-SCAFFOLD Degrades Under Tight Privacy; DP-FTRL's Budget-Accounting Advantage Is Confined to the Extreme-Tightness Regime

DP-SCAFFOLD exhibits a characteristic failure mode under tight privacy. On ResNet-20 CIFAR-10 at $\varepsilon = 0.5$ it reaches only 49.78% final accuracy, far below DP-FedGD (58.37%) and DP-FedSOFIM (58.86%; Table 3, Figure 1); at $\varepsilon = 1$ it again trails substantially (56.03% vs. 63.43% for DP-FedSOFIM; Table 3). The same pattern holds on ResNet-20 PathMNIST (Table 4), where it is the weakest method at $\varepsilon \in \{0.5, 1\}$ (61.05%, 64.97%). SCAFFOLD's variance-reduction relies on control variates maintained across rounds; under high noise these variates become corrupted and the client-drift correction amplifies rather than reduces variance. DP-SCAFFOLD recovers competitiveness once noise is low enough to preserve the variates: at $\varepsilon \in \{5, 10\}$ it

climbs to 66.42%/67.85% on ResNet-20 CIFAR-10 and 69.77%/70.13% on ResNet-20 PathMNIST (Tables 3–4), though it remains below DP-FedSOFIM throughout. On VGG-16, DP-SCAFFOLD is consistently outperformed by DP-FedSOFIM across all budgets (Tables 5–6), and its inability to maintain accurate control variates under stringent privacy makes it unsuitable for strong-privacy applications.

DP-FTRL presents a distinct profile. Its tree aggregation mechanism provides favorable effective privacy accounting at extreme budget tightness, yielding the best final-round accuracy on VGG-16 CIFAR-10 at $\varepsilon = 0.5$ (52.38%; Table 5). This is the single configuration across all 16 where DP-FTRL leads at the final round, and even here the margin over DP-FedSOFIM is only 0.41% (51.97%). Outside this narrow regime, DP-FTRL suffers from severe cold-start instability: its round-10 accuracy falls as low as 27.12% on ResNet-20 CIFAR-10 $\varepsilon = 10$ (Table 3) and 24.22% on ResNet-20 PathMNIST $\varepsilon = 10$ (Table 4), with round-10 standard deviations exceeding $\pm 7\%$ in multiple regimes. On ResNet-20 CIFAR-10 at $\varepsilon = 0.5$, DP-FTRL surges to 53.91% by round 20 before plateauing at 55.73% (Table 3), while DP-FedSOFIM continues to improve and reaches 58.86% at the final round. DP-FedSOFIM's server-side preconditioning degrades gracefully across the full privacy spectrum and is not susceptible to the cold-start pathology that renders DP-FTRL unreliable at all but the most extreme budgets.

### 5.3.5 Phenomenon 5: Dataset- and Backbone-Dependent Curvature Concentration

The patterns of Phenomena 1 and 2 reveal a consistent asymmetry across datasets (Tables 3–6, Figures 1–2): DP-FedSOFIM's early lead consolidates into a final-round win on PathMNIST across all eight configurations (both backbones, all four budgets), while on CIFAR-10 the late-round gap narrows at relaxed privacy budgets, particularly with ResNet-20. We interpret this geometrically, consistent with the operator and quadratic-form bounds of Lemma 4.14 and the bias-and-noise neighborhood guarantee of Theorem 4.21.

The benefit of Fisher-proxy preconditioning scales with the condition number of the effective loss landscape: greater curvature anisotropy implies larger and more durable gains from curvature-aware step scaling. We hypothesize that PathMNIST's tissue-level histopathology signal is concentrated in a small number of dominant gradient directions under both backbones. The momentum buffer aligns with these directions quickly, and because the remaining low-curvature directions carry little additional discriminative signal, DP-FedSOFIM's early lead is durable through round 70. On CIFAR-10 the effective curvature is more diffuse across both architectures: DP-FedSOFIM captures the dominant directions rapidly (producing the large early lead) but assigns lower weight to the many low-curvature directions that nonetheless carry slow, cumulative signal; adaptive baselines continue extracting that residual signal throughout and partially close the gap in late rounds. The CIFAR-10 late plateau is thus the signature of a diffuse loss landscape, not of premature convergence to a suboptimal basin.

Across backbones, DP-FedSOFIM generalizes well: it wins 6 of 8 final-round comparisons on each architecture, and three of the four non-winning margins are under 0.4% (Tables 3–6). The VGG-16 results confirm that the curvature-concentration effect extends beyond a single architecture. On VGG-16 PathMNIST, DP-FedSOFIM sweeps all four privacy budgets at the final round (Table 6); on VGG-16 CIFAR-10, the same diffuse-curvature dynamic present with ResNet-20 reappears at relaxed budgets, with the additional specific case at $\varepsilon = 0.5$ where DP-FTRL's tree aggregation provides a budget-accounting advantage that is marginal in magnitude and does not generalize to other budgets or datasets. This yields a practical guideline: DP-FedSOFIM is most advantageous in domains whose pretrained representation induces a structured, anisotropic loss landscape, such as medical imaging, and in any communication-constrained deployment where rounds-to-target rather than asymptotic accuracy is the binding constraint.

## 6 Conclusion

In this paper, we presented DP-FedSOFIM, a novel framework for differentially private second-order federated optimization. By shifting the computational burden of Fisher Information Matrix preconditioning entirely to the server and utilizing the Sherman–Morrison formula for $O(d)$ updates, we addressed the prohibitive memory constraints of existing second-order methods such as DP-FedNew and DP-FedFC, reducing client-side complexity from $O(d^2)$ to $O(d)$. Our theoretical analysis confirms that server-side preconditioning preserves

$(\varepsilon, \delta)$-differential privacy through the post-processing theorem, while convergence guarantees establish linear convergence to a noise-determined error floor under strongly convex objectives.

Empirically, we evaluated DP-FedSOFIM against eight baselines—DP-FedGD, DP-FedAvg (whose hyperparameter search over local steps yielded one as optimal, making it numerically equivalent to DP-FedGD in this setting), DP-FedFC, DP-SCAFFOLD, DP-FedAdam, DP-FedYogi, DP-FTRL, and DP-AdaFedProx—on CIFAR-10 and PathMNIST with ResNet-20 and VGG-16 backbones, $n = 20$ clients, and four privacy budgets ($\varepsilon \in \{0.5, 1, 5, 10\}$). The defining result is convergence speed. On ResNet-20, DP-FedSOFIM leads all methods at round 10 in 7 of 8 dataset/privacy configurations, with round-10 advantages over DP-FedGD as large as $+20.31\%$ on CIFAR-10 ($\varepsilon = 5$) and $+12.76\%$ on PathMNIST ($\varepsilon = 10$). This translates into roughly a 4–5$\times$ reduction in rounds-to-target: on ResNet-20 CIFAR-10 at $\varepsilon \in \{5, 10\}$, DP-FedSOFIM surpasses 95% of DP-FedGD's final accuracy at round 10 versus round 50 for DP-FedGD, and exceeds DP-FedGD's entire-run final accuracy before round 20. On VGG-16 the same acceleration holds at relaxed budgets, with DP-FedSOFIM's round-10 accuracy surpassing DP-FedGD's round-70 final accuracy by round 20 at $\varepsilon = 10$ on both datasets.

DP-FedSOFIM achieves the best final-round accuracy in 12 of 16 dataset/backbone/privacy configurations, including all eight PathMNIST regimes across both backbones. In the four configurations where it does not hold the top position, three margins are below 0.4% and within run-to-run variability, and in each case DP-FedSOFIM had led throughout training before competitors drew level only in the final rounds. Final-round gains over DP-FedGD reach up to $+4.46\%$ on CIFAR-10 ($\varepsilon = 10$, ResNet-20) and up to $+5.70\%$ on PathMNIST ($\varepsilon = 10$, VGG-16), with the larger medical-imaging margins reflecting the greater benefit of curvature-aware preconditioning on more anisotropic loss landscapes.

Our analysis identified five phenomena: substantially faster convergence from the earliest rounds, with the curvature-aware preconditioner front-loading its gains; a dataset-dependent late-round trajectory in which the early lead consolidates on PathMNIST and levels off on CIFAR-10, attributable to curvature concentration versus diffuseness in the effective loss landscape; DP-FedAdam emerging as the strongest adaptive baseline asymptotically but requiring many rounds to warm up its moment estimates; a characteristic failure mode of DP-SCAFFOLD under tight privacy, where corrupted control variates amplify rather than reduce variance; and DP-FTRL's tree-aggregation advantage being confined to the extreme-tight budget regime on VGG-16 CIFAR-10, accompanied by severe cold-start instability at all other configurations. DP-FedSOFIM, by contrast, degrades gracefully across the full privacy spectrum and across both architectures.

These results establish DP-FedSOFIM as a scalable, communication-efficient, and privacy-preserving solution for federated learning in privacy-critical domains such as healthcare and finance, where both strong formal privacy guarantees and high model utility are essential requirements.

## 7 Discussion and Future Work

We have introduced DP-FedSOFIM, a server-side second-order optimization framework that bridges the gap between the communication efficiency of first-order methods and the convergence stability of natural gradient descent. By leveraging the Sherman-Morrison matrix inversion identity, we maintain $O(d)$ memory and communication complexity, effectively overcoming the $O(d^2)$ bottleneck inherent in prior second-order federated methods like DP-FedNew (Krouka et al., 2025). Our theoretical framework establishes that preconditioning acts as a post-processing step, preserving the rigorous privacy guarantees of the Gaussian mechanism while significantly reducing the convergence error floor induced by noise. Beyond the strongly convex regime, the one-step descent argument underlying our analysis extends to smooth non-convex objectives without modification, yielding an $O(1/T)$ convergence rate to a stationary neighbourhood whose size is governed by the same clipping bias, privacy noise, and preconditioner coupling terms as in the convex case.

Empirically, DP-FedSOFIM exhibits strong early-round convergence across both ResNet-20 and VGG-16 backbones: the exponential moving average in the momentum buffer suppresses the Gaussian privacy noise within the first 10–20 rounds, so the rank-one Fisher proxy yields informative curvature from the earliest rounds. On ResNet-20, the optimizer leads all methods at round 10 in 7 of 8 dataset/privacy configurations; on VGG-16, where DP-AdaFedProx and DP-FTRL compete more closely at early rounds under tight budgets,

DP-FedSOFIM still recovers the lead by round 20 in nearly every regime and dominates from the midpoint of training onward. This stands in contrast to adaptive baselines such as DP-FedAdam, whose server-side moment estimates require several rounds to warm up before becoming effective, and to DP-FTRL, whose tree-aggregation mechanism yields a narrow advantage only at the extreme tight-budget regime ($\varepsilon = 0.5$) while suffering from severe cold-start instability elsewhere. These observations suggest that second-order methods are not only compatible with differential privacy but are arguably necessary for maintaining utility in communication-constrained environments. Several promising directions for future research remain:

**Structured Curvature:** Investigating Kronecker-factored (K-FAC) (Martens & Grosse, 2015) or block-diagonal approximations could capture higher-order inter-layer dependencies without sacrificing the $O(d)$ server-side efficiency.

**User-Level Privacy:** Transitioning from record-level to user-level DP presents a different sensitivity profile. Adapting the FIM preconditioning to handle the increased noise variance required for user-level protection is a critical next step for cross-device deployments.

**Adaptive Hyperparameters:** Automating the selection of the regularization parameter $\rho$ and momentum $\beta$ based on the target $\epsilon$ would further reduce the need for expensive grid searches in federated settings.

## Impact Statement

This work advances the field of Differentially Private Federated Learning (DP-FL), providing a scalable solution for training high-utility models on decentralized, sensitive data. The efficiency and convergence stability of DP-FedSOFIM have direct implications for high-stakes domains, such as:

- **Healthcare and Medical Imaging:** Enabling the training of diagnostic models across multiple hospitals without the need for data centralization, thereby respecting patient confidentiality and complying with regulations like GDPR or HIPAA.

- **Financial Analytics:** Facilitating collaborative fraud detection or risk assessment between institutions while protecting individual transaction records.

While our technical contribution tightens the privacy-utility tradeoff, we recognize that formal privacy guarantees are only one pillar of ethical AI. Practitioners must remain vigilant regarding algorithmic fairness, as the noise introduced by differential privacy can sometimes disproportionately affect accuracy for minority subgroups. Furthermore, the use of pre-trained feature extractors, while efficient, may inherit biases from the source data. We encourage the community to deploy DP-FedSOFIM alongside comprehensive fairness auditing and transparent data governance frameworks.

## Acknowledgements

The authors thank the Action Editor and the four anonymous reviewers for their insightful comments and constructive feedback, which substantially improved the manuscript.

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

## Appendices

## A Proofs of lemmas and main results

In this section, we provide the proofs of the lemmas and the main results.

*Proof of Lemma 4.10.* From (5),

$$\nu_t^2 = \frac{(C_g \sigma_g)^2}{n^3} \sum_{i=1}^{n} \frac{1}{|\mathcal{D}_i|^2}.$$

Since $|\mathcal{D}_i| \geq m_{\min}$, we have $|\mathcal{D}_i|^{-2} \leq m_{\min}^{-2}$, and hence

$$\sum_{i=1}^{n} \frac{1}{|\mathcal{D}_i|^2} \leq \frac{n}{m_{\min}^2}.$$

Substituting this into the expression for $\nu_t^2$ gives

$$\nu_t^2 \leq \frac{(C_g \sigma_g)^2}{n^3} \cdot \frac{n}{m_{\min}^2} = \frac{(C_g \sigma_g)^2}{n^2 m_{\min}^2} =: \nu^2.$$

Finally,

$$\mathbb{E}[\|\xi_t\|_2^2 \mid \mathcal{F}_t] = \mathrm{tr}\{\mathbb{E}[\xi_t \xi_t^\top \mid \mathcal{F}_t]\} = d\nu_t^2 \leq d\nu^2,$$

which proves the claim. □

*Proof of Lemma 4.11.* Fix $t \geq 0$. By (1), every clipped per-example gradient satisfies

$$\|\bar{g}(x,y)\|_2 \leq C_g.$$

Therefore, for each client $i$,

$$\left\| \frac{1}{|\mathcal{D}_i|} \sum_{(x,y) \in \mathcal{D}_i} \bar{g}(x,y) \right\|_2 \leq \frac{1}{|\mathcal{D}_i|} \sum_{(x,y) \in \mathcal{D}_i} \|\bar{g}(x,y)\|_2 \leq C_g.$$

where we used the triangle inequality. Averaging these client-wise quantities gives

$$\|g_{\mathrm{clip}}(\theta_t)\|_2 = \left\| \frac{1}{n} \sum_{i=1}^{n} \frac{1}{|\mathcal{D}_i|} \sum_{(x,y) \in \mathcal{D}_i} \bar{g}(x,y) \right\|_2$$

$$\leq \frac{1}{n} \sum_{i=1}^{n} \left\| \frac{1}{|\mathcal{D}_i|} \sum_{(x,y) \in \mathcal{D}_i} \bar{g}(x,y) \right\|_2 \leq \frac{1}{n} \sum_{i=1}^{n} C_g = C_g.$$

This proves (8).

Next, by (3),

$$G_t = g_{\mathrm{clip}}(\theta_t) + \xi_t.$$

Since $g_{\mathrm{clip}}(\theta_t)$ is $\mathcal{F}_t$-measurable and $\mathbb{E}[\xi_t \mid \mathcal{F}_t] = 0$,

$$\mathbb{E}\big[\|G_t\|_2^2 \mid \mathcal{F}_t\big] = \mathbb{E}\big[\|g_{\mathrm{clip}}(\theta_t) + \xi_t\|_2^2 \mid \mathcal{F}_t\big]$$

$$= \|g_{\mathrm{clip}}(\theta_t)\|_2^2 + 2\, g_{\mathrm{clip}}(\theta_t)^\top \mathbb{E}[\xi_t \mid \mathcal{F}_t] + \mathbb{E}[\|\xi_t\|_2^2 \mid \mathcal{F}_t]$$

$$= \|g_{\mathrm{clip}}(\theta_t)\|_2^2 + \mathrm{tr}\big(\mathbb{E}[\xi_t \xi_t^\top \mid \mathcal{F}_t]\big)$$

$$= \|g_{\mathrm{clip}}(\theta_t)\|_2^2 + d\nu_t^2.$$

The final bound follows from (8) and Lemma 4.10. □

*Proof of Lemma 4.12.* Fix $t \geq 0$. Using the recursion for $M_t$ and the decomposition $G_t = g_{\mathrm{clip}}(\theta_t) + \xi_t$,

$$M_t = \beta M_{t-1} + (1-\beta)g_{\mathrm{clip}}(\theta_t) + (1-\beta)\xi_t.$$

Conditional on $\mathcal{F}_t$, both $M_{t-1}$ and $g_{\mathrm{clip}}(\theta_t)$ are deterministic, whereas $\xi_t$ is mean-zero. Therefore

$$\mathbb{E}\big[\|M_t\|_2^2 \mid \mathcal{F}_t\big] = \|\beta M_{t-1} + (1-\beta)g_{\mathrm{clip}}(\theta_t)\|_2^2 + (1-\beta)^2 \mathbb{E}[\|\xi_t\|_2^2 \mid \mathcal{F}_t]. \tag{26}$$

Indeed, if we write

$$a_t := \beta M_{t-1} + (1-\beta)g_{\mathrm{clip}}(\theta_t),$$

then

$$\|a_t + (1-\beta)\xi_t\|_2^2 = \|a_t\|_2^2 + 2(1-\beta)a_t^\top \xi_t + (1-\beta)^2\|\xi_t\|_2^2,$$

and the mixed term vanishes after conditioning on $\mathcal{F}_t$.

Now use the convexity of the squared Euclidean norm: for every $a, b \in \mathbb{R}^d$ and every $\beta \in [0,1]$,

$$\|\beta a + (1-\beta)b\|_2^2 \leq \beta\|a\|_2^2 + (1-\beta)\|b\|_2^2.$$

Applying this to $a = M_{t-1}$ and $b = g_{\mathrm{clip}}(\theta_t)$, and invoking Lemma 4.11, gives

$$\|\beta M_{t-1} + (1-\beta)g_{\mathrm{clip}}(\theta_t)\|_2^2 \leq \beta\|M_{t-1}\|_2^2 + (1-\beta)C_g^2.$$

Substituting this and $\mathbb{E}[\|\xi_t\|_2^2 \mid \mathcal{F}_t] = d\nu^2$ into (26) proves (9).

Taking total expectation yields (10). To solve this recursion, let

$$b := (1-\beta)C_g^2 + (1-\beta)^2 d\nu^2.$$

Then $u_t \leq \beta u_{t-1} + b$, so by induction,

$$u_t \leq \beta^{t+1}u_{-1} + b\sum_{j=0}^{t}\beta^j.$$

Since $M_{-1} = 0_d$, we have $u_{-1} = 0$, hence

$$u_t \leq b\frac{1 - \beta^{t+1}}{1-\beta} = \big(1 - \beta^{t+1}\big)\big(C_g^2 + (1-\beta)d\nu^2\big) \leq C_g^2 + (1-\beta)d\nu^2.$$

This proves (11). □

*Proof of Lemma 4.14.* The identity (12) is the Sherman–Morrison formula applied to the rank-one perturbation $\rho I_d + M_t M_t^\top$:

$$(\rho I_d + uu^\top)^{-1} = \frac{1}{\rho}I_d - \frac{uu^\top}{\rho(\rho + \|u\|_2^2)}, \qquad u \in \mathbb{R}^d.$$

Substituting $u = M_t$ yields (12).

The matrix

$$\frac{M_t M_t^\top}{\rho(\rho + \|M_t\|_2^2)}$$

is positive semidefinite, hence

$$0 \preceq H_t \preceq \frac{1}{\rho}I_d.$$

The operator norm bound (13) follows immediately:

$$\|H_t v\|_2 \leq \|H_t\|_{\mathrm{op}}\|v\|_2 \leq \frac{1}{\rho}\|v\|_2.$$

Squaring both sides gives (14).

For the quadratic bounds, use (12) directly:

$$v^\top H_t v = \frac{1}{\rho}\|v\|_2^2 - \frac{(M_t^\top v)^2}{\rho(\rho + \|M_t\|_2^2)}.$$

The upper bound follows immediately because the second term is nonnegative:

$$v^\top H_t v \leq \frac{1}{\rho}\|v\|_2^2.$$

For the lower bound, by Cauchy–Schwarz, $(M_t^\top v)^2 \leq \|M_t\|_2^2\|v\|_2^2$. Hence

$$\frac{(M_t^\top v)^2}{\rho(\rho + \|M_t\|_2^2)} \leq \frac{\|M_t\|_2^2\|v\|_2^2}{\rho(\rho + \|M_t\|_2^2)}.$$

Therefore

$$v^\top H_t v \geq \left\{ \frac{1}{\rho} - \frac{\|M_t\|_2^2}{\rho(\rho + \|M_t\|_2^2)} \right\} \|v\|_2^2 = \frac{1}{\rho + \|M_t\|_2^2}\|v\|_2^2.$$

This proves (15). Substituting $v = \nabla F(\theta_t)$ gives

$$\nabla F(\theta_t)^\top H_t \nabla F(\theta_t) \geq \frac{1}{\rho + \|M_t\|_2^2}\|\nabla F(\theta_t)\|_2^2,$$

which proves (16). □

*Proof of Corollary 4.15.* By Lemma 4.12,

$$\mathbb{E}\|M_t\|_2^2 \leq \bar{M}^2 \qquad \text{for every } t \geq 0.$$

Therefore, by Markov's inequality,

$$\mathbb{P}\left(\|M_t\|_2^2 > M_\delta^2\right) \leq \frac{\bar{M}^2}{M_\delta^2} = \frac{\delta_M}{T}.$$

Applying the union bound over $t = 0, \ldots, T-1$ gives

$$\mathbb{P}\left(\max_{0 \leq t \leq T-1}\|M_t\|_2^2 > M_\delta^2\right) \leq \sum_{t=0}^{T-1}\mathbb{P}\left(\|M_t\|_2^2 > M_\delta^2\right) \leq \delta_M.$$

Thus, with probability at least $1 - \delta_M$,

$$\|M_t\|_2^2 \leq M_\delta^2 \qquad \text{for all } t = 0, \ldots, T-1.$$

On this event, Lemma 4.14 implies

$$\nabla F(\theta_t)^\top H_t \nabla F(\theta_t) \geq \frac{1}{\rho + \|M_t\|_2^2}\|\nabla F(\theta_t)\|_2^2 \geq \frac{1}{\rho + M_\delta^2}\|\nabla F(\theta_t)\|_2^2.$$

This proves the claim. □

*Proof of Lemma 4.16.* Set

$$\Delta_t := \theta_{t+1} - \theta_t = -\eta H_t G_t.$$

By $L$-smoothness, (6) gives

$$F(\theta_{t+1}) \leq F(\theta_t) + \langle \nabla F(\theta_t), \Delta_t \rangle + \frac{L}{2}\|\Delta_t\|_2^2. \tag{27}$$

Substituting $\Delta_t = -\eta H_t G_t$ yields

$$F(\theta_{t+1}) \leq F(\theta_t) - \eta \langle \nabla F(\theta_t), H_t G_t \rangle + \frac{L\eta^2}{2} \|H_t G_t\|_2^2. \tag{28}$$

We bound the linear and quadratic terms separately.

**Step 1: decomposition of the linear term.**

Using (3),

$$G_t = \nabla F(\theta_t) + \zeta_t + \xi_t.$$

Therefore

$$\langle \nabla F(\theta_t), H_t G_t \rangle = \nabla F(\theta_t)^\top H_t \nabla F(\theta_t) + \nabla F(\theta_t)^\top H_t \zeta_t + \nabla F(\theta_t)^\top H_t \xi_t. \tag{29}$$

All bounds below are *pathwise*, i.e. they hold for each realized $M_t$ and $\xi_t$. Hence no conditional independence between $H_t$ and $\xi_t$ is needed.

**Step 2: lower bound on the curvature term.**

Using the Sherman–Morrison representation (12), we also have the conservative bound

$$\nabla F(\theta_t)^\top H_t \nabla F(\theta_t) \geq \frac{1}{\rho} \|\nabla F(\theta_t)\|_2^2 - \frac{\|M_t\|_2^2}{\rho(\rho + \|M_t\|_2^2)} \|\nabla F(\theta_t)\|_2^2.$$

Since

$$\frac{1}{\rho(\rho + \|M_t\|_2^2)} \leq \frac{1}{\rho^2}$$

and $\|\nabla F(\theta_t)\|_2 \leq G_{\max}$, this yields

$$\nabla F(\theta_t)^\top H_t \nabla F(\theta_t) \geq \frac{1}{\rho} \|\nabla F(\theta_t)\|_2^2 - \frac{G_{\max}^2}{\rho^2} \|M_t\|_2^2. \tag{30}$$

**Step 3: control of the bias cross-term.**

Using Cauchy–Schwarz, (13), and Assumption 4.5,

$$\left| \nabla F(\theta_t)^\top H_t \zeta_t \right| \leq \|\nabla F(\theta_t)\|_2 \|H_t \zeta_t\|_2 \leq \frac{\|\nabla F(\theta_t)\|_2 \|\zeta_t\|_2}{\rho} \leq \frac{\|\nabla F(\theta_t)\|_2 \zeta_{\max}}{\rho}. \tag{31}$$

Apply Young's inequality $ab \leq \frac{\tau_1}{2} a^2 + \frac{1}{2\tau_1} b^2$ with

$$a = \|\nabla F(\theta_t)\|_2, \qquad b = \frac{\zeta_{\max}}{\rho},$$

to obtain

$$\left| \nabla F(\theta_t)^\top H_t \zeta_t \right| \leq \frac{\tau_1}{2} \|\nabla F(\theta_t)\|_2^2 + \frac{\zeta_{\max}^2}{2\tau_1 \rho^2}. \tag{32}$$

**Step 4: control of the noise cross-term.**

Similarly,

$$\left| \nabla F(\theta_t)^\top H_t \xi_t \right| \leq \|\nabla F(\theta_t)\|_2 \|H_t \xi_t\|_2 \leq \frac{\|\nabla F(\theta_t)\|_2 \|\xi_t\|_2}{\rho}. \tag{33}$$

Applying Young's inequality with

$$a = \|\nabla F(\theta_t)\|_2, \qquad b = \frac{\|\xi_t\|_2}{\rho},$$

gives

$$\left| \nabla F(\theta_t)^\top H_t \xi_t \right| \leq \frac{\tau_2}{2} \|\nabla F(\theta_t)\|_2^2 + \frac{\|\xi_t\|_2^2}{2\tau_2 \rho^2}. \tag{34}$$

Combining (29), (30), (32), and (34), we obtain the pathwise lower bound

$$\langle \nabla F(\theta_t), H_t G_t \rangle \geq \left( \frac{1}{\rho} - \frac{\tau_1 + \tau_2}{2} \right) \|\nabla F(\theta_t)\|_2^2 - \frac{G_{\max}^2}{\rho^2} \|M_t\|_2^2 \tag{35}$$
$$- \frac{\zeta_{\max}^2}{2\tau_1 \rho^2} - \frac{\|\xi_t\|_2^2}{2\tau_2 \rho^2}.$$

**Step 5: upper bound on the quadratic term.**

By (14),

$$\|H_t G_t\|_2^2 \leq \frac{1}{\rho^2} \|G_t\|_2^2.$$

Taking conditional expectation given $\mathcal{F}_t$ and then applying Lemma 4.11,

$$\mathbb{E}\big[ \|H_t G_t\|_2^2 \mid \mathcal{F}_t \big] \leq \frac{1}{\rho^2} \mathbb{E}\big[ \|G_t\|_2^2 \mid \mathcal{F}_t \big] \leq \frac{C_g^2 + d\nu^2}{\rho^2}. \tag{36}$$

**Step 6: combine the bounds and take conditional expectation.**

Substitute (35) into (28):

$$F(\theta_{t+1}) \leq F(\theta_t) - \eta \left( \frac{1}{\rho} - \frac{\tau_1 + \tau_2}{2} \right) \|\nabla F(\theta_t)\|_2^2 + \frac{\eta G_{\max}^2}{\rho^2} \|M_t\|_2^2$$
$$+ \frac{\eta \zeta_{\max}^2}{2\tau_1 \rho^2} + \frac{\eta}{2\tau_2 \rho^2} \|\xi_t\|_2^2 + \frac{L\eta^2}{2} \|H_t G_t\|_2^2.$$

Now we condition on $\mathcal{F}_t$. Since $F(\theta_t)$ and $\nabla F(\theta_t)$ are $\mathcal{F}_t$-measurable,

$$\mathbb{E}[\|\xi_t\|_2^2 \mid \mathcal{F}_t] = d\nu_t^2 \leq d\nu^2,$$

and (36) holds, we obtain

$$\mathbb{E}[F(\theta_{t+1}) \mid \mathcal{F}_t] \leq F(\theta_t) - \eta \left( \frac{1}{\rho} - \frac{\tau_1 + \tau_2}{2} \right) \|\nabla F(\theta_t)\|_2^2 + \frac{\eta G_{\max}^2}{\rho^2} \mathbb{E}\big[ \|M_t\|_2^2 \mid \mathcal{F}_t \big]$$
$$+ \frac{\eta \zeta_{\max}^2}{2\tau_1 \rho^2} + \frac{\eta \, d\nu^2}{2\tau_2 \rho^2} + \frac{L\eta^2}{2\rho^2} \big( C_g^2 + d\nu^2 \big).$$

This completes the proof. □

*Proof of Lemma 4.17.* Take total expectation in (18):

$$\mathbb{E}[F(\theta_{t+1})] \leq \mathbb{E}[F(\theta_t)] - \eta c_\nabla \, \mathbb{E}\|\nabla F(\theta_t)\|_2^2 + \frac{\eta G_{\max}^2}{\rho^2} \mathbb{E}\|M_t\|_2^2$$
$$+ \frac{\eta \zeta_{\max}^2}{2\tau_1 \rho^2} + \frac{\eta \, d\nu^2}{2\tau_2 \rho^2} + \frac{L\eta^2}{2\rho^2} \big( C_g^2 + d\nu^2 \big).$$

By Lemma 4.12,

$$\mathbb{E}\|M_t\|_2^2 \leq \bar{M}^2.$$

Substituting this bound yields (19). □

*Proof of Theorem 4.21.* By Lemma 4.17,

$$\mathbb{E}[F(\theta_{t+1})] \leq \mathbb{E}[F(\theta_t)] - \eta c_\nabla \, \mathbb{E}\|\nabla F(\theta_t)\|_2^2 + \Gamma.$$

By (7), strong convexity implies the pointwise bound

$$\|\nabla F(\theta_t)\|_2^2 \geq 2\mu\big(F(\theta_t) - F(\theta^\star)\big).$$

Taking expectations preserves the inequality:

$$\mathbb{E}\|\nabla F(\theta_t)\|_2^2 \geq 2\mu \, \mathbb{E}[F(\theta_t) - F(\theta^\star)].$$

Substituting this into the previous display gives

$$\mathbb{E}[F(\theta_{t+1}) - F(\theta^\star)] \leq \big(1 - 2\mu\eta c_\nabla\big)\mathbb{E}[F(\theta_t) - F(\theta^\star)] + \Gamma. \tag{37}$$

Define

$$\Delta_t := \mathbb{E}[F(\theta_t) - F(\theta^\star)].$$

Then (37) becomes

$$\Delta_{t+1} \leq r\Delta_t + \Gamma.$$

Repeatedly applying this recursion yields

$$\Delta_T \leq r^T \Delta_0 + \Gamma \sum_{j=0}^{T-1} r^j = r^T \Delta_0 + \Gamma \frac{1 - r^T}{1 - r},$$

which proves (21). Since $1 - r = 2\mu\eta c_\nabla$ and $1 - r^T \leq 1$, we obtain (22). $\qquad\square$

*Proof of Lemma 4.25.* By $L$-smoothness,

$$F(\theta_{t+1}^{\mathrm{GD}}) \leq F(\theta_t^{\mathrm{GD}}) - \eta\langle\nabla F(\theta_t^{\mathrm{GD}}), G_t\rangle + \frac{L\eta^2}{2}\|G_t\|_2^2.$$

Since $G_t = \nabla F(\theta_t^{\mathrm{GD}}) + \zeta_t + \xi_t$ and $\mathbb{E}[\xi_t \mid \mathcal{F}_t] = 0$,

$$\mathbb{E}[\langle\nabla F(\theta_t^{\mathrm{GD}}), G_t\rangle \mid \mathcal{F}_t] = \|\nabla F(\theta_t^{\mathrm{GD}})\|_2^2 + \langle\nabla F(\theta_t^{\mathrm{GD}}), \zeta_t\rangle.$$

Young's inequality gives

$$-\langle\nabla F(\theta_t^{\mathrm{GD}}), \zeta_t\rangle \leq \frac{\tau}{2}\|\nabla F(\theta_t^{\mathrm{GD}})\|_2^2 + \frac{\zeta_{\max}^2}{2\tau}.$$

Together with $\mathbb{E}[\|G_t\|_2^2 \mid \mathcal{F}_t] \leq C_g^2 + d\nu^2$, this proves the claim. $\qquad\square$

*Proof of Theorem 4.26.* Taking expectations in Lemma 4.25 and using strong convexity, $\|\nabla F(\theta)\|_2^2 \geq 2\mu\{F(\theta) - F(\theta^\star)\}$, gives

$$\mathbb{E}[F(\theta_{t+1}^{\mathrm{GD}}) - F(\theta^\star)] \leq (1 - 2\mu\eta c_{\mathrm{GD}})\mathbb{E}[F(\theta_t^{\mathrm{GD}}) - F(\theta^\star)] + \Gamma_{\mathrm{GD}}.$$

Iterating the recursion proves the result. $\qquad\square$

*Proof of Theorem 4.29.* Lemma 4.17 gives

$$\mathbb{E}[F(\theta_{t+1})] \leq \mathbb{E}[F(\theta_t)] - \eta c_\nabla \, \mathbb{E}\|\nabla F(\theta_t)\|_2^2 + \Gamma.$$

By the PL inequality (23),

$$\|\nabla F(\theta_t)\|_2^2 \geq 2\mu_{\mathrm{PL}}\big(F(\theta_t) - F(\theta^\star)\big)$$

holds pointwise, hence also after taking expectations:

$$\mathbb{E}\|\nabla F(\theta_t)\|_2^2 \geq 2\mu_{\mathrm{PL}} \, \mathbb{E}[F(\theta_t) - F(\theta^\star)].$$

Substituting this into the one-step descent bound yields

$$\mathbb{E}[F(\theta_{t+1}) - F(\theta^\star)] \leq \left(1 - 2\mu_{\mathrm{PL}}\eta c_\nabla\right)\mathbb{E}[F(\theta_t) - F(\theta^\star)] + \Gamma. \tag{38}$$

Defining

$$\Delta_t := \mathbb{E}[F(\theta_t) - F(\theta^\star)],$$

we obtain

$$\Delta_{t+1} \leq r_{\mathrm{PL}}\Delta_t + \Gamma.$$

Iterating the recursion and summing the geometric series proves (24); the simplified bound (25) follows from $1 - r_{\mathrm{PL}} = 2\mu_{\mathrm{PL}}\eta c_\nabla$. $\qquad\square$

*Proof of Proposition 4.19.* By $L$-smoothness,

$$F(\theta_{t+1}) \leq F(\theta_t) - \eta\langle\nabla F(\theta_t), H_t G_t\rangle + \frac{L\eta^2}{2}\|H_t G_t\|_2^2.$$

Write $G_t = \nabla F(\theta_t) + b_t + \xi_t$. Then

$$-\langle\nabla F(\theta_t), H_t G_t\rangle = -\nabla F(\theta_t)^\top H_t \nabla F(\theta_t) - \nabla F(\theta_t)^\top H_t b_t - \nabla F(\theta_t)^\top H_t \xi_t.$$

Using the same preconditioner bound (30),

$$\nabla F(\theta_t)^\top H_t \nabla F(\theta_t) \geq \frac{1}{\rho}\|\nabla F(\theta_t)\|_2^2 - \frac{G_{\max}^2}{\rho^2}\|M_t\|_2^2.$$

Also,

$$|\nabla F(\theta_t)^\top H_t b_t| \leq \frac{1}{\rho}\|\nabla F(\theta_t)\|_2\|b_t\|_2 \leq \frac{\tau_1}{2}\|\nabla F(\theta_t)\|_2^2 + \frac{B^2}{2\tau_1\rho^2},$$

and similarly,

$$\mathbb{E}\left[|\nabla F(\theta_t)^\top H_t \xi_t| \mid \mathcal{F}_t\right] \leq \frac{\tau_2}{2}\|\nabla F(\theta_t)\|_2^2 + \frac{d\nu^2}{2\tau_2\rho^2}.$$

Finally, since $\|H_t G_t\|_2 \leq \rho^{-1}\|G_t\|_2$,

$$\mathbb{E}[\|H_t G_t\|_2^2 \mid \mathcal{F}_t] \leq \frac{1}{\rho^2}\mathbb{E}[\|G_t\|_2^2 \mid \mathcal{F}_t] \leq \frac{G_0^2 + d\nu^2}{\rho^2}.$$

Combining the preceding bounds gives the conditional inequality. Taking expectations and using $\mathbb{E}\|M_t\|_2^2 \leq \overline{M}^2$ gives the unconditional bound. $\qquad\square$

*Proof of Theorem 4.31.* From Lemma 4.17,

$$\mathbb{E}[F(\theta_{t+1})] \leq \mathbb{E}[F(\theta_t)] - \eta c_\nabla \mathbb{E}\|\nabla F(\theta_t)\|_2^2 + \Gamma.$$

Rearranging gives

$$\eta c_\nabla \mathbb{E}\|\nabla F(\theta_t)\|_2^2 \leq \mathbb{E}[F(\theta_t)] - \mathbb{E}[F(\theta_{t+1})] + \Gamma.$$

Summing from $t = 0$ to $T - 1$ yields

$$\eta c_\nabla \sum_{t=0}^{T-1} \mathbb{E}\|\nabla F(\theta_t)\|_2^2 \leq F(\theta_0) - \mathbb{E}[F(\theta_T)] + T\Gamma.$$

Since $F(\theta_T) \geq F_{\inf}$, we obtain

$$\eta c_\nabla \sum_{t=0}^{T-1} \mathbb{E}\|\nabla F(\theta_t)\|_2^2 \leq F(\theta_0) - F_{\inf} + T\Gamma.$$

Dividing by $\eta c_\nabla T$ gives the first claim. The randomized-iterate statement follows because

$$\mathbb{E}\|\nabla F(\theta_R)\|_2^2 = \frac{1}{T}\sum_{t=0}^{T-1}\mathbb{E}\|\nabla F(\theta_t)\|_2^2.$$

$\square$

*Proof of Corollary 4.32.* The proof is identical to that of Theorem 4.31, using the descent recursion in Proposition 4.19 with $\Gamma_B$ in place of $\Gamma$. $\square$

*Proof of Lemma 4.35.* The mapping

$$(g_{1,t},\ldots,g_{n,t}) \mapsto \frac{1}{n}\sum_{i=1}^{n} g_{i,t}$$

is deterministic. Therefore $G_t$ is a deterministic post-processing of the released output $\mathcal{M}_t(\mathcal{D})$.

Since differential privacy is invariant under post-processing Dwork et al. (2014), the privacy guarantee is preserved. $\square$

*Proof of Lemma 4.36.* Condition on the previous server state $(\theta_t, M_{t-1})$. Once this state is fixed, the quantities

$$M_t = \beta M_{t-1} + (1-\beta)G_t, \qquad \widehat{\mathcal{I}}_t = \rho I_d + M_t M_t^\top, \qquad H_t = \widehat{\mathcal{I}}_t^{-1}, \qquad \theta_{t+1} = \theta_t - \eta_t H_t G_t$$

are deterministic functions of $G_t$.

Thus the mapping

$$G_t \mapsto (M_t, H_t, \theta_{t+1})$$

is deterministic. The result therefore follows from the post-processing property of differential privacy.

$\square$

## B  Computational Complexity Analysis

We now quantify the computational cost of DP-FedSOFIM and compare it with matrix-based second-order alternatives. The key point is that the proposed server-side preconditioner preserves the linear-in-d structure of first-order federated optimization while introducing only a negligible additional cost beyond standard aggregation.

**Lemma B.1** (Per-round complexity of DP-FedSOFIM)**.** *Let $n$ denote the number of participating clients and let $d$ denote the model dimension. Under full participation, one communication round of DP-FedSOFIM has total client-side computational cost $O(nd)$ and server-side computational cost $O(nd)$. The additional cost incurred by the SOFIM preconditioning step itself is only $O(d)$ per round, with $O(d)$ server memory.*

*Proof.* We separate the computation into the client and server components.

On the client side, each client $i$ computes per-example gradients in $\mathbb{R}^d$, clips them according to (1), forms the sum of clipped gradients, and adds Gaussian noise before normalization. At the level of vector operations, both gradient clipping and the formation of the released update require work linear in $d$. Thus the computational cost per client is $O(d)$, and summing over the $n$ participating clients yields an overall client-side cost of $O(nd)$ per round.

On the server side, the first step is the aggregation

$$G_t = \frac{1}{n}\sum_{i=1}^{n} g_{i,t},$$

which requires summing $n$ vectors in $\mathbb{R}^d$ and therefore costs $O(nd)$. The momentum update

$$M_t = \beta M_{t-1} + (1 - \beta)G_t$$

requires one vector scaling and one vector addition, hence $O(d)$ operations.

The distinctive step in DP-FedSOFIM is the application of the rank-one preconditioner. Using the Sherman–Morrison representation,

$$H_t G_t = \frac{1}{\rho} G_t - \frac{M_t^\top G_t}{\rho^2 + \rho\|M_t\|_2^2}\, M_t,$$

the preconditioned direction can be computed using two inner products, one scalar division, one scalar-vector multiplication, and one vector subtraction. Each of these operations is linear in $d$, so the total cost of forming $H_t G_t$ is $O(d)$. The parameter update

$$\theta_{t+1} = \theta_t - \eta H_t G_t$$

is again a single vector operation and therefore costs $O(d)$.

Combining these terms, the total server-side cost per round is

$$O(nd) + O(d) = O(nd),$$

with the aggregation step dominating the asymptotics. The additional curvature computation introduced by DP-FedSOFIM is therefore only $O(d)$ beyond the standard first-order federated pipeline. Finally, since the server stores only the current vectors $G_t$, $M_t$, and $\theta_t$, the additional server memory required by the preconditioner is $O(d)$.

In contrast, matrix-based second-order methods that maintain a dense curvature surrogate require at least $O(d^2)$ storage and typically $O(d^3)$ inversion or factorization cost per round. DP-FedSOFIM avoids these costs by exploiting the rank-one structure of the SOFIM approximation. $\qquad\square$

The preceding result shows that DP-FedSOFIM preserves the principal computational advantage of first-order federated optimization. In particular, the method does not require clients to transmit or store any matrix-valued curvature information, and all curvature-aware computation remains confined to a linear-time server-side post-processing step.

**Runtime Analysis**

Table 7 reports the average wall-clock time per communication round across 70 rounds on CIFAR-10 and PathMNIST under both backbone architectures ($n = 20$, non-IID Dirichlet $\alpha = 0.5$, seed=42). All methods use $K = 1$ local step per round except DP-SCAFFOLD and DP-AdaFedProx at $\varepsilon \geq 1$, which use the $K$ values selected by hyperparameter search at each privacy level. VGG-16 incurs approximately 3.5–4× longer per-round times than ResNet-20 across all methods and datasets, owing to its larger feature dimension (512 vs. 64), which increases the per-example gradient computation cost. PathMNIST runs are slightly slower than CIFAR-10 under the same backbone due to its larger training set ($\approx$4500 samples per client vs. $\approx$2500 for CIFAR-10). DP-FedSOFIM adds negligible overhead over DP-FedGD in all four settings ($<2\%$), confirming the $O(d)$ complexity of the Sherman–Morrison preconditioning step established in Lemma B.1. DP-SCAFFOLD incurs significantly higher training cost at $\varepsilon \geq 1$ due to its multiple local steps ($K = 5$ at $\varepsilon = 1$, $K = 7$ at $\varepsilon \geq 5$), reaching up to $\approx$6× the wall-clock time of DP-FedSOFIM on PathMNIST/VGG at $\varepsilon \geq 5$. DP-AdaFedProx likewise scales with $K$, requiring $\approx$4–5× longer total training time than DP-FedSOFIM at $\varepsilon \geq 1$.

## C  Variance Reduction via the Momentum Buffer

An important feature of DP-FedSOFIM is that the curvature proxy is not formed from a single privatized aggregate, but from the exponentially weighted moving average

$$M_t = \beta M_{t-1} + (1 - \beta)G_t.$$

Table 7: Average wall-clock time per round (seconds), $n = 20$ clients, non-IID Dirichlet $\alpha = 0.5$, seed=42, $T = 70$ rounds. Runtimes for all methods except DP-SCAFFOLD and DP-AdaFedProx are invariant to $\varepsilon$.

| | CIFAR-10 | | PathMNIST | |
|---|---|---|---|---|
| **Algorithm** | ResNet-20 | VGG-16 | ResNet-20 | VGG-16 |
| DP-FedGD | 0.04 | 0.15 | 0.05 | 0.23 |
| DP-FedSOFIM | 0.04 | 0.15 | 0.05 | 0.22 |
| DP-FedAvg | 0.04 | 0.14 | 0.05 | 0.23 |
| DP-FedAdam | 0.04 | 0.15 | 0.06 | 0.22 |
| DP-FedYogi | 0.04 | 0.15 | 0.05 | 0.22 |
| DP-FedFC | 0.04 | 0.14 | 0.05 | 0.22 |
| DP-FTRL | 0.04 | 0.16 | 0.05 | 0.23 |
| DP-AdaFedProx ($\varepsilon = 0.5$, $K = 1$) | 0.04 | 0.15 | 0.05 | 0.22 |
| DP-AdaFedProx ($\varepsilon \geq 1$, $K = 5$) | 0.18 | 0.71 | 0.23 | 1.07 |
| *DP-SCAFFOLD$^{\dagger}$ (per privacy level)* | | | | |
| $\varepsilon = 0.5$ ($K = 1$) | 0.04 | 0.14 | 0.05 | 0.21 |
| $\varepsilon = 1$ ($K = 5$) | 0.17 | 0.68 | 0.21 | 1.07 |
| $\varepsilon = 5$ ($K = 7$) | 0.24 | 0.93 | 0.30 | 1.43 |
| $\varepsilon = 10$ ($K = 7$) | 0.24 | 0.93 | 0.30 | 1.44 |

$^{\dagger}$ DP-SCAFFOLD uses $K \in \{1, 5, 7, 7\}$ local steps at $\varepsilon \in \{0.5, 1, 5, 10\}$ respectively. All other methods use $K = 1$. Total training time = avg round time $\times 70$ plus fixed initialisation overhead ($<1\,\mathrm{s}$).

Since the aggregated gradient $G_t$ contains Gaussian privacy noise, this recursion implicitly smooths the noise across rounds. The next result makes this effect precise.

**Lemma C.1** (Variance reduction induced by exponential averaging). *Suppose that the aggregated gradient admits the decomposition*

$$G_t = \nabla F(\theta_t) + \xi_t,$$

*where $\{\xi_t\}_{t \geq 0}$ are independent mean-zero Gaussian vectors with covariance $\nu^2 I_d$. If*

$$M_t = (1 - \beta) \sum_{s=0}^{t} \beta^{t-s} G_s, \qquad \beta \in [0, 1),$$

*then the noise component of $M_t$ has covariance*

$$\mathrm{Cov}(M_t) = \nu^2 (1 - \beta)^2 \sum_{s=0}^{t} \beta^{2(t-s)} I_d = \nu^2 (1 - \beta)^2 \frac{1 - \beta^{2(t+1)}}{1 - \beta^2} I_d.$$

*Consequently, as $t \to \infty$,*

$$\mathrm{Cov}(M_t) \longrightarrow \frac{1 - \beta}{1 + \beta} \nu^2 I_d.$$

*Proof.* Expanding the recursion for $M_t$ gives

$$M_t = (1 - \beta) \sum_{s=0}^{t} \beta^{t-s} \big( \nabla F(\theta_s) + \xi_s \big).$$

Only the Gaussian noise terms contribute to the covariance, so it suffices to consider

$$M_t^{\mathrm{noise}} = (1 - \beta) \sum_{s=0}^{t} \beta^{t-s} \xi_s.$$

Because the vectors $\xi_s$ are independent and satisfy

$$\mathbb{E}[\xi_s] = 0, \qquad \text{Cov}(\xi_s) = \nu^2 I_d,$$

all cross-covariance terms vanish, and we obtain

$$\text{Cov}(M_t) = (1-\beta)^2 \sum_{s=0}^{t} \beta^{2(t-s)} \text{Cov}(\xi_s)$$

$$= \nu^2 (1-\beta)^2 \sum_{s=0}^{t} \beta^{2(t-s)} I_d.$$

Evaluating the geometric sum yields

$$\sum_{s=0}^{t} \beta^{2(t-s)} = \sum_{j=0}^{t} \beta^{2j} = \frac{1 - \beta^{2(t+1)}}{1 - \beta^2},$$

and therefore

$$\text{Cov}(M_t) = \nu^2 (1-\beta)^2 \frac{1 - \beta^{2(t+1)}}{1 - \beta^2} I_d.$$

Since $1 - \beta^2 = (1-\beta)(1+\beta)$, the limit as $t \to \infty$ is

$$\text{Cov}(M_t) = \frac{1-\beta}{1+\beta} \nu^2 I_d.$$

This proves the claim. $\qquad\qquad\qquad\qquad\qquad\qquad\qquad\qquad\qquad\qquad\qquad\qquad\qquad\qquad$ $\square$

The lemma shows that the moving-average buffer attenuates the isotropic privacy noise by the factor $(1-\beta)/(1+\beta)$. For values such as $\beta = 0.9$, this factor is approximately $0.0526$, corresponding to a variance reduction of about $19\times$ relative to the raw per-round noise. This provides a useful quantitative explanation for why the rank-one preconditioner built from $M_t$ remains stable even when the individual privatized aggregates are substantially corrupted by Gaussian noise.

## D    Sherman–Morrison Derivation for the Regularized SOFIM Preconditioner

For completeness, we record the derivation of the closed-form inverse used in DP-FedSOFIM. At round $t$, the server constructs the regularized rank-one curvature approximation

$$\widehat{\mathcal{F}}_t = M_t M_t^\top + \rho I_d,$$

with regularization parameter $\rho > 0$. The corresponding preconditioner is

$$H_t = \widehat{\mathcal{F}}_t^{-1}.$$

A direct inversion of a dense $d \times d$ matrix would cost $O(d^3)$, but the rank-one structure of $\widehat{\mathcal{F}}_t$ permits an exact closed form.

The Sherman–Morrison identity states that for any invertible matrix $A$ and vectors $u, v$ such that $1 + v^\top A^{-1} u \neq 0$,

$$(A + uv^\top)^{-1} = A^{-1} - \frac{A^{-1} uv^\top A^{-1}}{1 + v^\top A^{-1} u}.$$

Applying this identity with

$$A = \rho I_d, \qquad u = v = M_t,$$

we obtain

$$H_t = (\rho I_d + M_t M_t^\top)^{-1}$$
$$= \frac{1}{\rho} I_d - \frac{\frac{1}{\rho} M_t M_t^\top \frac{1}{\rho}}{1 + M_t^\top \frac{1}{\rho} M_t}.$$

Since

$$1 + M_t^\top \frac{1}{\rho} M_t = \frac{\rho + \|M_t\|_2^2}{\rho},$$

this simplifies to

$$H_t = \frac{1}{\rho} I_d - \frac{M_t M_t^\top}{\rho^2 + \rho \|M_t\|_2^2},$$

which is exactly the formula used in the main text.

This representation is particularly valuable because one never needs to form the matrix $H_t$ explicitly. Indeed, applying the preconditioner to the aggregated gradient yields

$$H_t G_t = \left( \frac{1}{\rho} I_d - \frac{M_t M_t^\top}{\rho^2 + \rho \|M_t\|_2^2} \right) G_t = \frac{1}{\rho} G_t - \frac{M_t^\top G_t}{\rho^2 + \rho \|M_t\|_2^2} M_t.$$

Thus the action of $H_t$ on $G_t$ reduces to two inner products and a small number of vector operations, all of which are linear in $d$. This is why the preconditioning step adds only $O(d)$ computation and $O(d)$ storage to the standard federated gradient pipeline.

The formula also makes the geometry transparent. Along the direction spanned by $M_t$, the effective scaling is

$$\frac{1}{\rho + \|M_t\|_2^2},$$

whereas any direction orthogonal to $M_t$ is scaled by $1/\rho$. The preconditioner therefore contracts the update more strongly along the dominant historical gradient direction while leaving orthogonal directions less damped. This anisotropic rescaling is precisely the mechanism through which the rank-one SOFIM approximation captures curvature information at linear cost.

# E    Hockey-Stick Divergence and Privacy Accounting

We now describe the privacy accounting used to calibrate the Gaussian noise multiplier $\sigma_g$ for a target privacy budget $(\varepsilon, \delta)$ over $T$ communication rounds under full client participation. Since the client-side release mechanism in DP-FedSOFIM is identical to that of DP-FedGD, the privacy analysis is entirely determined by the noisy release of the clipped gradients; the momentum buffer, Fisher proxy construction, and Sherman–Morrison preconditioning are deterministic post-processing steps and therefore do not contribute additional privacy loss.

At round $t$, client $i$ releases

$$g_{i,t} = \frac{1}{|\mathcal{D}_i|} \left( \sum_{(x,y) \in \mathcal{D}_i} \bar{g}(x,y) + E_{i,t} \right), \qquad E_{i,t} \sim \mathcal{N}\left( 0, \frac{(C_g \sigma_g)^2}{n} I_d \right).$$

We adopt record-level replace-one adjacency: two federated datasets are neighboring if they differ in exactly one record in exactly one client's local dataset. Under this notion of adjacency, changing a single record can alter the sum of clipped gradients on client $i$ by at most $2C_g$ in Euclidean norm. After normalization by $|\mathcal{D}_i|$, the $\ell_2$-sensitivity of the released client update is therefore

$$\Delta_i = \frac{2C_g}{|\mathcal{D}_i|}.$$

To obtain a single conservative bound across all clients, we use

$$|\mathcal{D}_{\min}| := \min_i |\mathcal{D}_i|, \qquad \Delta := \frac{2C_g}{|\mathcal{D}_{\min}|}.$$

Because the Gaussian perturbation is added before normalization, the released vector $g_{i,t}$ has noise standard deviation

$$\sigma_{\text{release}} = \frac{C_g \sigma_g}{\sqrt{n}\, |\mathcal{D}_i|}.$$

Again, for conservative accounting, we upper bound using $|\mathcal{D}_{\min}|$, giving

$$\sigma_{\text{release}} = \frac{C_g \sigma_g}{\sqrt{n}\, |\mathcal{D}_{\min}|}.$$

For a single Gaussian mechanism with sensitivity $\Delta$ and noise standard deviation $\sigma_{\text{release}}$, the exact $(\varepsilon, \delta)$-tradeoff under the hockey-stick divergence is

$$\delta(\varepsilon) = \Phi\left(-\frac{\varepsilon \sigma_{\text{release}}}{\Delta} + \frac{\Delta}{2\sigma_{\text{release}}}\right) - e^\varepsilon \Phi\left(-\frac{\varepsilon \sigma_{\text{release}}}{\Delta} - \frac{\Delta}{2\sigma_{\text{release}}}\right),$$

where $\Phi$ denotes the standard normal distribution function. Under $T$ rounds with independent Gaussian perturbations, the corresponding composed expression becomes

$$\delta(\varepsilon) = \Phi\left(-\frac{\varepsilon \sigma_{\text{release}}}{\sqrt{T}\,\Delta} + \frac{\sqrt{T}\,\Delta}{2\sigma_{\text{release}}}\right) - e^\varepsilon \Phi\left(-\frac{\varepsilon \sigma_{\text{release}}}{\sqrt{T}\,\Delta} - \frac{\sqrt{T}\,\Delta}{2\sigma_{\text{release}}}\right).$$

Substituting the mechanism parameters

$$\Delta = \frac{2C_g}{|\mathcal{D}_{\min}|}, \qquad \sigma_{\text{release}} = \frac{C_g \sigma_g}{\sqrt{n}\, |\mathcal{D}_{\min}|},$$

yields

$$\delta(\varepsilon) = \Phi\left(\frac{\sqrt{nT}}{\sigma_g} - \frac{\varepsilon \sigma_g}{2\sqrt{nT}}\right) - e^\varepsilon \Phi\left(-\frac{\sqrt{nT}}{\sigma_g} - \frac{\varepsilon \sigma_g}{2\sqrt{nT}}\right).$$

Given a target pair $(\varepsilon, \delta)$, we determine the smallest $\sigma_g$ satisfying the inequality

$$\delta(\varepsilon) \leq \delta$$

by a one-dimensional numerical search, such as binary search over a suitably large interval.

A few remarks are worth making. First, because all clients participate at every round, there is no privacy amplification by subsampling in the present setting; the accounting above therefore corresponds directly to full participation. Second, the factor $1/\sqrt{n}$ appearing in the release noise scale arises solely from the client-side parameterization of the Gaussian perturbation in (2); it should not be interpreted as a consequence of subsampling or privacy amplification. Third, the hockey-stick divergence yields the exact privacy tradeoff for the Gaussian mechanism and is therefore tighter than moment-based upper bounds such as standard Rényi-DP conversions in many regimes.

Most importantly for the present paper, the privacy guarantee of DP-FedSOFIM is identical to that of the underlying DP-FedGD release mechanism. The server-side updates

$$M_t = \beta M_{t-1} + (1-\beta)G_t, \qquad H_t = (\rho I_d + M_t M_t^\top)^{-1}, \qquad \theta_{t+1} = \theta_t - \eta H_t G_t$$

depend only on previously released privatized quantities and therefore preserve privacy by the post-processing theorem. Consequently, once the noise multiplier $\sigma_g$ has been calibrated for the client-side Gaussian mechanism, the same $(\varepsilon, \delta)$-DP guarantee carries over unchanged to DP-FedSOFIM.

# F Privacy Analysis of the Gaussian Mechanism

This appendix derives the differential privacy parameters of the privatized client update defined in (2).

**Sensitivity of the clipped client update** Let $\mathcal{D}_i$ and $\mathcal{D}'_i$ be neighboring datasets under the replace-one adjacency of Definition 3.2, differing in exactly one record. Because each per-example gradient is clipped to have Euclidean norm at most $C_g$, the change in the summed gradient satisfies

$$\|S_{i,t}(\mathcal{D}_i) - S_{i,t}(\mathcal{D}'_i)\|_2 \leq 2C_g.$$

Since the released update is normalized by the dataset size $|D_i|$, the $\ell_2$-sensitivity of the released vector equals

$$\Delta_i = \frac{2C_g}{|D_i|}.$$

**Gaussian mechanism** Each client releases a privatized update obtained by adding Gaussian noise to the summed gradient prior to normalization:

$$g_{i,t} = \frac{1}{|D_i|}\left(S_{i,t} + E_{i,t}\right), \qquad E_{i,t} \sim \mathcal{N}\left(0, \frac{(C_g\sigma_g)^2}{n}I_d\right).$$

Equivalently, the released vector can be written as

$$g_{i,t} = \frac{1}{|D_i|}S_{i,t} + \zeta_{i,t}, \qquad \zeta_{i,t} \sim \mathcal{N}\left(0, \frac{(C_g\sigma_g)^2}{n|D_i|^2}I_d\right).$$

Thus the client update contains isotropic Gaussian noise with standard deviation

$$\frac{C_g\sigma_g}{\sqrt{n}\,|D_i|}.$$

**Composition across rounds** The above mechanism is applied independently at each training round. Let $(\varepsilon_t, \delta_t)$ denote the privacy guarantee of a single round. Applying the composition result of Theorem 4.37 yields the overall differential privacy guarantee for the full DP-FedSOFIM training procedure.

# G IID Experimental Results

Tables 8–9 report test accuracy trajectories under the IID data partitioning scheme for CIFAR-10 and PathMNIST respectively. In the IID setting, data is partitioned uniformly at random across clients, eliminating label heterogeneity. Hyperparameters were retuned independently for the IID setting using the same grid as the non-IID experiments.

The IID results are broadly consistent with the non-IID findings: DP-FedSOFIM maintains competitive or superior performance across all privacy regimes. Notably, DP-FedGD and DP-FedAvg produce near-identical results in both settings, confirming that with a single local iteration ($K = 1$), FedAvg reduces to FedGD. DP-FedFC exhibits higher variance under IID at tight privacy budgets ($\varepsilon = 0.5, 1$), likely due to sensitivity of the covariance preconditioning step to the Gaussian noise scale.

Table 8: Test accuracy (%) on CIFAR-10 under IID partitioning across federated rounds. Results shown at 10-round intervals (mean ± std over 3 seeds). Best result per privacy regime is in **bold**.

| Method | Federated Round | | | | | | |
|---|---|---|---|---|---|---|---|
| | 10 | 20 | 30 | 40 | 50 | 60 | 70 |
| $\varepsilon = 0.5$ | | | | | | | |
| DP-FedGD | 38.56±1.96 | 49.61±1.17 | 54.36±1.20 | 57.04±1.20 | 58.79±0.86 | 60.13±0.82 | 61.02±0.35 |
| DP-FedAvg | 38.56±1.96 | 49.61±1.17 | 54.36±1.20 | 57.04±1.20 | 58.79±0.86 | 60.13±0.82 | 61.02±0.35 |
| DP-FedFC | 23.90±5.11 | 36.66±2.97 | 46.10±0.69 | 51.71±0.67 | 55.21±0.79 | 56.76±0.85 | 58.27±0.88 |
| DP-SCAFFOLD | 33.06±2.58 | 43.07±1.62 | 47.70±2.11 | 49.65±2.11 | 51.18±1.11 | 51.96±1.19 | 53.22±0.43 |
| DP-FedAdam | 35.78±2.54 | 48.05±1.25 | 53.85±0.68 | 56.68±1.04 | 58.89±0.88 | 59.90±0.77 | **61.07±0.38** |
| DP-FedYogi | 30.69±3.48 | 44.11±1.78 | 50.97±1.09 | 55.21±0.93 | 57.78±0.72 | 59.21±0.75 | 60.46±0.54 |
| DP-FedSOFIM | **43.53±1.05** | **51.57±0.92** | **56.45±0.69** | **58.41±0.98** | **59.62±0.98** | **60.34±0.91** | 60.34±0.61 |
| $\varepsilon = 1$ | | | | | | | |
| DP-FedGD | 40.61±1.54 | 51.69±0.70 | 56.71±0.51 | 59.39±0.76 | 61.37±0.54 | 62.63±0.47 | 63.41±0.44 |
| DP-FedAvg | 40.61±1.54 | 51.69±0.70 | 56.71±0.51 | 59.39±0.76 | 61.37±0.54 | 62.63±0.47 | 63.41±0.44 |
| DP-FedFC | 23.22±5.51 | 35.48±3.85 | 46.44±1.21 | 53.32±0.97 | 57.49±1.31 | 59.80±1.05 | 61.37±0.69 |
| DP-SCAFFOLD | 45.61±1.87 | 53.17±0.52 | 56.78±0.46 | 57.54±0.54 | 57.76±1.31 | 58.79±0.95 | 58.89±0.43 |
| DP-FedAdam | 51.61±1.51 | **60.21±0.67** | 62.60±0.57 | 63.52±0.44 | 63.82±0.39 | 63.64±0.49 | 63.85±0.43 |
| DP-FedYogi | 43.82±7.72 | 55.46±2.39 | 59.29±1.00 | 60.31±0.43 | 60.13±0.82 | 59.80±1.18 | 59.18±1.13 |
| DP-FedSOFIM | **53.36±0.72** | 60.09±0.27 | **62.80±0.53** | **63.72±0.77** | **64.24±0.53** | **64.61±0.45** | **64.39±0.55** |
| $\varepsilon = 5$ | | | | | | | |
| DP-FedGD | 41.55±1.39 | 52.73±0.44 | 57.65±0.14 | 60.50±0.51 | 62.35±0.48 | 63.44±0.48 | 64.30±0.51 |
| DP-FedAvg | 41.55±1.39 | 52.73±0.44 | 57.65±0.14 | 60.50±0.51 | 62.35±0.48 | 63.44±0.48 | 64.30±0.51 |
| DP-FedFC | 53.98±1.17 | 62.39±0.52 | 65.13±0.50 | 66.31±0.27 | 66.93±0.26 | 67.35±0.23 | 67.47±0.04 |
| DP-SCAFFOLD | 58.04±0.21 | 63.18±0.77 | 64.92±0.50 | 65.83±0.34 | 66.30±0.18 | 66.69±0.17 | 67.16±0.26 |
| DP-FedAdam | 42.03±5.69 | 58.58±2.51 | 63.04±0.85 | 66.43±0.78 | 67.28±0.27 | 68.02±0.06 | 67.94±0.15 |
| DP-FedYogi | 53.62±1.41 | 64.10±0.13 | 66.56±0.42 | 67.41±0.31 | 67.96±0.14 | 68.19±0.19 | **68.39±0.13** |
| DP-FedSOFIM | **61.49±0.73** | **65.82±0.29** | **67.46±0.09** | **67.70±0.14** | **68.01±0.11** | **68.28±0.11** | 68.11±0.08 |
| $\varepsilon = 10$ | | | | | | | |
| DP-FedGD | 41.69±1.42 | 52.77±0.45 | 57.75±0.15 | 60.61±0.51 | 62.34±0.47 | 63.52±0.52 | 64.29±0.50 |
| DP-FedAvg | 41.69±1.42 | 52.77±0.45 | 57.75±0.15 | 60.61±0.51 | 62.34±0.47 | 63.52±0.52 | 64.29±0.50 |
| DP-FedFC | 23.02±5.11 | 36.41±3.12 | 47.30±0.99 | 53.66±1.34 | 57.57±1.04 | 59.89±0.73 | 61.50±0.79 |
| DP-SCAFFOLD | 59.05±0.28 | 64.05±0.69 | 65.81±0.33 | 66.63±0.25 | 67.09±0.31 | 67.46±0.31 | 67.84±0.24 |
| DP-FedAdam | 47.14±2.41 | 58.58±1.23 | 64.37±1.26 | 66.01±0.48 | 67.27±0.47 | 67.88±0.12 | 68.05±0.09 |
| DP-FedYogi | 45.48±2.66 | 57.83±1.56 | 63.90±1.45 | 65.80±0.56 | 67.37±0.47 | 67.62±0.18 | 68.12±0.40 |
| DP-FedSOFIM | **61.68±0.76** | **66.13±0.36** | **67.73±0.13** | **68.26±0.17** | **68.56±0.04** | **68.87±0.11** | **69.00±0.20** |

Table 9: Test accuracy (%) on PathMNIST under IID partitioning across federated rounds. Results shown at 10-round intervals (mean ± std over 3 seeds). Best result per privacy regime is in **bold**.

| Method | Federated Round | | | | | | |
|---|---|---|---|---|---|---|---|
| | 10 | 20 | 30 | 40 | 50 | 60 | 70 |
| $\varepsilon = 0.5$ | | | | | | | |
| DP-FedGD | 51.90±0.68 | 57.77±0.77 | 59.87±1.38 | 62.21±0.59 | 63.31±0.78 | 64.18±0.96 | 64.33±0.74 |
| DP-FedAvg | 51.90±0.68 | 57.77±0.77 | 59.87±1.38 | 62.21±0.59 | 63.31±0.78 | 64.18±0.96 | 64.33±0.74 |
| DP-FedFC | 24.73±5.66 | 47.06±6.77 | 55.71±4.63 | 60.99±1.90 | 62.92±0.43 | 64.26±0.45 | 65.33±0.43 |
| DP-SCAFFOLD | 50.51±0.39 | 56.79±1.46 | 57.73±2.24 | 60.72±0.22 | 61.73±0.67 | 62.05±1.00 | 62.12±0.59 |

*Continued from previous page*

| Method | Federated Round | | | | | | |
|---|---|---|---|---|---|---|---|
| | 10 | 20 | 30 | 40 | 50 | 60 | 70 |
| DP-FedAdam | 50.61±3.57 | 57.98±1.04 | 61.26±0.71 | 63.17±0.23 | 64.34±0.71 | 64.81±0.85 | 65.63±0.67 |
| DP-FedYogi | 43.56±4.08 | 56.38±0.98 | 58.77±1.22 | 61.62±0.81 | 63.10±0.86 | 64.18±0.83 | 64.76±0.89 |
| DP-FedSOFIM | **54.49±2.68** | **60.34±2.30** | **63.85±0.84** | **64.86±1.73** | **66.44±0.70** | **66.38±1.14** | **67.44±0.52** |
| $\varepsilon = 1$ | | | | | | | |
| DP-FedGD | 52.36±1.19 | 58.05±0.64 | 60.39±1.13 | 62.49±0.93 | 63.64±0.98 | 64.77±0.91 | 65.14±0.58 |
| DP-FedAvg | 52.36±1.19 | 58.05±0.64 | 60.39±1.13 | 62.49±0.93 | 63.64±0.98 | 64.77±0.91 | 65.14±0.58 |
| DP-FedFC | 20.16±4.96 | 43.98±7.91 | 53.34±5.86 | 60.10±2.91 | 63.67±0.86 | 65.74±0.17 | 67.16±0.40 |
| DP-SCAFFOLD | 57.00±0.97 | 60.50±1.02 | 63.86±0.82 | 64.97±0.73 | 66.19±0.60 | 65.81±0.60 | 66.54±0.29 |
| DP-FedAdam | 58.73±2.14 | **64.81±0.60** | **66.83±0.66** | **68.04±0.68** | **68.39±0.61** | **68.41±0.31** | 68.91±0.42 |
| DP-FedYogi | 40.51±2.53 | 53.85±7.85 | 63.61±2.99 | 66.45±1.08 | 66.72±0.06 | 67.57±0.41 | 67.61±0.24 |
| DP-FedSOFIM | **58.76±0.27** | 64.67±0.47 | 66.40±0.46 | 67.31±0.60 | 67.69±0.48 | 68.07±0.61 | **68.95±0.15** |
| $\varepsilon = 5$ | | | | | | | |
| DP-FedGD | 52.59±1.63 | 58.21±0.87 | 60.60±1.26 | 62.45±1.26 | 63.92±1.09 | 64.93±0.91 | 65.73±0.63 |
| DP-FedAvg | 52.59±1.63 | 58.21±0.87 | 60.60±1.26 | 62.45±1.26 | 63.92±1.09 | 64.93±0.91 | 65.73±0.63 |
| DP-FedFC | 59.55±0.71 | 64.68±1.15 | 67.17±0.62 | 68.50±0.27 | 69.21±0.37 | 69.69±0.38 | 70.03±0.34 |
| DP-SCAFFOLD | 61.15±1.27 | 65.32±0.55 | 67.34±0.29 | 68.39±0.05 | 69.04±0.14 | 69.50±0.26 | 69.81±0.40 |
| DP-FedAdam | 48.69±4.47 | 61.69±3.13 | 66.88±1.89 | 68.25±0.39 | 70.26±0.64 | 70.67±0.12 | 71.56±0.19 |
| DP-FedYogi | 56.46±1.59 | 64.47±1.57 | 67.22±0.20 | 69.05±0.20 | 70.06±0.50 | 70.36±0.24 | 70.76±0.29 |
| DP-FedSOFIM | **62.76±2.37** | **67.25±2.01** | **69.31±0.83** | **69.68±0.31** | **70.34±0.69** | **71.30±0.15** | **71.65±0.14** |
| $\varepsilon = 10$ | | | | | | | |
| DP-FedGD | 52.57±1.62 | 58.20±0.92 | 60.61±1.31 | 62.43±1.32 | 63.89±1.08 | 64.98±0.90 | 65.80±0.68 |
| DP-FedAvg | 52.57±1.62 | 58.20±0.92 | 60.61±1.31 | 62.43±1.32 | 63.89±1.08 | 64.98±0.90 | 65.80±0.68 |
| DP-FedFC | 23.79±4.87 | 46.25±6.80 | 54.31±3.53 | 59.38±0.71 | 61.62±1.57 | 62.93±1.69 | 64.03±1.45 |
| DP-SCAFFOLD | 61.51±1.28 | 65.73±0.61 | 67.60±0.40 | 68.52±0.16 | 69.32±0.31 | 69.61±0.33 | 69.84±0.36 |
| DP-FedAdam | 57.62±3.14 | 62.45±5.07 | 67.95±0.44 | 66.90±0.88 | 69.32±0.78 | 69.80±0.49 | 70.73±0.21 |
| DP-FedYogi | 57.99±2.34 | 64.96±2.97 | 65.62±1.39 | 67.16±0.97 | 70.17±0.43 | 69.67±0.69 | 70.23±0.48 |
| DP-FedSOFIM | **62.86±2.29** | **67.20±2.09** | **69.31±0.75** | **69.90±0.29** | **70.48±0.73** | **71.59±0.11** | **71.75±0.16** |

## H  Ablation Study and Hyperparameter Sensitivity

### H.1  Component Ablation

Tables 10–11 and Figures 7–8 report the contribution of each component of DP-FedSOFIM under the non-IID (Dirichlet $\alpha = 0.5$) setting on CIFAR-10 and PathMNIST respectively, with 20 clients. We compare three variants: (i) *Grad-only*, which applies no EMA and no Sherman–Morrison (SM) preconditioning, reducing to standard DP-FedGD; (ii) *EMA-only*, which applies exponential moving average accumulation of gradients but omits the SM curvature correction; and (iii) *DP-FedSOFIM (full)*, the complete method.

The benefit of each component depends jointly on the privacy budget and the curvature structure of the dataset. At $\varepsilon = 0.5$, per-round Gaussian noise dominates any directional curvature signal: all three variants perform comparably on CIFAR-10, and on PathMNIST the EMA component provides only a small lift. The SM step is most beneficial at $\varepsilon = 1$, where noise is high enough to corrupt isotropic gradient steps yet low enough for the momentum buffer to accumulate a stable curvature signal; the full method outperforms EMA-only by approximately 7 percentage points on both datasets (Tables 10–11). At $\varepsilon \geq 5$, the picture diverges by dataset: on CIFAR-10 the diffuse curvature landscape (Phenomenon 5) means EMA-smoothed gradient directions already span the informative subspace, so EMA-only largely matches or marginally surpasses the full method (67.97% vs. 67.32% at $\varepsilon = 5$; 59.04% vs. 58.43% at $\varepsilon = 0.5$; Table 10), whereas on

Figure 5: Convergence trajectories on CIFAR-10, 20 clients, IID across privacy regimes.

PathMNIST the concentrated curvature structure continues to reward explicit preconditioning and the full method retains a consistent advantage across all budgets (Table 11).

In summary, the SM curvature correction is most effective when (i) privacy noise is moderate to high and (ii) the loss landscape has structured anisotropy that the rank-one proxy can capture. It provides limited benefit when noise is so large that curvature estimation is unreliable ($\varepsilon = 0.5$), or when the landscape is diffuse enough that EMA smoothing alone covers the informative gradient subspace (CIFAR-10, $\varepsilon \geq 5$).

Table 10: Ablation study on CIFAR-10, 20 clients, Non-IID (Dirichlet $\alpha = 0.5$). Final test accuracy (%) at round 70 (mean $\pm$ std over 3 seeds). Best result per privacy regime is in **bold**.

| Method | $\varepsilon = 0.5$ | $\varepsilon = 1$ | $\varepsilon = 5$ | $\varepsilon = 10$ |
|---|---|---|---|---|
| Grad-only (no EMA, no SM) | 58.72$\pm$1.02 | 52.46$\pm$2.35 | 57.17$\pm$0.67 | 54.73$\pm$1.64 |
| EMA-only (no SM) | **59.04$\pm$0.78** | 54.03$\pm$1.91 | **67.97$\pm$0.15** | 68.45$\pm$0.02 |
| DP-FedSOFIM (full) | 58.43$\pm$0.46 | **61.35$\pm$0.94** | 67.32$\pm$0.19 | **68.60$\pm$0.09** |

Table 11: Ablation study on PathMNIST, 20 clients, Non-IID (Dirichlet $\alpha = 0.5$). Final test accuracy (%) at round 70 (mean $\pm$ std over 3 seeds). Best result per privacy regime is in **bold**.

| Method | $\varepsilon = 0.5$ | $\varepsilon = 1$ | $\varepsilon = 5$ | $\varepsilon = 10$ |
|---|---|---|---|---|
| Grad-only (no EMA, no SM) | 64.41$\pm$0.99 | 60.25$\pm$2.66 | 55.07$\pm$1.96 | 53.33$\pm$3.97 |
| EMA-only (no SM) | **65.66$\pm$1.14** | 61.67$\pm$2.82 | 70.77$\pm$0.09 | 70.94$\pm$0.08 |
| DP-FedSOFIM (full) | 65.24$\pm$1.40 | **68.48$\pm$0.63** | **71.32$\pm$0.20** | **71.63$\pm$0.20** |

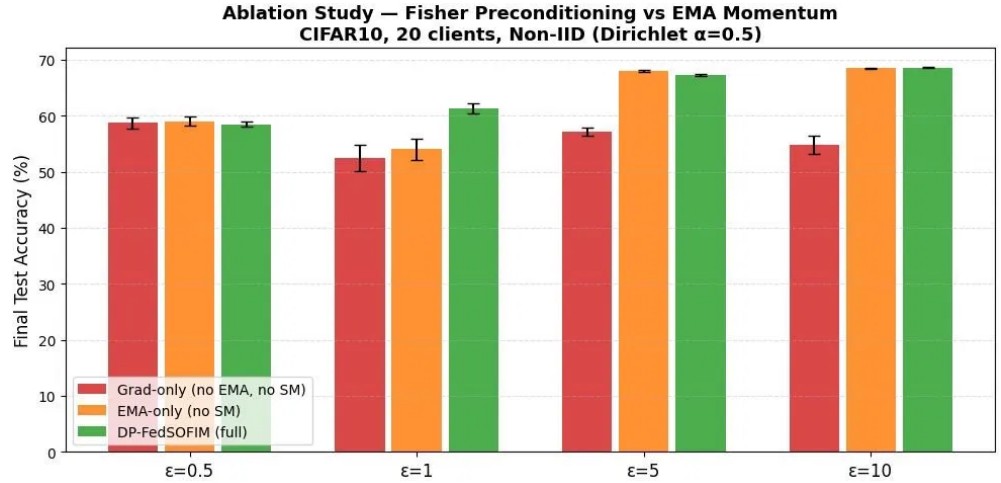

Figure 6: Convergence trajectories on PathMNIST, 20 clients, IID across privacy regimes.

Figure 7: Ablation study — Fisher preconditioning vs. EMA momentum on CIFAR-10, 20 clients, Non-IID (Dirichlet $\alpha = 0.5$). Final test accuracy at round 70 across privacy regimes.

## H.2   Sensitivity to $\rho$

Table 12 report final test accuracy at round 70 as a function of the EMA decay parameter $\rho \in \{0.01, 0.1, 0.5, 1.0, 2.0, 5.0, 10.0\}$ on CIFAR-10 and PathMNIST respectively, 20 clients, Non-IID (Dirichlet $\alpha = 0.5$), at $\varepsilon \in \{1, 5\}$.

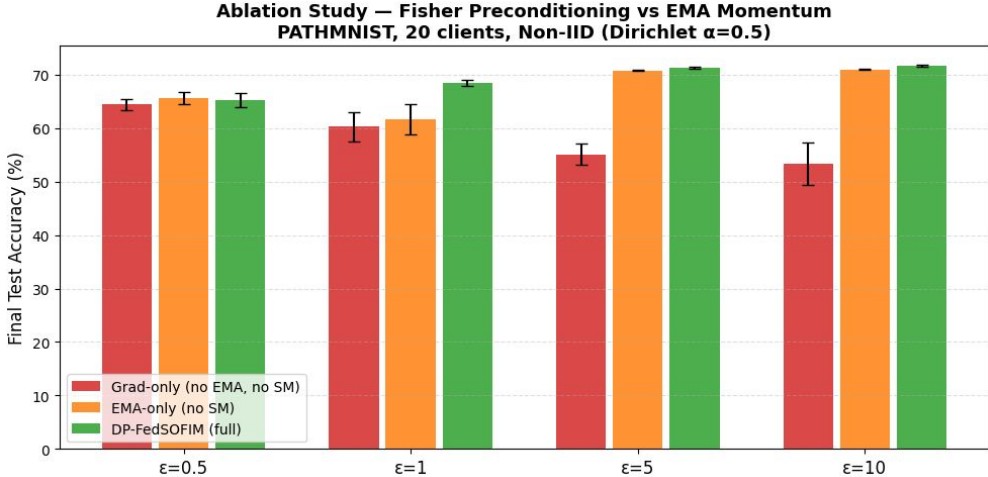

Figure 8: Ablation study — Fisher preconditioning vs. EMA momentum on PathMNIST, 20 clients, Non-IID (Dirichlet $\alpha = 0.5$). Final test accuracy at round 70 across privacy regimes.

On CIFAR-10, performance peaks near $\rho = 1.0$ at $\varepsilon = 5$ and increases monotonically with $\rho$ at $\varepsilon = 1$, where higher momentum better suppresses the large Gaussian noise. On PathMNIST, the optimal $\rho$ shifts to 0.5 at $\varepsilon = 5$ and again to 10.0 at $\varepsilon = 1$, consistent with the noise-dominance interpretation. In both cases the method is robust across the range $\rho \in [0.5, 5.0]$ at $\varepsilon = 5$, with degradation only at the extremes. We select $\rho = 1.0$ as the default for $\varepsilon \geq 5$ and $\rho = 10.0$ for tight privacy budgets.

Table 12: $\rho$ sensitivity of DP-FedSOFIM. Final test accuracy (%) at round 70 (mean $\pm$ std over 3 seeds), Non-IID (Dirichlet $\alpha = 0.5$), $n = 20$ clients. Best result per $\varepsilon$ is in **bold**.

| | CIFAR-10 | | PathMNIST | |
|---|---|---|---|---|
| $\rho$ | $\varepsilon = 1$ | $\varepsilon = 5$ | $\varepsilon = 1$ | $\varepsilon = 5$ |
| 0.01 | 51.08±1.39 | 59.37±1.72 | 54.37±3.60 | 49.69±5.28 |
| 0.1 | 51.36±1.35 | 63.02±0.43 | 57.75±2.63 | 63.74±5.81 |
| 0.5 | 52.18±1.47 | 67.32±0.19 | 61.13±2.41 | **71.32±0.20** |
| 1.0 | 53.11±1.39 | **68.03±0.05** | 61.49±1.71 | 70.92±0.13 |
| 2.0 | 54.57±1.39 | 67.55±0.22 | 62.32±0.50 | 69.83±0.22 |
| 5.0 | 57.94±1.17 | 65.56±0.53 | 66.92±1.43 | 67.71±0.14 |
| 10.0 | **61.35±0.94** | 61.99±0.62 | **68.48±0.63** | 64.94±0.05 |

## I  Non-IID Partition Statistics

Tables 13 and 14 report the per-client, per-class sample counts for PathMNIST and CIFAR-10 respectively, generated with a fixed seed under the Dirichlet ($\alpha = 0.5$) partition used in all non-IID experiments. These statistics confirm the moderate label heterogeneity described in Section 5.1: clients are rarely missing classes entirely (mean 0.1 absent classes per client on PathMNIST), but a single class typically contributes 40–70% of a client's local data.

## J  McNemar's Test for Statistical Significance

**Setup.**  We conduct McNemar's test (McNemar, 1947; Dietterich, 1998) comparing DP-FedSOFIM against each baseline across all four experimental settings at every privacy budget $\varepsilon \in \{0.5, 1, 5, 10\}$. McNemar's test is a paired non-parametric test operating on the per-sample predictions of two classifiers evaluated on

Table 13: Per-client, per-class sample counts for **PathMNIST** (89,996 training samples, 9 classes, $n = 20$ clients, Dirichlet $\alpha = 0.5$, fixed seed). Class labels: $0 =$ adipose, $1 =$ background, $2 =$ debris, $3 =$ lymphocytes, $4 =$ mucus, $5 =$ smooth muscle, $6 =$ normal colon mucosa, $7 =$ cancer-assoc. stroma, $8 =$ adenocarcinoma epithelium. Entries of 0 indicate a completely absent class.

| Client | cls 0 | cls 1 | cls 2 | cls 3 | cls 4 | cls 5 | cls 6 | cls 7 | cls 8 | Total |
|---|---|---|---|---|---|---|---|---|---|---|
| 0 | 8 | 259 | 41 | 7 | 89 | 2 689 | 694 | 2 985 | 1 840 | 8 612 |
| 1 | 288 | 95 | 148 | 629 | 492 | 6 | 78 | 74 | 908 | 2 718 |
| 2 | 12 | 29 | 776 | 498 | 114 | 1 732 | 8 | 851 | 1 887 | 5 907 |
| 3 | 13 | 578 | 194 | 63 | 147 | 119 | 447 | 202 | 326 | 2 089 |
| 4 | 1 498 | 280 | 934 | 1 088 | 1 | 1 345 | 1 682 | 248 | 299 | 7 375 |
| 5 | 79 | 160 | 53 | 87 | 445 | 21 | 214 | 90 | 75 | 1 224 |
| 6 | 187 | 142 | 36 | 977 | 55 | 7 | 739 | 754 | 233 | 3 130 |
| 7 | 506 | 1 193 | 324 | 2 038 | 3 | 69 | 13 | 433 | 76 | 4 655 |
| 8 | 0 | 5 | 149 | 174 | 3 046 | 0 | 97 | 463 | 543 | 4 477 |
| 9 | 20 | 101 | 400 | 936 | 241 | 11 | 873 | 265 | 403 | 3 250 |
| 10 | 2 052 | 148 | 130 | 319 | 745 | 59 | 1 247 | 49 | 32 | 4 781 |
| 11 | 9 | 1 781 | 2 556 | 426 | 1 211 | 149 | 76 | 587 | 36 | 6 831 |
| 12 | 261 | 844 | 1 499 | 438 | 26 | 13 | 28 | 23 | 351 | 3 483 |
| 13 | 496 | 627 | 2 424 | 245 | 26 | 3 035 | 118 | 102 | 852 | 7 925 |
| 14 | 317 | 242 | 147 | 1 207 | 262 | 807 | 14 | 586 | 0 | 3 582 |
| 15 | 525 | 62 | 1 | 264 | 61 | 26 | 30 | 7 | 69 | 1 045 |
| 16 | 130 | 550 | 13 | 17 | 804 | 378 | 44 | 246 | 401 | 2 583 |
| 17 | 40 | 37 | 23 | 170 | 113 | 430 | 62 | 3 | 178 | 1 056 |
| 18 | 127 | 1 878 | 511 | 46 | 19 | 1 263 | 1 391 | 426 | 4 325 | 9 986 |
| 19 | 2 798 | 498 | 1 | 772 | 106 | 23 | 31 | 1 007 | 51 | 5 287 |
| **Total** | 9 366 | 9 509 | 10 360 | 10 201 | 8 105 | 14 181 | 8 836 | 11 401 | 14 885 | 96 844[†] |

[†] The column total of 96,844 reflects samples assigned to the federated training split; 89,996 samples are used in training after excluding held-out validation samples. Summary statistics: min $= 1{,}045$ (client 15), max $= 9{,}986$ (client 18), mean $= 4{,}500$, std $= 2{,}526$. Mean KL divergence from the uniform class distribution: 0.581 (max 1.115; where $0 =$ perfectly IID). Mean absent classes per client: 0.1; maximum: 2 (clients 8 and 14).

the *same N* test samples within a single model run. For each pair (DP-FedSOFIM, baseline) we record $n_{10}$ (samples correct for SOFIM, wrong for baseline) and $n_{01}$ (the converse). The continuity-corrected statistic

$$\chi^2 = \frac{(|n_{10} - n_{01}| - 1)^2}{n_{10} + n_{01}}$$

is compared against $\chi_1^2$ at $\alpha = 0.05$. A single seed is used by design; pooling across seeds breaks the paired structure. With $N = 10{,}000$ (CIFAR-10) and $N = 7{,}180$ (PathMNIST) statistical power is high. Tests are at the final round (Round 70).

**Discussion.** DP-FedSOFIM is statistically significantly better than all baselines at $\varepsilon \in \{1, 5, 10\}$ on CIFAR-10/VGG, and at $\varepsilon \in \{1, 10\}$ on CIFAR-10/ResNet. On PathMNIST with both architectures, significance is established at $\varepsilon = 10$ against all baselines, and largely at $\varepsilon \in \{0.5, 1\}$ as well.

**Non-significant cases at $\varepsilon = 5$.** On CIFAR-10/ResNet, differences against DP-FedAdam, DP-FedYogi, DP-FedFC, and DP-AdaFedProx are not statistically significant at Round 70. On PathMNIST/ResNet, DP-FedAdam, DP-FedYogi, and DP-FedFC are likewise non-significant; on PathMNIST/VGG, DP-FedAdam is non-significant. These cases reflect convergence behaviour rather than an absence of advantage: SOFIM reaches high accuracy substantially faster in earlier rounds. For instance, at $\varepsilon = 5$ on CIFAR-10/ResNet, SOFIM achieves 61.05% accuracy at Round 10 versus 43.92% for DP-FedAdam and 55.37% for DP-FedYogi, with parity only around Round 50–60. In communication-constrained federated learning, this convergence-speed advantage is a practically important property not captured by final-round McNemar tests.

**High-noise regime ($\varepsilon = 0.5$).** At $\varepsilon = 0.5$, the advantage of DP-FedSOFIM does not hold uniformly on CIFAR-10: several adaptive baselines achieve statistically significant differences on CIFAR-10/ResNet, and performance is statistically comparable across most methods on CIFAR-10/VGG. This is consistent with the known sensitivity of second-order methods to gradient noise — at $\varepsilon = 0.5$, the per-example DP noise dominates the Fisher information estimate, diminishing the preconditioner's effectiveness. On PathMNIST,

Table 14: Per-client, per-class sample counts for **CIFAR-10** (50,000 training samples, 10 classes, $n = 20$ clients, Dirichlet $\alpha = 0.5$, fixed seed). Class labels: $0 =$ airplane, $1 =$ automobile, $2 =$ bird, $3 =$ cat, $4 =$ deer, $5 =$ dog, $6 =$ frog, $7 =$ horse, $8 =$ ship, $9 =$ truck. Entries of 0 indicate a completely absent class.

| Client | cls 0 | cls 1 | cls 2 | cls 3 | cls 4 | cls 5 | cls 6 | cls 7 | cls 8 | cls 9 | Total |
|---|---|---|---|---|---|---|---|---|---|---|---|
| 0 | 14 | 122 | 240 | 126 | 630 | 74 | 193 | 71 | 230 | 6 | 1 706 |
| 1 | 122 | 37 | 1 488 | 88 | 12 | 43 | 0 | 123 | 43 | 419 | 2 375 |
| 2 | 17 | 113 | 97 | 122 | 647 | 1 030 | 609 | 822 | 27 | 14 | 3 498 |
| 3 | 467 | 1 253 | 2 | 151 | 7 | 149 | 20 | 130 | 66 | 367 | 2 612 |
| 4 | 794 | 1 | 117 | 0 | 40 | 0 | 5 | 83 | 86 | 180 | 1 306 |
| 5 | 38 | 386 | 1 | 967 | 777 | 134 | 436 | 44 | 16 | 216 | 3 015 |
| 6 | 29 | 729 | 335 | 95 | 126 | 1 200 | 5 | 9 | 45 | 783 | 3 356 |
| 7 | 32 | 106 | 266 | 19 | 226 | 472 | 101 | 29 | 90 | 174 | 1 515 |
| 8 | 161 | 619 | 20 | 585 | 46 | 66 | 55 | 45 | 222 | 120 | 1 939 |
| 9 | 173 | 72 | 524 | 102 | 225 | 326 | 68 | 32 | 256 | 766 | 2 544 |
| 10 | 33 | 175 | 929 | 508 | 477 | 35 | 116 | 39 | 37 | 187 | 2 536 |
| 11 | 138 | 877 | 1 | 105 | 62 | 2 | 1 019 | 0 | 1 191 | 1 | 3 396 |
| 12 | 45 | 3 | 1 | 751 | 138 | 104 | 7 | 100 | 951 | 1 212 | 3 312 |
| 13 | 43 | 83 | 186 | 669 | 14 | 3 | 54 | 1 333 | 373 | 168 | 2 926 |
| 14 | 192 | 4 | 51 | 90 | 303 | 83 | 744 | 1 241 | 588 | 33 | 3 329 |
| 15 | 44 | 16 | 64 | 311 | 1 047 | 25 | 727 | 44 | 494 | 208 | 2 980 |
| 16 | 331 | 163 | 119 | 3 | 9 | 92 | 22 | 67 | 10 | 75 | 891 |
| 17 | 7 | 92 | 12 | 95 | 11 | 746 | 91 | 413 | 26 | 46 | 1 539 |
| 18 | 449 | 0 | 104 | 192 | 93 | 0 | 723 | 204 | 246 | 14 | 2 025 |
| 19 | 1 871 | 149 | 443 | 21 | 110 | 416 | 5 | 171 | 3 | 11 | 3 200 |
| **Total** | 5 000 | 5 000 | 5 000 | 5 000 | 5 000 | 5 000 | 5 000 | 5 000 | 5 000 | 5 000 | 50 000 |

Summary statistics: $\min = 891$ (client 16), $\max = 3{,}498$ (client 2), mean $= 2{,}500$, std $= 779.6$. Mean KL divergence from the uniform class distribution: 0.666 (max 1.039; where $0 =$ perfectly IID). Mean absent classes per client: 0.3; maximum: 2 (client 4 missing classes 3 and 5; client 18 missing classes 1 and 5).

SOFIM remains significantly better than most baselines at $\varepsilon = 0.5$ across both architectures, suggesting that the dataset's class-structure regularity makes curvature information more recoverable under noise. The $\varepsilon = 0.5$ regime on CIFAR-10 represents a boundary condition for preconditioner-based approaches under extreme privacy budgets.

**Isolated non-significant cases.** DP-FedFC achieves statistically comparable final-round accuracy to SOFIM on PathMNIST/ResNet at $\varepsilon \in \{0.5, 1, 5\}$, with SOFIM numerically higher in each case. DP-AdaFedProx is comparable to SOFIM on PathMNIST/VGG at $\varepsilon = 1$ (0.6124 vs. 0.6116), and DP-FedYogi is comparable on PathMNIST/VGG at $\varepsilon = 10$ (0.6649 vs. 0.6627); in these two cases the baseline is marginally higher but the difference does not reach significance. **Result legend:** $\uparrow^*$ SOFIM significantly better; $\dagger$ significant difference, baseline numerically higher; $-$ not significant.

Table 15: McNemar's test: DP-FedSOFIM vs. baselines — CIFAR-10. Left: ResNet-20. Right: VGG-16.

| Baseline | Acc | $n_{10}$ | $n_{01}$ | $\chi^2$ | $p$ | Res. |
|---|---|---|---|---|---|---|
| $\varepsilon = 0.5$ *SOFIM acc = 0.5691, N = 10,000* | | | | | | |
| DP-FedGD | 0.5862 | 529 | 700 | 23.515 | <0.0001 | † |
| DP-FedAvg | 0.5862 | 529 | 700 | 23.515 | <0.0001 | † |
| DP-FedAdam | 0.5856 | 488 | 653 | 23.572 | <0.0001 | † |
| DP-FedYogi | 0.5846 | 568 | 723 | 18.370 | <0.0001 | † |
| DP-FedFC | 0.5381 | 914 | 604 | 62.899 | <0.0001 | ↑* |
| DP-SCAFFOLD | 0.4789 | 1231 | 329 | 520.385 | <0.0001 | ↑* |
| DP-FTRL | 0.5605 | 1258 | 1172 | 2.973 | 0.0847 | − |
| DP-AdaFedProx | 0.5862 | 529 | 700 | 23.515 | <0.0001 | † |
| $\varepsilon = 1$ *SOFIM acc = 0.6206* | | | | | | |
| DP-FedGD | 0.6254 | 499 | 547 | 2.112 | 0.1462 | − |
| DP-FedAvg | 0.6254 | 499 | 547 | 2.112 | 0.1462 | − |
| DP-FedAdam | 0.6149 | 252 | 195 | 7.016 | 0.0081 | ↑* |
| DP-FedYogi | 0.5562 | 912 | 268 | 350.381 | <0.0001 | ↑* |
| DP-FedFC | 0.5851 | 882 | 527 | 88.940 | <0.0001 | ↑* |
| DP-SCAFFOLD | 0.5663 | 1419 | 876 | 128.002 | <0.0001 | ↑* |
| DP-FTRL | 0.5908 | 1103 | 805 | 46.231 | <0.0001 | ↑* |
| DP-AdaFedProx | 0.5916 | 1273 | 983 | 37.022 | <0.0001 | ↑* |
| $\varepsilon = 5$ *SOFIM acc = 0.6731* | | | | | | |
| DP-FedGD | 0.6385 | 951 | 605 | 76.494 | <0.0001 | ↑* |
| DP-FedAvg | 0.6385 | 951 | 605 | 76.494 | <0.0001 | ↑* |
| DP-FedAdam | 0.6714 | 161 | 144 | 0.839 | 0.3596 | − |
| DP-FedYogi | 0.6772 | 213 | 254 | 3.426 | 0.0642 | − |
| DP-FedFC | 0.6711 | 474 | 454 | 0.389 | 0.5328 | − |
| DP-SCAFFOLD | 0.6644 | 737 | 650 | 5.332 | 0.0209 | ↑* |
| DP-FTRL | 0.6146 | 1248 | 663 | 178.470 | <0.0001 | ↑* |
| DP-AdaFedProx | 0.6729 | 575 | 573 | 0.001 | 0.9765 | − |
| $\varepsilon = 10$ *SOFIM acc = 0.6849* | | | | | | |
| DP-FedGD | 0.6389 | 967 | 507 | 142.931 | <0.0001 | ↑* |
| DP-FedAvg | 0.6389 | 967 | 507 | 142.931 | <0.0001 | ↑* |
| DP-FedAdam | 0.6772 | 323 | 246 | 10.151 | 0.0014 | ↑* |
| DP-FedYogi | 0.6795 | 323 | 269 | 4.745 | 0.0294 | ↑* |
| DP-FedFC | 0.6024 | 1416 | 591 | 338.304 | <0.0001 | ↑* |
| DP-SCAFFOLD | 0.6759 | 535 | 445 | 8.083 | 0.0045 | ↑* |
| DP-FTRL | 0.6304 | 1073 | 528 | 184.844 | <0.0001 | ↑* |
| DP-AdaFedProx | 0.6760 | 472 | 383 | 9.057 | 0.0026 | ↑* |

| Baseline | Acc | $n_{10}$ | $n_{01}$ | $\chi^2$ | $p$ | Res. |
|---|---|---|---|---|---|---|
| $\varepsilon = 0.5$ *SOFIM acc = 0.5195, N = 10,000* | | | | | | |
| DP-FedGD | 0.5220 | 553 | 578 | 0.509 | 0.4754 | − |
| DP-FedAvg | 0.5235 | 516 | 556 | 1.419 | 0.2336 | − |
| DP-FedAdam | 0.5220 | 521 | 546 | 0.540 | 0.4625 | − |
| DP-FedYogi | 0.5164 | 612 | 581 | 0.754 | 0.3851 | − |
| DP-FedFC | 0.4905 | 980 | 690 | 50.013 | <0.0001 | ↑* |
| DP-SCAFFOLD | 0.4583 | 1109 | 497 | 232.454 | <0.0001 | ↑* |
| DP-FTRL | 0.5362 | 643 | 810 | 18.965 | <0.0001 | † |
| DP-AdaFedProx | 0.5235 | 516 | 556 | 1.419 | 0.2336 | − |
| $\varepsilon = 1$ *SOFIM acc = 0.5463* | | | | | | |
| DP-FedGD | 0.5389 | 292 | 218 | 10.449 | 0.0012 | ↑* |
| DP-FedAvg | 0.5366 | 388 | 291 | 13.573 | 0.0002 | ↑* |
| DP-FedAdam | 0.5393 | 336 | 266 | 7.909 | 0.0049 | ↑* |
| DP-FedYogi | 0.4983 | 1092 | 612 | 134.648 | <0.0001 | ↑* |
| DP-FedFC | 0.5120 | 913 | 570 | 78.870 | <0.0001 | ↑* |
| DP-SCAFFOLD | 0.4823 | 1412 | 772 | 186.960 | <0.0001 | ↑* |
| DP-FTRL | 0.5397 | 557 | 491 | 4.031 | 0.0447 | ↑* |
| DP-AdaFedProx | 0.4987 | 1389 | 913 | 98.013 | <0.0001 | ↑* |
| $\varepsilon = 5$ *SOFIM acc = 0.5818* | | | | | | |
| DP-FedGD | 0.5477 | 856 | 515 | 84.318 | <0.0001 | ↑* |
| DP-FedAvg | 0.5477 | 856 | 515 | 84.318 | <0.0001 | ↑* |
| DP-FedAdam | 0.5493 | 681 | 356 | 101.230 | <0.0001 | ↑* |
| DP-FedYogi | 0.5659 | 569 | 410 | 25.499 | <0.0001 | ↑* |
| DP-FedFC | 0.5677 | 520 | 379 | 21.802 | <0.0001 | ↑* |
| DP-SCAFFOLD | 0.5633 | 687 | 502 | 28.474 | <0.0001 | ↑* |
| DP-FTRL | 0.5486 | 895 | 563 | 75.145 | <0.0001 | ↑* |
| DP-AdaFedProx | 0.5724 | 601 | 507 | 7.806 | 0.0052 | ↑* |
| $\varepsilon = 10$ *SOFIM acc = 0.5919* | | | | | | |
| DP-FedGD | 0.5490 | 848 | 419 | 144.581 | <0.0001 | ↑* |
| DP-FedAvg | 0.5490 | 848 | 419 | 144.581 | <0.0001 | ↑* |
| DP-FedAdam | 0.5789 | 603 | 473 | 15.466 | 0.0001 | ↑* |
| DP-FedYogi | 0.5714 | 664 | 459 | 37.058 | <0.0001 | ↑* |
| DP-FedFC | 0.5415 | 951 | 447 | 180.979 | <0.0001 | ↑* |
| DP-SCAFFOLD | 0.5711 | 547 | 339 | 48.362 | <0.0001 | ↑* |
| DP-FTRL | 0.5513 | 849 | 443 | 126.954 | <0.0001 | ↑* |
| DP-AdaFedProx | 0.5762 | 491 | 334 | 29.498 | <0.0001 | ↑* |

Table 16: McNemar's test: DP-FedSOFIM vs. baselines — PathMNIST. Left: ResNet-20. Right: VGG-16.

| Baseline | Acc | $n_{10}$ | $n_{01}$ | $\chi^2$ | $p$ | Res. |
|---|---|---|---|---|---|---|
| $\varepsilon = 0.5$ *SOFIM acc* = 0.6436, $N$ = 7,180 | | | | | | |
| DP-FedGD | 0.6308 | 424 | 332 | 10.954 | 0.0009 | $\uparrow^*$ |
| DP-FedAvg | 0.6308 | 424 | 332 | 10.954 | 0.0009 | $\uparrow^*$ |
| DP-FedAdam | 0.6294 | 364 | 262 | 16.296 | 0.0001 | $\uparrow^*$ |
| DP-FedYogi | 0.6237 | 409 | 266 | 29.873 | <0.0001 | $\uparrow^*$ |
| DP-FedFC | 0.6396 | 449 | 420 | 0.902 | 0.3422 | – |
| DP-SCAFFOLD | 0.5866 | 654 | 245 | 185.166 | <0.0001 | $\uparrow^*$ |
| DP-FTRL | 0.5996 | 778 | 462 | 80.020 | <0.0001 | $\uparrow^*$ |
| DP-AdaFedProx | 0.6308 | 424 | 332 | 10.954 | 0.0009 | $\uparrow^*$ |
| $\varepsilon = 1$ *SOFIM acc* = 0.6721 | | | | | | |
| DP-FedGD | 0.6545 | 332 | 205 | 29.564 | <0.0001 | $\uparrow^*$ |
| DP-FedAvg | 0.6545 | 332 | 205 | 29.564 | <0.0001 | $\uparrow^*$ |
| DP-FedAdam | 0.6650 | 171 | 120 | 8.591 | 0.0034 | $\uparrow^*$ |
| DP-FedYogi | 0.6256 | 535 | 201 | 150.664 | <0.0001 | $\uparrow^*$ |
| DP-FedFC | 0.6713 | 385 | 379 | 0.033 | 0.8565 | – |
| DP-SCAFFOLD | 0.6389 | 611 | 372 | 57.624 | <0.0001 | $\uparrow^*$ |
| DP-FTRL | 0.6294 | 581 | 274 | 109.516 | <0.0001 | $\uparrow^*$ |
| DP-AdaFedProx | 0.6550 | 518 | 395 | 16.302 | 0.0001 | $\uparrow^*$ |
| $\varepsilon = 5$ *SOFIM acc* = 0.7100 | | | | | | |
| DP-FedGD | 0.6671 | 576 | 268 | 111.669 | <0.0001 | $\uparrow^*$ |
| DP-FedAvg | 0.6671 | 576 | 268 | 111.669 | <0.0001 | $\uparrow^*$ |
| DP-FedAdam | 0.7063 | 170 | 143 | 2.160 | 0.1417 | – |
| DP-FedYogi | 0.7075 | 144 | 126 | 1.070 | 0.3009 | – |
| DP-FedFC | 0.7064 | 235 | 209 | 1.408 | 0.2354 | – |
| DP-SCAFFOLD | 0.7024 | 301 | 246 | 5.331 | 0.0210 | $\uparrow^*$ |
| DP-FTRL | 0.6538 | 690 | 286 | 166.403 | <0.0001 | $\uparrow^*$ |
| DP-AdaFedProx | 0.6951 | 358 | 251 | 18.450 | <0.0001 | $\uparrow^*$ |
| $\varepsilon = 10$ *SOFIM acc* = 0.7155 | | | | | | |
| DP-FedGD | 0.6691 | 581 | 248 | 132.960 | <0.0001 | $\uparrow^*$ |
| DP-FedAvg | 0.6691 | 581 | 248 | 132.960 | <0.0001 | $\uparrow^*$ |
| DP-FedAdam | 0.7084 | 207 | 156 | 6.887 | 0.0087 | $\uparrow^*$ |
| DP-FedYogi | 0.7081 | 207 | 154 | 7.490 | 0.0062 | $\uparrow^*$ |
| DP-FedFC | 0.6600 | 681 | 283 | 163.495 | <0.0001 | $\uparrow^*$ |
| DP-SCAFFOLD | 0.7052 | 266 | 192 | 11.635 | 0.0006 | $\uparrow^*$ |
| DP-FTRL | 0.6606 | 625 | 231 | 180.431 | <0.0001 | $\uparrow^*$ |
| DP-AdaFedProx | 0.6985 | 337 | 215 | 26.524 | <0.0001 | $\uparrow^*$ |

| Baseline | Acc | $n_{10}$ | $n_{01}$ | $\chi^2$ | $p$ | Res. |
|---|---|---|---|---|---|---|
| $\varepsilon = 0.5$ *SOFIM acc* = 0.6053, $N$ = 7,180 | | | | | | |
| DP-FedGD | 0.5751 | 444 | 227 | 69.532 | <0.0001 | $\uparrow^*$ |
| DP-FedAvg | 0.5783 | 410 | 216 | 59.503 | <0.0001 | $\uparrow^*$ |
| DP-FedAdam | 0.5787 | 414 | 223 | 56.672 | <0.0001 | $\uparrow^*$ |
| DP-FedYogi | 0.5777 | 436 | 238 | 57.580 | <0.0001 | $\uparrow^*$ |
| DP-FedFC | 0.5838 | 442 | 288 | 32.067 | <0.0001 | $\uparrow^*$ |
| DP-SCAFFOLD | 0.5581 | 530 | 191 | 158.452 | <0.0001 | $\uparrow^*$ |
| DP-FTRL | 0.5928 | 404 | 314 | 11.032 | 0.0009 | $\uparrow^*$ |
| DP-AdaFedProx | 0.5783 | 410 | 216 | 59.503 | <0.0001 | $\uparrow^*$ |
| $\varepsilon = 1$ *SOFIM acc* = 0.6116 | | | | | | |
| DP-FedGD | 0.6040 | 130 | 76 | 13.636 | 0.0002 | $\uparrow^*$ |
| DP-FedAvg | 0.5967 | 209 | 102 | 36.129 | <0.0001 | $\uparrow^*$ |
| DP-FedAdam | 0.6063 | 162 | 124 | 4.787 | 0.0287 | $\uparrow^*$ |
| DP-FedYogi | 0.5967 | 563 | 456 | 11.026 | 0.0009 | $\uparrow^*$ |
| DP-FedFC | 0.5568 | 611 | 218 | 185.361 | <0.0001 | $\uparrow^*$ |
| DP-SCAFFOLD | 0.5918 | 523 | 381 | 21.992 | <0.0001 | $\uparrow^*$ |
| DP-FTRL | 0.5939 | 277 | 150 | 37.180 | <0.0001 | $\uparrow^*$ |
| DP-AdaFedProx | 0.6124 | 456 | 462 | 0.027 | 0.8689 | – |
| $\varepsilon = 5$ *SOFIM acc* = 0.6578 | | | | | | |
| DP-FedGD | 0.6146 | 611 | 301 | 104.694 | <0.0001 | $\uparrow^*$ |
| DP-FedAvg | 0.6146 | 611 | 301 | 104.694 | <0.0001 | $\uparrow^*$ |
| DP-FedAdam | 0.6525 | 226 | 188 | 3.307 | 0.0690 | – |
| DP-FedYogi | 0.6384 | 369 | 230 | 31.793 | <0.0001 | $\uparrow^*$ |
| DP-FedFC | 0.6444 | 321 | 225 | 16.529 | <0.0001 | $\uparrow^*$ |
| DP-SCAFFOLD | 0.6341 | 438 | 268 | 40.455 | <0.0001 | $\uparrow^*$ |
| DP-FTRL | 0.6084 | 647 | 292 | 133.457 | <0.0001 | $\uparrow^*$ |
| DP-AdaFedProx | 0.6400 | 424 | 296 | 22.401 | <0.0001 | $\uparrow^*$ |
| $\varepsilon = 10$ *SOFIM acc* = 0.6627 | | | | | | |
| DP-FedGD | 0.6148 | 636 | 292 | 126.777 | <0.0001 | $\uparrow^*$ |
| DP-FedAvg | 0.6148 | 636 | 292 | 126.777 | <0.0001 | $\uparrow^*$ |
| DP-FedAdam | 0.6412 | 486 | 332 | 28.617 | <0.0001 | $\uparrow^*$ |
| DP-FedYogi | 0.6649 | 345 | 361 | 0.319 | 0.5724 | – |
| DP-FedFC | 0.6053 | 710 | 298 | 167.580 | <0.0001 | $\uparrow^*$ |
| DP-SCAFFOLD | 0.6418 | 361 | 211 | 38.813 | <0.0001 | $\uparrow^*$ |
| DP-FTRL | 0.6116 | 658 | 291 | 141.155 | <0.0001 | $\uparrow^*$ |
| DP-AdaFedProx | 0.6426 | 396 | 252 | 31.557 | <0.0001 | $\uparrow^*$ |

