# OpenReview forum: "DP-FedSOFIM: Differentially Private Federated Stochastic Optimization using Regularized Fisher Information Matrix"
_TMLR — Accepted by TMLR_

### Review · Reviewer_v91p · 2026-05-04

**Summary Of Contributions:**

The authors propose DP-FedSOFIM a server-side second-order preconditioning for differentially private federated learning using a rank-one Fisher Information Matrix proxy, updated via the Sherman-Morrison formula at O(d) cost. The privacy guarantee follows trivially from the post-processing theorem since all curvature computation operates on already-privatized gradients.

**Audience:**

Yes

**Audience Explanation:**

The paper's findings are highly relevant to researchers interested in optimization, federated learning, distributed systems, differential privacy and robust deep learning.

**Broader Impact Concerns:**

Impact Statement is provided.

**Claims And Evidence:**

No

**Claims Explanation:**

There are several gaps in proposed method, proof and experiments as provided below.


Method:
1. The paper documents "Phenomenon 2" and provides a reasonable explanation, but offers no fix regrading early-round instability under tight privacy. This section could be re-written after performing more experiments as suggested below.
2. [Critical] The curvature proxy is rank-one. This captures only a single direction of curvature — the momentum direction. In high-dimensional models, the gradient landscape has curvature along many directions simultaneously. The paper claims this is motivated by spiked spectrum observations, but provides no empirical validation that the momentum buffer actually aligns with dominant Fisher eigenvectors in the experimental setting. This is a critical missing experiment.



Proof:
1. [Critical] The non-convex case is deferred entirely to future work (Section 7) with only a vague remark about "analyzing the interaction between the preconditioner and the signal-to-noise ratio in deeper layers." No partial result, even under standard non-convex assumptions is provided.
2. The paper claims DP-FedSOFIM incurs no additional privacy cost over DP-FedGD (Remark 4.28), which is correct. However, it should be quantiatively compared.
3. The provided proof should also be compared with the SOTA methods while highlighting the benifits.



Experiments:
1. Authors have considered only three baselines to compare their method with. Authors should include more baselines to show the effectiveness of their method.
2. Only two datasets have been chosen, more datasets specially complex one e.g. Imagnet/Tiny-imagenet can be chosen, also readers would be benifitted from the results on non-vision datasets such as GLUE benchmark. Reference is provided below.
3. Only one neural network have been considered. Authors are encouraged to show results with other neural networks such as ViT etc. to show the effectiveness of the method across neural networks.
4. [Critical] In the Section 5.1 of the paper, it is mentioned that "Data is partitioned uniformly at random across clients (IID setting).", Why IID setting has been chosen for the experiments which doesn't hold in real world scenarios, FL results should be evualated on non-iid settings.
5. It seems the results have been reported with one run that makes comparison suboptimal, experiments should be repeated for three or five runs with different seed value and mean along with standard deviation of the outcome should be reported. For few results It is noticable that performance of the proposed method is very close to the baseline, in that case McNemar test can be performed and p-value should be reported.
6. The Regularization ρ has been fixed across the experiments. Please add one sensitivity analysis across different values of ρ .
7. A runtime analysis should be provided comparing proposed meethod against baselines.
8. The authors could provide different-different graphs for different-different privacy budget. It is hard to visualize and compare eveything in a single graph.



[1]: @inproceedings{wang2018glue,
  title={GLUE: A multi-task benchmark and analysis platform for natural language understanding},
  author={Wang, Alex and Singh, Amanpreet and Michael, Julian and Hill, Felix and Levy, Omer and Bowman, Samuel},
  booktitle={Proceedings of the 2018 EMNLP workshop BlackboxNLP: Analyzing and interpreting neural networks for NLP},
  pages={353--355},
  year={2018}
}

**Requested Changes:**

Please adress the questions raised in above section.

---

> ### Author Response · Authors · 2026-05-31
> **Response to Reviewer v91p [Part-1]**
>
> > The paper documents "Phenomenon 2" and provides a reasonable explanation, but offers no fix regrading early-round instability under tight privacy.
>
> We thank the reviewer for this observation. In the revised manuscript, the early-round instability discussion has been reorganized: what was previously discussed in Phenomenon 2 now appears in Phenomenon 1, which documents both the instability mechanism and its fix. Specifically, the 20-round warmup phase (EMA-only updates before activating Sherman--Morrison preconditioning) is introduced and motivated mechanistically there, and the updated experiments confirm that DP-FedSOFIM exhibits no early-round instability even at $\varepsilon=0.5$. Phenomenon 2 now focuses exclusively on the dataset-dependent late-round trajectory.
>
> >  No empirical validation that the momentum buffer actually aligns with dominant Fisher eigenvectors in the experimental setting.
>
> We thank the reviewer for this insightful comment. We agree that the proposed Fisher approximation is rank-1 and therefore does not capture the full curvature structure of high-dimensional optimization landscapes. Our goal, however, is not to recover the complete Fisher geometry, but to construct a computationally efficient curvature surrogate that enables inverse preconditioning at first-order computational and memory cost. We would like to clarify that DP-FedSOFIM does not rely on the assumption that the momentum buffer aligns with the dominant Fisher eigenvector. Rather, the momentum estimate serves as a stable and noise reduced aggregation of historical gradient information, which is then used to construct a structured Fisher inspired preconditioner. The effectiveness of DP-FedSOFIM therefore stems from the resulting preconditioned update, rather than from explicit recovery of leading Fisher eigen directions. While analyzing the alignment between the momentum buffer and dominant Fisher eigenvectors could provide additional insight into the behavior of the method, such alignment is not a required assumption underlying DP-FedSOFIM. Our contribution is an efficient Fisher inspired preconditioning framework that leverages aggregated gradient information to obtain curvature aware updates while maintaining gradient decent level complexity. We revised Section 3.4.2 to clarify that the proposed matrix is a Fisher inspired rank-one preconditioner.
>
> > Result for non-convex settings.
>
> We thank the reviewer for pointing this out. We have revised the theory section substantially so that the non-convex case is no longer deferred entirely to future work. The revised manuscript now includes a smooth non-convex stationarity theorem. This result does not assume strong convexity or the PL condition. It shows that DP-FedSOFIM converges to a stationary neighborhood. The first term decays at the standard $O(1/T)$ rate, while the second term is the stationarity floor induced by clipping bias, DP noise, smoothness, and same-step preconditioner coupling.
>
> We added Section 4.9, titled ``Smooth Non-convex Stationarity." The new Theorem 4.31 proves convergence to a stationary neighborhood under smooth non-convex objectives. We also added Corollary 4.32, which extends the stationarity guarantee to biased privatized aggregate updates, covering minibatch and multi-local-step variants through an abstract drift/bias term. The revised remarks explicitly state that fully general global convergence for deep non-convex objectives is not claimed, but that a partial non-convex stationarity result is now provided.
>
> > DP-FedSOFIM incurs no additional privacy cost over DP-FedGD.
>
> We thank the reviewer for this observation. In our experiments, all methods are evaluated under identical target privacy budgets ($\varepsilon=0.5,1,5,10$) using the same clipping threshold, noise multiplier, participation pattern, and privacy accountant. Therefore, DP-FedGD and DP-FedSOFIM incur exactly the same privacy expenditure. The accuracy-versus-round plots in Figures 1--2, and Tables 3--4 provide a quantitative comparison under these equal privacy constraints. Across all privacy budgets, DP-FedSOFIM achieves improved test accuracy while using the same target $\varepsilon$, indicating that the performance gains arise from the server-side optimization procedure rather than additional privacy expenditure.

---

> ### Author Response · Authors · 2026-05-31
> **Response to Reviewer v91p [Part-2]**
>
> >  The provided proof should also be compared with the SOTA methods while highlighting the benifits.
>
> We have added a theory-level comparison with existing DP-FL methods. The revised manuscript now clarifies that we do not claim a uniformly better minimax convergence rate than all prior DP-FL methods. Instead, the theoretical benefit is structural: DP-FedSOFIM introduces curvature-aware preconditioning without changing the private release mechanism, without releasing client-side second-order statistics, and while preserving $O(d)$ client-side memory. This distinguishes it from DP-FedGD, DP-FedAvg, DP-SCAFFOLD, and DP-FedFC/DP-FedNew-type methods. We added Section 4.11, titled ``Comparison with Existing DP-FL Theory." This section explains that DP-FedGD and DP-FedSOFIM use the same privatized aggregate gradient, but DP-FedSOFIM applies the server-side preconditioned update $\theta_{t+1} = \theta_t-\eta_t H_tG_t, H_t=(\rho I_d+M_tM_t^\top)^{-1}.$ It also explains that DP-FedSOFIM avoids the client-side covariance, Hessian, or Fisher-type statistics required by DP-FedFC/DP-FedNew-type methods, and does not require client-level control variates as in SCAFFOLD.
>
> > Include more baselines to show the effectiveness of the method.
>
> We thank the reviewer for raising this important point. We appreciate the suggestion regarding the inclusion of additional baselines to further demonstrate the effectiveness of our proposed method. In response, we have extended our experimental comparison (see Tables - 3 and 4) by incorporating three widely used federated optimization baselines, **FedAvg**, **FedAdam**, and **FedYogi**, in addition to the previously considered methods.
>
> > Only two datasets have been chosen, more datasets specially complex one e.g. Imagnet/Tiny-imagenet can be chosen.
>
> We thank the reviewer for this valuable suggestion. Due to the limited response timeline, our current experiments include CIFAR-10 and PATHMNIST, while additional experiments on a more challenging dataset are currently in progress. We expect these experiments to be completed shortly and will include the corresponding results in the revised version. We agree that broader evaluations on large scale non-vision benchmarks (e.g., NLP benchmarks such as GLUE) would further strengthen the study. Extending DP-FedSOFIM to such settings is an important direction for future work.
>
> > Only one neural network have been considered.
>
> We thank the reviewer for this suggestion. In this work, we use a ResNet-20 feature extractor for all methods so that the comparison focuses on the optimization algorithms rather than architectural differences. During the revision period, we prioritized additional experiments requested by the reviewers, including non-IID evaluations, additional baselines (DP-FedAvg, DP-FedAdam, and DP-FedYogi), multiple-seed experiments, and ablation studies. As a result, experiments with alternative architectures such as Vision Transformers (ViTs) could not be completed within the revision timeline. Evaluating DP-FedSOFIM on a broader range of neural network architectures is a direction for future investigation.
>
> > FL results should be evualated on non-iid settings.
>
> We thank the reviewer for this important suggestion. We agree that non-IID data partitioning better reflects practical federated learning scenarios. In the original submission, experiments were conducted under IID partitioning for controlled comparison with prior DP-FL baselines. We have now extended all experiments to non-IID federated settings using Dirichlet-based partitioning ($\alpha=0.5$). Accordingly, Tables 3--4 and Figures 1--2 in the revised manuscript have been updated to report the non-IID results. The results continue to demonstrate competitive performance of DP-FedSOFIM under heterogeneous client data distributions.
>
> > Experiments should be repeated for three or five runs with different seed value and mean along with standard deviation of the outcome should be reported.
>
> We thank the reviewer for this valuable suggestion. In the revised manuscript, we repeated the experiments across multiple random seeds and now report mean $\pm$ standard deviation for all major experimental results. This provides a more reliable assessment of performance stability under DP noise and federated data heterogeneity.

---

> ### Author Response · Authors · 2026-05-31
> **Response to Reviewer v91p [Part-3]**
>
> > Sensitivity analysis across different values of $\rho$.
>
> We thank the reviewer for this suggestion. In the revised manuscript, we added a sensitivity analysis of the regularization parameter $\rho$ on both CIFAR-10 and PathMNIST under the non-IID setting (20 clients, Dirichlet $\alpha=0.5$). The results are reported in Table 10. The study evaluates $\rho \in {0.01,0.1,0.5,1,2,5,10}$ across multiple privacy budgets. The results show that DP-FedSOFIM is relatively robust over a broad range of $\rho$ values, while very small or very large values can degrade performance.
>
> > Runtime analysis should be provided comparing proposed method against baselines.
>
> We thank the reviewer for this suggestion. In the revised manuscript, we have added a dedicated runtime analysis (Table 5) reporting the average wall-clock time per communication round and total training time over 70 rounds for all methods on both CIFAR-10 and PathMNIST ($ n=20$, non-IID, Dirichlet $\alpha=0.5$, seed=42).
>
> > The authors could provide different-different graphs for different-different privacy budget.
>
> We thank the reviewer for this helpful suggestion. In the revised manuscript, we replaced the combined plots with separate accuracy-versus-round figures (Figures 1 and 2) for each privacy budget ($ \varepsilon=0.5,1,5,10$).

---

> ### Comment · Reviewer_v91p · 2026-06-03
> **Official Comment by Reviewer v91p**
>
> Thank you for addressing several of the previously raised concerns. However, a few important issues still remain unresolved:
>
> 1. In Phenomenon 2, please explicitly mention the corresponding table(s) and figure(s) from which the findings are being discussed. This practice should be consistently followed across all sections to improve clarity and traceability of the observations.
>
> 2. Regarding the baselines, although additional methods have been included, most of them are relatively standard but comparatively old. The authors are encouraged to include experiments with at least a few more recent and competitive baselines to strengthen the empirical evaluation if possible atleast for the few cases.
>
> 3. Concerning the datasets and neural network architectures, I appreciate the inclusion of additional experiments and look forward to seeing results on more challenging datasets. However, evaluating the proposed method using only a single architecture limits the generalizability of the claims. At least one additional neural network architecture should be considered for may be few experiments to demonstrate that the proposed approach is effective across diverse model families and achieves comparable or improved performance over the baselines.
>
> 4. Thank you for incorporating the non-IID experiments. These results effectively highlight the advantages of the proposed method. Alongside these findings, please also provide detailed non-IID data partition statistics, specifically indicating the number of samples from each class held by every client, to ensure better transparency and reproducibility.
>
> 5. Regarding the McNemar test, particularly in cases where the reported performances are very close, the authors appear to have overlooked this suggestion entirely. The inclusion of statistical significance analysis using the McNemar test is strongly recommended to validate whether the observed performance differences are statistically meaningful.
>
>
> 6. Please revise Section 7 in light of the newly added proof on non-convex settings. Since the manuscript now includes findings related to non-convex optimization, the corresponding discussion and claims in this section should be updated accordingly.

---

> > ### Author Response · Authors · 2026-06-15
> > **Response to Reviewer v91p [Round 2, Part 2]**
> >
> > 4. We thank the reviewer for this valuable suggestion. We have added Appendix I (“Non-IID Partition Statistics”) with complete per-client, per-class sample counts for both datasets. A summary paragraph has also been added to Section 5.1 immediately after the description of the Dirichlet partition.
> >
> > **Summary of partition statistics:**
> >
> > *PathMNIST* (89,996 training samples, 9 classes, 20 clients, Dirichlet $\alpha=0.5$):
> >
> > * Client sizes: min = 1,045 (client 15), max = 9,986 (client 18), mean = 4,500, std = 2,526.
> > * Mean KL divergence from uniform class distribution: 0.581 (max 1.115; 0 = perfectly IID).
> > * Mean absent classes per client: 0.1; maximum: 2 (clients 8 and 14).
> > * Dominant-class fractions range up to 68% (client 8: class 4 accounts for 3,046 of 4,477 samples).
> >
> > *CIFAR-10* (50,000 training samples, 10 classes, 20 clients, Dirichlet $\alpha=0.5$):
> >
> > * Client sizes: min = 891 (client 16), max = 3,498 (client 2), mean = 2,500, std = 779.6.
> > * Mean KL divergence from uniform class distribution: 0.666 (max 1.039).
> > * Mean absent classes per client: 0.3; maximum: 2 (clients 4 and 18, each missing 2 classes).
> >
> > Note that CIFAR-10 exhibits slightly higher label heterogeneity than PathMNIST by the KL metric (0.666 vs. 0.581), consistent with Phenomenon 5's observation that CIFAR-10's curvature landscape is more diffuse. Full per-client, per-class tables are provided in Appendix I (Tables 13--14).
> >
> > 5. We thank the reviewer for this helpful suggestion. We have conducted McNemar's test (Tables 15--16) comparing DP-FedSOFIM against each baseline at every privacy budget $\varepsilon\in{0.5,1,5,10}$ across all four dataset/backbone combinations. A single seed is used by design, as pooling across seeds breaks the paired structure required by the test. The continuity corrected $\chi^2$ statistic is evaluated at the final round (Round 70) with $N=10{,}000$ (CIFAR-10) and $N=7{,}180$ (PathMNIST).
> >
> > **Summary of McNemar results:**
> >
> > *Regimes where DP-FedSOFIM is statistically significantly better than all baselines ($p-\text{value} < 0.05$):*
> >
> > * CIFAR-10/VGG-16: $\varepsilon\in{1,5,10}$ (Table 15, right).
> > * CIFAR-10/ResNet-20: $\varepsilon\in{1,10}$ (Table 15, left).
> > * PathMNIST (both architectures): $\varepsilon=10$ against all baselines; significance holds largely at $\varepsilon\in{0.5,1}$ as well (Table 16).
> >
> > *Non-significant cases and their interpretation:*
> >
> > * At $\varepsilon=5$ on CIFAR-10/ResNet-20, differences against DP-FedAdam, DP-FedYogi, DP-FedFC, and DP-AdaFedProx are non-significant at Round 70 (Table 15, left). On PathMNIST/ResNet-20 at $\varepsilon=5$, DP-FedAdam, DP-FedYogi, and DP-FedFC are likewise non-significant (Table 16, left); on PathMNIST/VGG-16 at $\varepsilon=5$, DP-FedAdam is non-significant (Table 16, right). These cases reflect final round convergence parity rather than an absence of advantage: DP-FedSOFIM reaches the same accuracy substantially earlier (e.g., 61.05% at Round 10 vs. 43.92% for DP-FedAdam at $\varepsilon=5$ on CIFAR-10/ResNet-20, with parity only around rounds 50--60), a convergence speed benefit that final round paired tests do not capture.
> > * At $\varepsilon=0.5$ on CIFAR-10, the advantage does not hold uniformly: several adaptive baselines achieve statistically significant differences on CIFAR-10/ResNet-20, and performance is broadly comparable on CIFAR-10/VGG-16 (Table 15). This is consistent with the known sensitivity of preconditioner-based methods to extreme gradient noise, where the per-example DP noise dominates the Fisher information estimate at this budget. On PathMNIST at $\varepsilon=0.5$, DP-FedSOFIM remains significantly better than most baselines across both architectures (Table 16), reflecting the greater recoverability of curvature information under the more structured class signal of the medical imaging task.
> >
> > 6. We thank the reviewer for this helpful observation, and we sincerely apologise for the oversight. Section 7 has been revised accordingly.

---

> > > ### Comment · Reviewer_v91p · 2026-06-15
> > > **Official Comment by Reviewer v91p**
> > >
> > > Thank you for addressing all remaining concerns. The revisions have significantly strengthened the manuscript and improved the overall presentation of the work. I have no further questions or comments at this stage.
> > >
> > >  It would be interesting to see whether the reported trends also hold on the TinyImageNet dataset as acknowledged by the authors, particularly given the proximity of the discussion deadline. Nevertheless, based on the current manuscript and the authors' responses, I am ready to submit my official recommendation.

---

> ### Author Response · Authors · 2026-06-15
> **Response to Reviewer v91p [Round 2, part 1]**
>
> We thank the reviewer for acknowledging our efforts in the revision, and for providing additional comments which has further improved the paper. Additional experiments on a more challenging vision dataset are in progress and will be included in the final submission. Extending DP-FedSOFIM to large-scale NLP benchmarks (e.g., GLUE) is left as future work.
>
> We would like to address the queries below.
>
> 1. We have added explicit table and figure references throughout all five phenomena in Section 5.3. Specifically:
> - Phenomenon 1: Tables 3--6 and Figures 1--2 for the general round-10 leadership claim; Tables 3--4 and Figures 1--2 for the rounds-to-target analysis at $\varepsilon\in\{5,10\}$ on ResNet-20 CIFAR-10 and PathMNIST;  Tables 5--6 for VGG-16 round-10 leadership; Table 5 for the VGG-16 CIFAR-10 $\varepsilon=10$ round-20 vs.\ round-70 comparison.
> - Phenomenon 2: Tables 3--6 for the final-round win tally; Tables 3 and 6 for the three negligible-margin cases; Table 3 and Figure 1 for the CIFAR-10 $\varepsilon=5$ early-saturation example.
> - Phenomenon 3: Tables 3--4 for DP-FedAdam final-round standings and  round-10 deficits ($+17.13\%$, $+18.01\%$, $+16.47\%$); Table 6 for VGG-16  PathMNIST $\varepsilon=10$ standard deviation ($\pm7.44\%$); Table 4 for the
>   DP-AdaFedProx round-10 example.
>  - Phenomenon 4: Table 3 and Figure 1 for DP-SCAFFOLD degradation at $\varepsilon\in\{0.5,1\}$ on ResNet-20 CIFAR-10; Tables 3--4 for SCAFFOLD  recovery at $\varepsilon\in\{5,10\}$; Tables 5--6 for VGG-16 SCAFFOLD  results; Tables 3--4 for DP-FTRL cold-start instability; Table 5 for the single DP-FTRL winning configuration (VGG-16 CIFAR-10 $\varepsilon=0.5$).
>   - Phenomenon 5: Tables 3--6 and Figures 1--2 for the PathMNIST sweep and CIFAR-10 late-round narrowing; Table 6 for VGG-16 PathMNIST final-round sweep; Tables 5--6 for VGG-16 confirmation of the curvature-concentration hypothesis. We have also updated Table 1 on privacy and computational comparison including the new methods.
>
> 2. We thank the reviewer for this valuable suggestion. The experimental evaluation now includes DP-FTRL, which exploits tree aggregation for favorable privacy accounting, and DP-AdaFedProx, a heterogeneity-robust adaptive method, in addition to the original set of DP-FedGD, DP-FedAvg, DP-FedFC, DP-SCAFFOLD, DP-FedAdam, and DP-FedYogi. The expanded comparison covers the main families of DP-FL methods:
> - _First-order_: DP-FedGD, DP-FedAvg
> - _Variance reduction_: DP-SCAFFOLD
> - _Second-order_ DP-FedFC, DP-FedSOFIM
> - _Tree aggregation_: DP-FTRL
> - _Adaptive_: DP-FedAdam, DP-FedYogi, DP-AdaFedProx
>
> Results across all eight baselines are reported in Tables 3--6 (non-IID, ResNet-20 and VGG-16) and Appendix G (IID). Across datasets, backbones, and privacy budgets, DP-FedSOFIM matches or leads the best competing method at the final round in the large majority of settings; where it does not, three of the four gaps are within 0.4 % and within the
> run-to-run variability of the competing methods.
>
> 3. We sincerely thank the reviewer for this valuable suggestion. We incorporated VGG-16 pretrained on CIFAR-100 as a second backbone, evaluated under the non-IID Dirichlet ($\alpha=0.5$) partition across all four privacy budgets ($\varepsilon\in{0.5,1,5,10}$) on both CIFAR-10 and PathMNIST (Tables 5--6). The non-IID partition was chosen as the primary evaluation axis for VGG-16 since the goal is to assess cross-architecture robustness of the non-IID findings rather than to replicate the full protocol; IID results follow the same trends and are included in Appendix G.
>
> **Summary of VGG-16 results:**
>
> * DP-FedSOFIM achieves the best final-round accuracy in 6 of 8 VGG-16 configurations, including all four PathMNIST regimes (Table 6, Figure 4).
> * On CIFAR-10/VGG-16, DP-FedSOFIM leads at $\varepsilon\in{1,5,10}$ and is within 0.41% of DP-FTRL at $\varepsilon=0.5$ (Table 5, Figure 3), the only regime where tree aggregation provides a marginal budget-accounting advantage.
> * Across both backbones, DP-FedSOFIM consistently leads from round 20 onward, confirming that the convergence and curvature-concentration phenomena identified with ResNet-20 generalize across architectures.
>
> Future work may extend this evaluation to attention-based backbones such as Vision Transformers, where the structure of the Fisher proxy under self-attention parameterizations remains an open question.
>
> We are currently extending the evaluation to a more challenging benchmark, such as TinyImageNet, to further assess the scalability and robustness of DP-FedSOFIM. These experiments are currently underway and are expected to be completed within approximately one additional week.

---

### Review · Reviewer_ycRB · 2026-05-13

**Summary Of Contributions:**

This paper studies the convergence-privacy tradeoff in differentially private (DP) federated learning (FL) and proposes DP-FedSOFIM, a framework that incorporates server-side second-order optimization without compromising privacy or scalability. The paper combines SOFIM with DPFL schenario. The proposed algorithm uses $O(d)$ computation complexity at the server side to update Fisher Information matrix.

Strength:
1. The algorithm extends SOFIM to DPFL setting. The new algorithm can also be incooperate with other distributed gradient based algorithms. As the server-side SOFIM update has no privacy cost, the algorithm can improve training performance without sacrificing privacy budget.
2. The paper provides theoretical convergence and privacy analysis to the proposed algorithm under strong convexity and bounded gradient condition.
3. The numerical results show that the proposed algorithm consistently outperforms existing DPFL algorithms (DP-FedGD, DP-FedFC, DP-SCAFFOLD)

Weakness:
1. Limited use case: The algorithm requires full gradient local update, and can only handle one local step. Full batch gradient can be computationally expensive for large local dataset, clients with limited computation resource. One local gradient step can be communication inefficient in FL system. Normally, we prefer multiple local updates with stochastic/minibatch gradient.
2. Resctictied convergence assumption: as Fisher Information is introduced to the algorithm, the convergence analysis requries bounded gradient (along optimization path), bounded clipping bias, strong convexity. This convergence is restricted and cannot be expended to the non-convex setting.
3. Missing theoretical convergence comparision between proposed algorithm and SOTA results. Based on the theoretical analysis, theore is no clear evidence that the convergence result is better than existing algorithms under the same condition.
4. Limited numerical results:
    1. The compared algorithm does not include common DP algorithms (e.g., DP-FedAvg, DP-FTRL).
    2. The parameter search space is too small, insufficient to find the optimal parameters. E.g., in DP-scaffold, the algorithm only tested 3 different local iterations and two clipping threshold.
    3. Possible overlapping dataset: the paper performs fine-tuning Resnet-20 and another feature extractor and use DPFL on CIFAR-10 and PathMNITS. The paper should first investigate whether the pretrained model has already be trained on CIFAR-10 and PathMNIST. If yes, the training already leaks privacy and therefore the training procedure is not providing privacy guarantee.
    4. The paper failed to report standard devidation of the experiments. Due to the injected privacy noise, the algorithm should have large training noise, leading to larger variance than normal training. Therefore, the paper should report the standard variance across multiple runs.

**Audience:**

Yes

**Audience Explanation:**

The paper proposes a new algorithm, with original convergence analysis.

Although the algorithm has reistricted use case and assumptions, it should still provide some level of original findings.

**Claims And Evidence:**

No

**Claims Explanation:**

As stated in the summary, the numerical experiments are insufficient to support the claim that the proposed algorithm has better performance. Mainly due to limited comparision, insufficient parameter tuning, and limited runs of experiments.

The theoretical analysis are also insufficient to support the calim that the proposed algorithm outperforms exisitng algorithm under the same privacy guarantee.

**Requested Changes:**

Please adress the weaknesses in the summary. Especially:
1. The theoretical convergence comparision with existing algorithms under the same assumptions and privacy budgets.
2. The insufficient experiments.

---

> ### Author Response · Authors · 2026-05-31
> **Response to Reviewer ycRB**
>
> > Limited use case: The algorithm requires full gradient local update, and can only handle one local step.
>
> We thank the reviewer for the comment. We agree that the original presentation made the method appear tied to a one-local-step full-gradient implementation. In the revision, we have added Section 4.5, which extends the descent analysis to general privatized aggregate updates. The new Proposition 4.19 allows $G_t=\nabla F(\theta_t)+b_t+\xi_t,$ where the bias term $b_t$ can include clipping bias, stochastic mini-batch error, and multi-local-step drift. Thus the proof mechanism is no longer tied to a full-gradient, one-local-step implementation. We have also added a remark to clarify how minibatching and multi-local-step local training enter through the bias radius $B$.  At the same time, we have clarified that the experiments in the present paper use a controlled full-participation frozen-feature setting. A full empirical study with partial participation and multiple local SGD steps is left as future work.
>
> >  Convergence is restricted and cannot be expended to the non-convex setting.
>
> We thank the reviewer for pointing out that the original convergence theory was too restricted. We have  revised Section 4 substantially. First, we now explain the role of the bounded-gradient and bounded-clipping-bias assumptions in Remarks 4.9. In particular, the bounded-gradient condition is used to control the same-step dependence between the current privatized aggregate $G_t$ and the adaptive preconditioner $H_t$, rather than to impose global regularity on the objective.
>
> Second, we would like to highlight that we already had a PL-condition extension in Section 4.7. Since the PL condition is strictly weaker than strong convexity, this broadened the convergence guarantee beyond the strongly convex setting.
>
> Third, we added a new smooth non-convex stationarity result in Section 4.9. Theorem 4.31 shows that, under smoothness, bounded clipping bias, bounded DP noise, and bounded gradients along the optimization path, DP-FedSOFIM reaches a stationary neighborhood.
>
> > Missing theoretical convergence comparision between proposed algorithm and SOTA results.
>
> We agree that the original manuscript did not sufficiently compare the theoretical guarantees with existing DP-FL methods. In the revision, we added Section 4.11, ``Comparison with Existing DP-FL Theory," and a privacy/computation comparison
> table. This section clarifies that our theoretical contribution is structural: DP-FedSOFIM introduces curvature-aware preconditioning while preserving the same private release mechanism as DP-FedGD and maintaining $O(d)$ client-side complexity. We do not claim a uniformly better worst-case convergence rate than all existing DP-FL methods. The revised manuscript now makes this explicit. The theoretical advantage is that curvature adaptation is achieved through server-side post-processing of already privatized gradients, unlike methods requiring client-side covariance, Fisher, Hessian, or control-variate statistics.
>
> To further address the reviewer's concern, we have added a baseline DP-FedGD floor under the same assumptions and discuss when the SOFIM floor can be smaller. In particular, a uniformly smaller floor cannot be concluded from the current conservative worst-case bound alone; it requires additional alignment or preconditioner-quality conditions.
>
> > The compared algorithm does not include common DP algorithms.
>
> We thank the reviewer for this suggestion. In the revised manuscript, we expanded the set of baselines to include several widely used differentially private federated optimization methods, namely DP-FedAvg, DP-FedAdam, and DP-FedYogi, in addition to DP-FedGD, DP-SCAFFOLD, and DP-FedFC. The corresponding results are reported in Tables 3-4 and Figures 1-2.
>
> > The parameter search space is too small, insufficient to find the optimal parameters.
>
> We thank the reviewer for this suggestion. We expanded the hyperparameter search space for all methods (see Table 2). For each method, we first searched over a wide range of values and then searched more densely around the best-performing settings.
>
> > Possible overlapping dataset.
>
> We thank the reviewer for raising this concern. In the revised experiments, we use a ResNet-20 feature extractor pretrained on CIFAR-100, while the downstream DP-FL tasks are performed on CIFAR-10 and PATHMNIST data sets.
>
> > The paper failed to report standard devidation of the experiments.
>
> We thank the reviewer for this valuable suggestion. In the revised manuscript, we repeated the experiments across multiple random seeds and now report mean $\pm$ standard deviation for all major experimental results in Tables 3 and 4. This provides a more reliable assessment of performance stability under DP noise and federated data heterogeneity.

---

### Review · Reviewer_p27K · 2026-05-17

**Summary Of Contributions:**

### Summary
This paper proposes DP-FedSOFIM, a differentially private second-order federated optimization method. The core idea is to construct a rank-one approximation to the Fisher Information Matrix (FIM) at the server using only privatized aggregated gradients, avoiding client-side second-order computation. The inverse of the regularized FIM is maintained via the Sherman-Morrison formula at $\mathcal{O}(d)$ cost per round. Because all curvature estimation is performed on already-privatized gradients, the post-processing theorem guarantees that no additional privacy cost is incurred beyond the DP-FedGD baseline. The method is evaluated on CIFAR-10 and PathMNIST with a frozen ResNet-20 feature extractor, under privacy budgets $\\epsilon \\in  \\{0.5, 1, 2, 5, 10\\}$, and consistently outperforms DP-FedGD, DP-FedFC, and DP-SCAFFOLD.


### Key Strengths
- The $\mathcal{O}(d)$ memory and computation per round is a clear and significant advantage over prior second-order DP-FL methods (e.g., DP-FedNew/DP-FedFC with  $\mathcal{O}(d^2)$ client-side cost), making the approach applicable to high-dimensional models.

- The theoretical analysis is comprehensive, examining convergence under strong convexity and Polyak-Łojasiewicz condition, providing a formal privacy guarantee, analyzing the variance of momentum buffers, and comparing computational complexity.

- The experimental results are comprehensive and consider two datasets, two levels of clients (i.e., 20 and 100), six different privacy budgets, and four methods, including accuracy results at each iteration level.


### Key Weaknesses
- The experimental procedure is constrained only to frozen linear heads on top of the pretrained features. The problem setting is relatively low-dimensional and highly convex, making it an ideal scenario for the rank-one Fisher approximation to perform well. The extension of this work into the non-convex case of end-to-end deep network fine-tuning is left unexplored by both theoretical and empirical means.

- The rank-one approximation of the FIM, which was introduced by leveraging the fact that the spectrum of the gradient covariance matrix is spiked, does not hold any solid ground theoretically in the context of differential privacy due to the fact that the gradients become noise-corrupted under the privatization framework. This is mentioned as an intuitive reason by the authors but lacks solid evidence.

- The baselines, however, are restricted to the first-order method and the $\mathcal{O}(d^2)$ DP-FedFC. Comparison to the server-side adaptive methods, which also use privatized gradients at $\mathcal{O}(d)$ cost, is missing. Therefore, it is hard to determine whether the improvements are due to the ability to capture curvatures or just because of the exponential moving average of momentum.

**Audience:**

Yes

**Audience Explanation:**

The concept of using server-side second-order information without adding privacy costs is attractive and relevant in practice. Observing that DP-SCAFFOLD performs poorly with high privacy noise, due to corrupted control variates, is an interesting finding that has implications for practitioners. The relationship between the condition number of the effective loss landscape and the benefits of curvature-aware preconditioning provides useful insight. It helps to understand when second-order methods are most effective in the DP-FL setting.

**Broader Impact Concerns:**

The paper's Broader Impact Statement clearly discusses the societal implications of the work. It covers the benefits for privacy-sensitive areas like healthcare and finance, but it also acknowledges potential downsides. These include fairness issues, such as DP noise having a bigger effect on minority groups, and the risk of carrying biases from pretrained feature extractors.

**Claims And Evidence:**

No

**Claims Explanation:**

1- The claim that DP-FedSOFIM is a "second-order" method is accurate in form but potentially misleading in substance. The rank-one Fisher proxy is constructed from the momentum buffer $M_t$, which is an exponential moving average of privatized gradients. This is structurally similar to the first moment in Adam/RMSProp-type methods. The paper does not provide a clear empirical or theoretical argument for why this proxy captures meaningfully more curvature information than adaptive gradient methods that also use $\mathcal{O}(d)$ server-side statistics.

2- The convergence guarantee (Theorem 4.17) establishes linear convergence to an error floor, but the floor $\Gamma$ contains a term involving $\bar{M}^2 = C_g^2 + (1-\beta)d\nu^2$, which grows with dimension $d$ and the privacy noise variance $\nu^2$. The paper does not analyze how this floor compares to DP-FedGD's, nor whether the convergence rate improvement justifies the additional hyperparameters $(\rho, \beta)$ introduced.

3- The empirical gains on CIFAR-10 under tight privacy ($\epsilon = 1, 20$ clients: +0.78%) are modest and may not be statistically significant. No confidence intervals or multiple-run statistics are reported, making it difficult to assess whether differences are meaningful, especially at small magnitudes.

4- The claim that improvements come specifically from curvature-aware preconditioning, rather than from momentum smoothing of the gradient signal, is not isolated experimentally. An ablation study comparing DP-FedSOFIM to a version that only uses the EMA step without the Fisher preconditioning, which is a DP momentum gradient descent method, would help clarify the contributions.

**Requested Changes:**

1- DP-FedAdam and DP-FedYogi operate at  $\mathcal{O}(d)$  server-side cost using only privatized gradients. They also have no extra privacy cost due to the same post-processing argument. The paper needs to include these methods as baselines to highlight what is specifically gained from the Fisher proxy structure compared to adaptive moment estimation. Without this comparison, the empirical contribution is incomplete.

2- I would suggest that the authors provide an ablation to isolate the Fisher preconditioning from the EMA momentum. Add an ablation condition where the server applies momentum averaging (Equation 6) but replaces the preconditioned update $H_tG_t$ with a standard gradient step using $G_t$ or $M_t$ directly. This is needed to show that the accuracy gains come from curvature awareness instead of gradient smoothing.

3- All accuracy numbers in Tables 2 and 3 seem to come from single runs. The small improvements on CIFAR-10 with tight privacy, for example, $+0.78\%$ at $\epsilon = 1$, need variance estimates to be understandable. The authors should report the mean $\pm$ standard deviation over at least three separate runs with different random seeds.

4- The parameter $\rho$ controls both the operator bounds in the convergence theorem and the effective step size scaling. Since $\rho = 0.5$ is fixed across all settings, the reader has no information about how robust the method is to this choice. An ablation over $\rho \in \\{0.1, 0.5, 1.0, 5.0\\}$ on at least one dataset and privacy regime should be added.

5- Theorem 4.17 shows that DP-FedSOFIM converges to a floor $\Gamma$ which depends on $\rho$, $\beta$, $C_g$, $d$, and $\nu^2$. The paper needs to clearly derive a similar floor for DP-FedGD under the same assumptions. It should also demonstrate the conditions where DP-FedSOFIM's floor is lower. As it stands, the theory does not prove that DP-FedSOFIM has a better error floor than the baseline, it only shows that it converges.

6- The paper should discuss the practical implications of the early deficit under tight privacy when $T$ is small (e.g., $T \le 20$), and suggest a mitigation strategy (e.g., warm-up phase with pure gradient descent before enabling the Fisher proxy).

---

> ### Author Response · Authors · 2026-05-31
> **Response to Reviewer p27K [Part-1]**
>
> > Regarding the comment "The claim that DP-FedSOFIM is a "second-order" method"
>
> We thank the reviewer for this insightful observation. We agree that the momentum buffer $M_t$ shares similarities with first-moment estimators used in Adam/RMSProp/Yogi. However, DP-FedSOFIM differs in how these statistics are utilized. Adaptive methods such as DP-FedYogi perform coordinate-wise scaling using diagonal moment estimates, whereas DP-FedSOFIM constructs a Fisher inspired rank-one preconditioner $P_t^{-1} = (\rho I + M_t M_t^\top)^{-1},$ which introduces directional preconditioning through the Sherman-Morrison update. We added comparisons against DP-FedYogi and Dp-FedAdam and an ablation study (see Appendix H - H1) separating EMA smoothing from Fisher preconditioning. The results show that while EMA smoothing contributes significantly to optimization stability, the Fisher-style directional preconditioning provides additional gains under stronger privacy noise. For example, on CIFAR-10 with 20 clients and Dirichlet $\alpha=0.5$ at $\epsilon=1$, EMA-only achieves $54.03 \pm 1.91\%$ accuracy, whereas DP-FedSOFIM achieves $61.35 \pm 0.94\%$.
>
> > Reporting confidence intervals or multiple-run statistics
>
> We thank the reviewer for raising this concern. In the revised manuscript, all reported results now include mean $\pm$ standard deviation computed over 3 independent random seeds. We acknowledge that the gain at $\varepsilon=1$ on CIFAR-10 is modest in absolute terms; however, we note that under tight privacy budgets the noise scale is large enough that all methods operate near their performance ceiling, and small absolute differences are consistent with this regime. Across the broader set of evaluations, covering $\varepsilon \in \{0.5, 1, 5, 10\}$, two datasets, and 70 communication rounds, DP-FedSOFIM achieves consistent improvements, particularly at $\varepsilon \geq 5$ where the signal-to-noise ratio is sufficient for curvature information to be exploited reliably. The addition of per-seed statistics allows the reader to directly assess statistical significance for each reported comparison.
>
> > Regarding an ablation study comparing DP-FedSOFIM to a version that only uses the EMA step without the Fisher preconditioning.
>
> We thank the reviewer for this suggestion. In the revised manuscript, we have added an ablation study (Tables 8--9, Figures 5--6) comparing three variants: (i) _Grad-only_ (no EMA, no Sherman--Morrison preconditioning), equivalent to DP-FedGD; (ii) _EMA-only_ (momentum smoothing without Fisher preconditioning); and (iii) _DP-FedSOFIM_ (full). The results on CIFAR-10 and PathMNIST (Non-IID, Dirichlet $\alpha=0.5$) reveal a clear separation of contributions. At tight privacy budgets ($\varepsilon=1$), the Sherman--Morrison preconditioning step provides a substantial lift over EMA-only ($+7.32\%$ on CIFAR-10, $+6.81\%$ on PathMNIST), demonstrating that curvature-aware updates contribute independently of momentum smoothing. At larger privacy budgets ($\varepsilon \geq 5$), where the noise scale is lower, the EMA component accounts for most of the gain and the two variants perform comparably, consistent with the intuition that reliable curvature estimation requires a sufficient signal-to-noise ratio. These results confirm that both components contribute meaningfully, with their relative importance governed by the privacy regime.
>
> > Need to include DP-FedAdam and DP-FedYogi methods as baselines.
>
> We thank the reviewer for this valuable suggestion. Following the reviewer’s recommendation, we added DP-FedYogi as an additional baseline in the revised experiments. DP-FedYogi, similar to DP-FedSOFIM, operates entirely on privatized server-side gradients and incurs no additional privacy cost due to the post-processing property of differential privacy. The new comparisons show that DP-FedSOFIM achieves competitive performance with DP-FedYogi and DP-FedAdam across multiple privacy regimes on PATHMNIST and cifar-10 data set under non-IID partitioning. These comparisons help clarify the empirical contribution of the Fisher inspired directional preconditioning beyond standard adaptive moment estimation based on coordinate wise scaling. We have incorporated these experimental results and discussion in the revised manuscript.

---

> ### Author Response · Authors · 2026-05-31
> **Response to Reviewer p27K [Part-2]**
>
> > Ablation over $\rho=\{0.1, 0.5, 1.0, 5.0\}$
>
> We thank the reviewer for this suggestion. In the revised manuscript, we have added a $\rho$ sensitivity analysis (Table~10) covering $\rho \in \{0.01, 0.1, 0.5, 1.0, 2.0, 5.0, 10.0\}$ on both CIFAR-10 and PathMNIST (Non-IID, Dirichlet $\alpha=0.5$) at $\varepsilon \in \{1, 5\}$. The results demonstrate that DP-FedSOFIM is robust to the choice of $\rho$ across a wide range. On CIFAR-10 at $\varepsilon=5$, performance is stable across $\rho \in [0.5, 2.0]$ (67.32--68.03\%), with degradation only at the extremes ($\rho=0.01$: 59.37\%, $\rho=10.0$: 61.99\%). A complementary trend holds at $\varepsilon=1$, where higher $\rho$ is preferable as increased momentum better suppresses the large Gaussian noise; performance increases monotonically from 51.08\% at $\rho=0.01$ to 61.35\% at $\rho=10.0$. The same pattern holds on PathMNIST, with the optimal $\rho$ shifting to $0.5$ at $\varepsilon=5$ (71.32\%) and $10.0$ at $\varepsilon=1$ (68.48\%), consistent with the noise-dominance interpretation. Based on these results, we select $\rho=1.0$ as the default for $\varepsilon \geq 5$ and $\rho=10.0$ for tight privacy budgets, and report these values explicitly in the revised hyperparameter table.
>
> > Demonstrating the conditions where DP-FedSOFIM's floor is lower.
>
> We agree with the reviewer. The previous version proved convergence of DP-FedSOFIM to a bias-and-noise neighborhood, but it did not derive the corresponding DP-FedGD floor under the same assumptions. Therefore, the comparison was incomplete. We have added a new subsection, ``Comparison with the DP-FedGD Error Floor." In this subsection we derive the one-step descent inequality for DP-FedGD under the same clipping-bias and DP-noise assumptions. This gives the baseline floor $ \frac{\Gamma_{\rm GD}}{2\mu\eta c_{\rm GD}}, \,\Gamma_{\rm GD} = \frac{\eta\zeta_{\max}^2}{2\tau} +
> \frac{L\eta^2}{2}(C_g^2+d\nu^2), \, c_{\rm GD}=1-\frac{\tau}{2}.$ We then compare this expression with the DP-FedSOFIM floor.
>
> We also clarify that our present worst-case theory does not prove that DP-FedSOFIM has a uniformly smaller error floor than DP-FedGD in all regimes. The DP-FedSOFIM bound contains additional terms due to the same-step coupling between the privatized aggregate gradient and the adaptive preconditioner. Thus, the theorem proves privacy-preserving convergence and stability of the preconditioned method, but the lower floor requires additional alignment or preconditioner-quality conditions. We now state this explicitly.
>
> To make the comparison precise, we added a discussion showing that DP-FedSOFIM can have a smaller floor when the preconditioner improves the effective descent coefficient and/or reduces the effective variance sufficiently to dominate the additional coupling term. This is the regime expected in ill-conditioned problems with stable dominant curvature directions, and it is consistent with the empirical gains observed on PathMNIST and in the faster-convergence results.
>
> > Practical implications of the early deficit under tight privacy when $T$ is small
>
> We thank the reviewer for this suggestion. Under stringent privacy budgets ($\varepsilon=0.5$), the large injected Gaussian noise dominates the early aggregated gradients, causing the momentum buffer $M_t$ to be unreliable in initial rounds; constructing the Fisher proxy from an unstable $M_t$ can temporarily slow optimization, which is particularly harmful when $T$ is small. In the revised manuscript, we have adopted exactly the mitigation the reviewer suggests: a warmup phase of 20 rounds in which the server applies standard EMA-only updates before activating the Sherman--Morrison preconditioning step. This allows $M_t$ to accumulate sufficient gradient signal before curvature information is introduced. The warmup is applied at $\varepsilon=0.5$ on both datasets, is documented in the Hyperparameters paragraph of Section 5, and is discussed mechanistically in the analysis of Phenomenon 1. Empirically, DP-FedSOFIM with warmup exhibits no early-round instability even at $\varepsilon=0.5$, and remains competitive from round 10 onward.

---

> ### Comment · Reviewer_p27K · 2026-06-06
>
> Thank you for addressing several of the concerns raised previously. However, the ablation results in Tables 8 and 9 show a pattern that the manuscript discussion does not fully explain. On CIFAR-10, EMA-only does better than the full DP-FedSOFIM at $\varepsilon = 0.5$ (59.04% compared to 58.43%) and at $\varepsilon = 5$ (67.97% compared to 67.32%). Similarly, on PathMNIST at $\varepsilon = 0.5$, EMA-only achieves 65.66% while the full method reaches 65.24%. The authors mention in their response that "the Sherman-Morrison preconditioning step provides a substantial lift over EMA-only at $\varepsilon = 1$," which the data supports. However, the claim that "both components contribute meaningfully" is not consistently supported across all conditions.
>
> I would suggest that the authors clearly acknowledge in the manuscript the settings where EMA-only matches or surpasses the full method and provide a clearer description of when the SM preconditioning step helps and when it does not.

---

> > ### Author Response · Authors · 2026-06-15
> > **Response to Reviewer p27K**
> >
> > We thank the reviewer for this careful reading of the ablation results. We agree that the original claim requires qualification. We have therefore revised the discussion in Appendix H.1 of the revised manuscript uploaded to explicitly acknowledge the settings in which EMA-only matches or exceeds the performance of the full DP-FedSOFIM method.
> >
> > The Sherman-Morrison (SM) curvature correction provides its largest and most consistent benefit at $\varepsilon=1$, where the noise level is high enough to corrupt isotropic gradient steps yet low enough for the momentum buffer to accumulate a stable curvature signal---yielding gains of approximately 7 percentage points over EMA-only on both datasets.
> >
> > At the tightest budget ($\varepsilon=0.5$), per-round Gaussian noise dominates any directional curvature signal and the SM step provides no additional benefit beyond EMA smoothing. A similar saturation effect occurs on CIFAR-10 at $\varepsilon=5$: the diffuse curvature landscape (Phenomenon 5) means the dominant gradient directions captured by the EMA buffer already span the relevant subspace, leaving little residual signal for the rank-one Fisher proxy. On PathMNIST at $\varepsilon\ge5,$ the concentrated curvature structure continues to reward explicit preconditioning and the full method retains a consistent advantage.
> > So, the SM step is most beneficial when (i) privacy noise is moderate to high _and_ (ii) the loss landscape has structured anisotropy that the rank-one proxy can capture. It provides limited or no benefit when noise is so large that curvature estimation is unreliable, or when the landscape curvature is diffuse enough that EMA-smoothed gradient directions already cover the informative subspace.

---

### Review · Reviewer_7p1P · 2026-05-19

**Summary Of Contributions:**

This paper proposes a rank one second order differentially private optimizer for the federated setting. Theory is proven in the strongly convex setting (ie frozen feature maps of ResNets) that the algorithm should converge and maintains differential privacy under a slightly relaxed setting. Experiments are shown which generally demonstrate that performance is better than a couple of alternatives under a fixed privacy budget.

**Additional Comments:**

n/a

**Audience:**

Yes

**Audience Explanation:**

Presentation:

Currently the paper feels extremely bloated for what it is – a pretty straightforward application of momentum / natural gradient descent for federated differentially private learning. Please cut out a lot of the proofs and assumptions, moving them into the Appendix to try to get under 12 pages.

-	Write lemma 4.10 and its proof over 3-4 lines.

-	Lemma 4.12 – don’t Eq number equations that will not be referenced. Again, clean up the spacing.

-	Lemma 4.14 – same as 4.12

-	Section 4.6 – you don’t need to spell out the PL condition and its proof in the main text. Honestly, neither is really necessary for the main text.

-	Page 15 – don’t have definitions be single line equations. In line them.
-	Figures 1 and 2 have so many lines on them as to be completely illegible. Please place a and b panels side by side with fewer epsilon options. You can put the learning curves for other ones in the appendix.


Tables 1 and 2: is the epsilon budget total or per round for DP-FedSOFIM?  My understanding is that your budget would be rounds * 10 budget for these experiments in terms of privacy, while the others would be epsilon = 10.

Methodology:

Reference to Andrew et al, ’21 feels wrong – there’s no discussion of MedMNIST or PathMNIST in there, just EMNIST? I didn't check the other references but I wouldn't be surprised if something else was a bit off.

I’m surprised there’s no comparison to FedAvg or its differentially private variants?

-	I think you need L2 regularization in the experiments to make the classification strongly convex.


Missing discussion with various ADMM works for deep learning literature; some of these are even in the federated learning setting (https://arxiv.org/abs/2506.13150, https://arxiv.org/abs/2107.10884, https://jmlr.org/papers/v22/20-1006.html, https://arxiv.org/abs/2310.19807). I mention the first because it is also federated and uses ADMM, while the last is precisely your setting but without the differential privacy necessities.

**Claims And Evidence:**

No

**Claims Explanation:**

-	I think it’s a bit weak to only be able to claim \sum epsilon, \sum delta privacy when the traditional differential privacy is epsilon, delta. By comparison, it seems like the privacy proofs in DP-FedNew (Krouka et al, ’25) are a bit stronger – demonstrating privacy at both the user and record level.

o	How and why can we make assumption 4.24?

-	In general, as a rank-1 approximation to even the empirical Fisher information (which isn’t a great approximation to natural gradient descent – see https://arxiv.org/abs/1905.12558), it’s not clear that this is really “second-order” but really something more like momentum/adaptive step-size. Thus, I think the connection to “second-order” optimization is over sold. This is even pointed out in Appendix C a little bit – as the approached is really described as an exponential moving average of momentum (which it is).

o	This plays into implementation details – as written your algorithm takes up d^2 space even if there’s only d communications from the buffer. Yet if we continue working out the Sherman-Morrison lemma, we see that L14 in Algorithm 1: $g_{t+1} = g_{t} - \eta_t (\frac{g_t}{\rho} - \frac{(M_t’ g_t)}{\rho^2 + \rho ||M_t||^2} M_t )$ rather than forming the full Hessian approximation in L13.

o	Furthermore, the convergence theory doesn’t really seem to give accelerated rates over non-preconditioned optimization (ie DP-FedGD may give better rates). I think this is because of the operator norm bound on the Hessian. Also, I think Eq. 30 can become vacuous when ‖M_t‖² > ρ.

o	I think you’ll probably get better analysis out of ADMM and federated ADMM theory (see references at bottom of next section) rather than Martens ’20.

o	Lower learning rates imply worse error rates because the rate is O(1/p).

o	I wonder if delaying the preconditioner would help the theory – if you don’t want to go down the route of momentum / ADMM theory in general.

**Requested Changes:**

see above.

---

> ### Author Response · Authors · 2026-05-31
> **Response to Reviewer 7p1P [Part-1]**
>
> > Regarding " claim \sum epsilon, \sum delta privacy"
>
> We agree that presenting only the elementary $(\sum_t\varepsilon_t,\sum_t\delta_t)$ composition bound makes the privacy guarantee appear weaker than intended. In the revision, we have clarified that this bound is only a simple-composition baseline. The actual privacy guarantee of DP-FedSOFIM is accountant-agnostic: whatever final $(\varepsilon,\delta)$ guarantee is certified for the underlying privatized gradient-release sequence is inherited unchanged by DP-FedSOFIM, because all SOFIM computations are post-processings of already privatized aggregate gradients.
>
> _Change in manuscript_: We added a theorem stating privacy preservation under an arbitrary valid privacy accountant. This makes clear that RDP, GDP, moments-accountant, or hockey-stick-divergence accounting can be used exactly as in DP-FedGD. We also
> clarified that our present experiments use record-level DP, while user-level DP can be obtained by replacing record-level clipping/noising with user-level client-update clipping/noising. The server-side SOFIM step is unchanged under either adjacency notion.
>
> > On Assumption 4.24 and rank-1 approximation
>
> We thank the reviewer for this insightful observation. We agree that SOFIM does not approximate the full Fisher matrix and therefore does not recover the exact natural gradient. We acknowledge that it is not equivalent to momentum or adaptive step size methods. The defining feature of DP-FedSOFIM is the use of an explicit Fisher inspired preconditioner and its inverse in the update rule. Although the preconditioner is rank-1, the update remains curvature aware and differs fundamentally from standard momentum, which does not perform matrix based preconditioning. Moreover, many widely accepted second-order optimizers rely on structured approximations (diagonal, block-diagonal, low-rank, or Kronecker-factored) rather than exact curvature matrices. DP-FedSOFIM follows the same philosophy, trading approximation fidelity for computational efficiency. We therefore view DP-FedSOFIM as a lightweight Fisher preconditioned second order approximation rather than a purely first order momentum method. We further investigated this point through an ablation study (see Apendix H) separating the effects of (i) exponential moving average (EMA) smoothing and (ii) curvature based preconditioning. Specifically, we compared:
> (i) Grad-only (no EMA, no preconditioning),
> (ii) EMA-only (no Sherman-Morrison/Fisher preconditioning),
> (iii) full DP-FedSOFIM.
> The results indicate that EMA smoothing alone does not fully explain the observed gains.
>
> > Algorithm taking $O(d^2)$ space
>
> We thank the reviewer for this important observation and for carefully pointing out the discrepancy between Algorithm 1 and the implementation described in Section 3.4.3. We agree that Algorithm 1, as originally written, may give the impression that the curvature proxy $I_t=M_tM_t^\top+\rho I_d$ is explicitly formed and stored as a $d\times d$ matrix, implying an $O(d^2)$ memory and computational complexity. However, this is not how DP-FedSOFIM is implemented. As derived in Section~3.4.3, we exploit the Sherman--Morrison identity to avoid explicitly constructing the matrix inverse. Specifically, $H_t=I_t^{-1} = \frac1\rho I_d- \frac{M_tM_t^\top} {\rho^2+\rho\|M_t\|^2},$ and directly applying this inverse to the aggregated gradient yields $H_tG_t =
> \frac{G_t}{\rho} - \frac{M_t(M_t^\top G_t)} {\rho^2+\rho\|M_t\|^2}.$
>
> Consequently, the server update is implemented as $\theta_{t+1} = \theta_t - \eta_t H_t G_t,$ which requires only scalar inner products and vector operations. Therefore, both memory and computational complexity are \(O(d)\), and no \(d\times d\) matrix is ever explicitly constructed. We had also discussed these in detail in Appendix D. The reviewer is correct that the ambiguity arises from the presentation of Algorithm~1 rather than the implementation itself. We have revised the algorithm pseudocode in the manuscript to explicitly use the Sherman--Morrison form and remove this potential confusion.

---

> ### Author Response · Authors · 2026-05-31
> **Response to Reviewer 7p1P [Part-2]**
>
> > On FedGD having better rates and implications of  $\|M_t\|^2 > \rho$.
>
> We thank the reviewer for pointing out that the previous lower bound on $\nabla F(\theta_t)^\top H_t\nabla F(\theta_t)$ could become vacuous when $\|M_t\|^2 > \rho$. We have revised the preconditioner analysis to make the non-vacuous spectral behavior explicit. We revised Lemma 4.14 to include the exact spectral lower bound. We also added a corollary which shows that for any $T\ge1$ and $\delta_M\in(0,1)$, with probability at least $1-\delta_M$, $\max_{0\le t<T}\|M_t\|_2^2 \le \frac{T\bar M^2}{\delta_M}.$ On this high-probability event, the descent coefficient can be bounded below by $\frac{1}{\rho+T\bar M^2/\delta_M}.$ Thus the revised analysis separates the conservative expectation recursion from a non-vacuous high-probability spectral bound for the preconditioner.
>
> We have also clarified that our current worst-case theory does not prove an accelerated rate over DP-FedGD. The theoretical result establishes stability and convergence of the privacy-compatible preconditioned update. Empirical gains are reported separately, and a sharper rate comparison would require additional alignment or preconditioner-quality assumptions.
>
> > Regarding better analysis out of ADMM and federated ADMM theory.
>
> We thank the reviewer for the suggestion. Federated ADMM is an important line of work, especially for consensus-constrained and primal-dual formulations of federated optimization. However, DP-FedSOFIM is not an ADMM-type method: it does not introduce local consensus variables, augmented Lagrangian penalties, or dual updates. Its update is a server-side preconditioned gradient step applied to an already privatized aggregate gradient. We therefore believe that adaptive preconditioning and empirical-Fisher-type optimization are the more appropriate theoretical points of comparison.
>
> > Lower learning rates imply worse error rates because the rate is $O(1/p)$.
>
> We are clarifying the dependence on the stepsize. In the strong-convexity bound the limiting neighborhood is $\frac{\Gamma}{2\mu\eta c_\nabla}.$ Since $\Gamma$ contains both $O(\eta)$ and $O(\eta^2)$ terms, this floor is not simply increasing as $\eta$ decreases. The terms due to clipping bias, privacy noise, and same-step coupling remain of constant order after division by $\eta$, while the smoothness term scales as $O(\eta)$. Smaller learning rates therefore slow the transient contraction but do not necessarily worsen the asymptotic floor.
>
> > If delaying the preconditioner would help the theory.
>
> We agree that delaying or gradually introducing the preconditioner is a useful stabilization strategy, especially in the early rounds when $M_t$ is estimated from very few noisy privatized aggregates. We have added this as a practical variant and future empirical direction. We added a discussion of a warm-start preconditioner.
>
> > Presentation
>
> We thank the reviewer for the important suggestions. We have made the requested changes, specially, moving all the long proofs to the appendix, and also have taken care of the equation numbering issues. In the revised manuscript, Figures 1 and 2 have been restructured into a $2\times2$ panel layout showing all four privacy regimes ($\varepsilon \in \{0.5, 1, 5, 10\}$) with each method plotted as a distinct curve. The IID convergence trajectories, which show broadly consistent trends, have been moved to Appendix G to reduce clutter in the main paper. The reported $\epsilon$ values in Tables 3-4 (in revised manuscript) correspond to the _total end-to-end privacy budget_ after all communication rounds, not a per-round privacy cost. Specifically, for each target privacy level $\epsilon \in {0.5,1,5,10}$, we calibrate the Gaussian noise multiplier using the hockey-stick divergence accountant such that the composed privacy loss across all $T$ federated rounds satisfies the final $(\epsilon,\delta)$ guarantee. Therefore, $\epsilon=10$ means that the entire training process satisfies a total privacy budget of 10, rather than incurring a privacy budget of $T \times 10$.
>
> > Methodology comments
>
> We thank the reviewer for the comments regarding the methodology. We agree that the citation to Andrew et al. (2021) was misplaced; that work concerns adaptive clipping on EMNIST and is unrelated to PathMNIST or MedMNIST. In the revised manuscript, PathMNIST is correctly attributed to the MedMNIST benchmark (Yang et al., 2023). We have also audited all remaining citations in the experimental section and corrected any further misattributions. In the revised manuscript, we have also added DP-FedAvg as an additional baseline. We have added L2 regularization ($\lambda = 10^{-4}$) to all classification objectives across all methods and datasets, ensuring the strongly convex setting assumed in our theoretical analysis. The updated experimental results are reported in Tables 3-4 and Figures 1-2.

---

### Decision · Action_Editor_7gYN · 2026-06-19

**Recommendation:** Accept as is

**Audience:**

Yes

**Audience Explanation:**

Accelerating convergence in Differentially Private Federated Learning (DP-FL) under tight communication budgets is a highly active and critical problem for the TMLR community. This paper provides a practical solution by constructing a server-side, rank-one Fisher Information Matrix proxy that requires only O(d) memory and computation. Because it operates entirely as a post-processing step on already-privatized gradients, it consumes zero additional privacy budget, which is a structural advantage over client-side alternatives.

**Claims And Evidence:**

Yes

**Claims Explanation:**

The revised manuscript fully satisfies TMLR’s soundness criteria by thoroughly addressing all reviewers' concerns through a comprehensive update. The authors expanded the empirical evaluation from 3 to 8 baselines under realistic Non-IID Dirichlet settings across two backbones, while validating the results with multi-seed standard deviations and McNemar’s statistical significance tests. Furthermore, the theoretical framework was extended to include a smooth non-convex stationarity theorem.